# BMP suppresses Wnt signaling via the Bcl11b-regulated NuRD complex to maintain intestinal stem cells

Yehua Li [ID][1,5], Xiaodan Wang [ID][1,5], Meimei Huang [ID][1,5], Xu Wang[2], Chunlin Li[1], Siqi Li[2], Yuhui Tang[1], Shicheng Yu[2], Yalong Wang [ID][2], Wanglu Song [ID][1], Wei Wu[3], Yuan Liu [ID][1 ✉] & Ye-Guang Chen [ID][1,2,4 ✉]

## Abstract

**Lgr5+ intestinal stem cells (ISCs) are crucial for the intestinal epithelium renewal and regeneration after injury. However, the mechanism underlying the interplay between Wnt and BMP signaling in this process is not fully understood. Here we report that Bcl11b, which is downregulated by BMP signaling, enhances Wnt signaling to maintain Lgr5+ ISCs and thus promotes the regeneration of the intestinal epithelium upon injury. Loss of *Bcl11b* function leads to a significant decrease of Lgr5+ ISCs in both intestinal crypts and cultured organoids. Mechanistically, BMP suppresses the expression of Bcl11b, which can positively regulate Wnt target genes by inhibiting the function of the Nucleosome Remodeling and Deacetylase (NuRD) complex and facilitating the β-catenin-TCF4 interaction. Bcl11b can also promote intestinal epithelium repair after injuries elicited by both irradiation and DSS-induced inflammation. Furthermore, *Bcl11b* deletion prevents proliferation and tumorigenesis of colorectal cancer cells. Together, our findings suggest that BMP suppresses Wnt signaling via Bcl11b regulation, thus balancing homeostasis and regeneration in the intestinal epithelium.**

**Keywords** Bcl11b; Intestinal Stem Cells; BMP Signaling; Wnt Signaling; Colorectal Cancer

**Subject Categories** Signal Transduction; Stem Cells & Regenerative Medicine

## Introduction

The constant renewal of the intestinal epithelium relies on stem cells located at the base of crypts (Clevers, 2013; van der Flier and Clevers, 2009). These actively proliferating stem cells are marked by the Wnt target gene *Lgr5* (Barker et al, 2007). Lgr5+ intestinal stem cells (ISCs) are not only crucial for the daily renewal of intestinal epithelium but also play a significant role in radiation-induced intestinal regeneration (Metcalfe et al, 2014). The number of Lgr5+ ISCs in each crypt is tightly controlled by the growth factors in the niche. Wnt/β-catenin signaling is required for the pluripotency maintenance of Lgr5+ ISCs (Clevers et al, 2014). Loss of *Tcf4*, a critical mediator of Wnt/β-catenin signaling, prevents the establishment of proliferative crypts in mice during late gestation (Korinek et al, 1998). Disruption of Wnt/β-catenin signaling by other means, such as conditional deletion of *Ctnnb1* (Fevr et al, 2007; Ireland et al, 2004) or transgenic overexpression of the Wnt inhibitor Dickkopf-1 (Dkk1) (Kuhnert et al, 2004; Pinto et al, 2003), further emphasizes the essential role of the canonical Wnt pathway in maintaining intestinal epithelium homeostasis. Lgr5 functions as the coreceptor for Wnt agonist R-spondin1 (Rspo-1) (de Lau et al, 2011), and Ascl2 amplifies the Wnt/Rspo-1 signaling through an autoactivation loop (Schuijers et al, 2015a; van der Flier et al, 2009), all of which underscore the important role of Wnt signaling in Lgr5+ ISCs. In addition to Wnt signaling, we and others have showed that bone morphogenic protein (BMP) signaling inhibits both pluripotency and proliferation of ISCs by suppressing their signature gene expression (He et al, 2004; Qi et al, 2017).

B-Cell CLL/Lymphoma 11B (Bcl11b) is a zinc finger transcription factor that plays critical roles in T cell development (Ikawa et al, 2010; Li et al, 2010a; Wakabayashi et al, 2003), odontogenic (Golonzhka et al, 2009), epidermal homeostasis (Golonzhka et al, 2009), and other biological processes (Liu et al, 2010). Bcl11b has been shown to mark a quiescent stem cell population in the mammary gland, and its conditional deletion leads to a loss of epithelial cell regeneration capacity (Cai et al, 2017).

The NuRD complex mediates gene repression through histone deacetylase activity and chromatin remodeling. This complex is composed of at least seven proteins, including metastasis tumor-associated (MTA) 1, 2, or 3, histone deacetylase (HDAC) 1 or 2, retinoblastoma-binding protein (RBBP) 4 or 7, GATA zinc finger domain (GATAD) 2A or B, chromodomain helicase DNA binding protein (CHD) 3, 4, or 5, cyclin-dependent kinase 2-associated protein 1 (CDK2AP1), and methyl-CpG-binding domain proteins

[1]The State Key Laboratory of Membrane Biology, Tsinghua-Peking Center for Life Sciences, School of Life Sciences, Tsinghua University, Beijing 100084, China. [2]Guangzhou National Laboratory, Guangzhou 510700, China. [3]MOE Key Laboratory of Protein Sciences, School of Life Sciences, Tsinghua University, Beijing 100084, China. [4]The MOE Basic Research and Innovation Center for the Targeted Therapeutics of Solid Tumors, School of Basic Medical Sciences, Jiangxi Medical College, Nanchang University, Nanchang 330031, China. [5]These authors contributed equally: Yehua Li, Xiaodan Wang, Meimei Huang. ✉E-mail: liu-yuan@mail.tsinghua.edu.cn; ygchen@tsinghua.edu.cn

(MBD) 2 or 3 (Leighton and Williams, 2020). Several components of the NuRD complex are highly expressed in various types of cancer and stem cells (Aguilera et al, 2011; Huang et al, 2021; Li et al, 2019; Li et al, 2020b; Mor et al, 2018). For instance, RBBP4 promotes colon cancer development by promoting Wnt signaling (Li et al, 2020b).

In this study, we provide evidences showing the critical role of BMP-suppressed gene *Bcl11b* in maintenance of Lgr5$^+$ ISCs. Bcl11b activates Wnt signaling by enhancing the β-catenin/TCF4 interaction. In addition, Bcl11b directly binds to Wnt target genes and disrupts the NuRD complex, thus activating these target genes. Consistently, the regeneration capacity of the intestinal epithelium in *Bcl11b*-deficient mice is significantly reduced. Furthermore, *Bcl11b* deletion impairs tumorigenesis of colon cancer cells. Collectively, our results indicate that Bcl11b bridges BMP signaling and Wnt signaling to modulate intestinal epithelium homeostasis.

## Results

### BMP inhibits *Bcl11b* expression in Lgr5$^+$ ISCs

Previously, we showed that BMP could directly suppress ISC signature genes in mouse Lgr5$^+$ ISCs (Qi et al, 2017). Among these genes downregulated by BMP, *Bcl11b* was confirmed to be inhibited by BMP and the inhibitory effect was abolished by BMP receptor *Alk3* knockout (KO) in cultured organoids (Fig. 1A). Consistently, *Bcl11b* was upregulated in *Alk3* cKO intestinal epithelium (Fig. 1B). By analyzing our single cell-RNA-seq data (Liu et al, 2023) and our spatial RNA-seq data (see the Methods), we found that *Bcl11b* expression was enriched in Lgr5$^+$ ISCs located at the bottom of the crypts (Fig. 1C–E).

To confirm its expression in Lgr5$^+$ ISCs, we isolated GFP$^{high}$ stem cells and GFP$^{low}$ transit-amplifying (TA) cells from the crypts of *Lgr5-GFP-IRES-creERT2* mice, and performed quantitative RT-PCR (qRT-PCR) analysis. The results revealed high expression of *Bcl11b* in stem cells, lower in TA cells and the lowest in villus, a pattern similar to the reported ISC signature genes *Lgr5*, *Ascl2*, and *Olfm4* (Fig. 1F). Further analysis of BCL11B protein in control and *Alk3* cKO mice validated the negative regulation of *Bcl11b* by BMP, as BCL11B protein level increased in the bottom crypts of *Alk3* cKO mice (Fig. 1G; Appendix Fig. S1A). Moreover, BMP treatment rapidly suppressed Bcl11b expression as early as the induction of *Id1*, a well-known BMP target gene (Appendix Fig. S1B). We further uncovered the direct binding of the BMP downstream transcription factors Smad1 and Smad4 to the *Bcl11b* gene promoter, confirming the direct regulation of *Bcl11b* by BMP signaling (Fig. 1H). Together, these results suggest that BMP directly inhibits *Bcl11b* expression in Lgr5$^+$ ISC.

### *Bcl11b* is critical for Lgr5$^+$ stem cell maintenance

To explore the physiological function of *Bcl11b* in the intestinal epithelium, we generated *Villin-CreERT2;Lgr5-GFP-IRES-CreERT2;Bcl11b$^{fl/fl}$* (indicated as *Bcl11b* cKO) mice by crossing *Bcl11b$^{fl/fl}$* mice (Li et al, 2010b) with *Villin-CreERT2;Lgr5-GFP-IRES-CreERT2* mice to achieve inducible conditional deletion of *Bcl11b* in the intestinal epithelium. To avoid the possible stem cell toxicity induced by Cre activation (Bohin et al, 2018), we used

*Villin-CreERT2;Lgr5-GFP-IRES-CreERT2* mice as control. After tamoxifen injection, *Bcl11b* was completely knocked out (Appendix Fig. S1C). Although the overall architecture of the epithelium, the cell proliferation and differentiation were not noticeably changed in *Bcl11b* cKO mice (Appendix Fig. S1D), we observed a significant decrease of Lgr5$^+$ cells in the intestinal epithelium (Fig. 2A; Appendix Fig. S1E). Most of the examined ISC signature genes were also downregulated in *Bcl11b*-deficient crypts (Fig. 2B).

Consistently, the number of buddings was decreased in 4-hydroxytamoxifen (4-OHT)-induced *Bcl11b* cKO organoids (Fig. 2C), indicating reduced stem cell activity. This finding was supported by decreased Lgr5$^+$ cell number and *Olfm4* expression in organoids derived from the crypts of *Bcl11b* cKO mice (Fig. 2D; Appendix Fig. S1F). To better assess the stemness of ISCs, we sorted single Lgr5-GFP$^+$ stem cells from control or *Bcl11b* cKO mice using flow cytometry and checked the colony formation of single cells. The single Lgr5-GFP$^+$ cells were seeded into plates and treated with 4-OHT to induce *Bcl11b* cKO. As expected, *Bcl11b* deletion attenuated the colony formation capacity of isolated single Lgr5$^+$ ISCs (Fig. 2E; Appendix Fig. S1G). In line with this, the expression of Lgr5$^+$ ISC signature genes was significantly reduced in the *Bcl11b* knockout organoids (Fig. 2F).

To further examine the function of *Bcl11b* in Lgr5$^+$ ISCs, we overexpressed Bcl11b in the organoids derived from mouse small intestine. After doxycycline (Dox)-induced overexpression of *Bcl11b* (Appendix Fig. S1H), we observed faster growth and more buddings of organoids as well as elevated expression of ISC signature genes (Fig. 2G,H; Appendix Fig. S1I). These data support the promoting function of *Bcl11b* in Lgr5$^+$ ISC maintenance. Moreover, *Bcl11b* overexpression counteracted the growth inhibition and signature gene downregulation induced by the withdrawal of the BMP antagonist Noggin (Fig. 2I,J). Taken together, these data suggest that Bcl11b is required for Lgr5$^+$ stem cell maintenance during intestinal epithelium homeostasis and mediates the suppression of Lgr5$^+$ ISCs by BMP.

### Bcl11b is required for intestinal epithelium regeneration after injury

Given the crucial role of Lgr5$^+$ stem cells in the regeneration of the intestinal epithelium (Liu and Chen, 2020; Metcalfe et al, 2014), we hypothesized that *Bcl11b* may function in the Lgr5$^+$ ISC-mediated repair of intestinal epithelium injuries as well. We first examined *Bcl11b* expression after *Lgr5-GFP-IRES-CreERT2* mice were exposed to a low dose X-ray (6 Gy), which does not totally ablate Lgr5$^+$ ISCs. The expression of *Bcl11b* and other ISC signature genes were analyzed in the sorted GFP$^+$ stem cells at different times. Both *Bcl11b* and the signature genes dramatically decreased 1 day after X-ray irradiation, and then recovered to the control level at day 3 post-irradiation (Fig. 3A), faster than other stem cell signature genes. Among Bcl11b$^+$ cells, Lgr5$^-$ population was more stable than Lgr5$^+$ cells upon X-ray irradiation (Fig. 3B). Together, these data suggest that *Bcl11b* may play important role in intestinal epithelium regeneration.

To further explore the function of *Bcl11b* in epithelial regeneration, we compared the recovery of injured intestinal epithelium in control and *Bcl11b* cKO mice after X-ray irradiation (Fig. 3C). Upon irradiation, the recovery of intestinal epithelium and body weight was impaired in *Bcl11b* knockout mice at day 11,

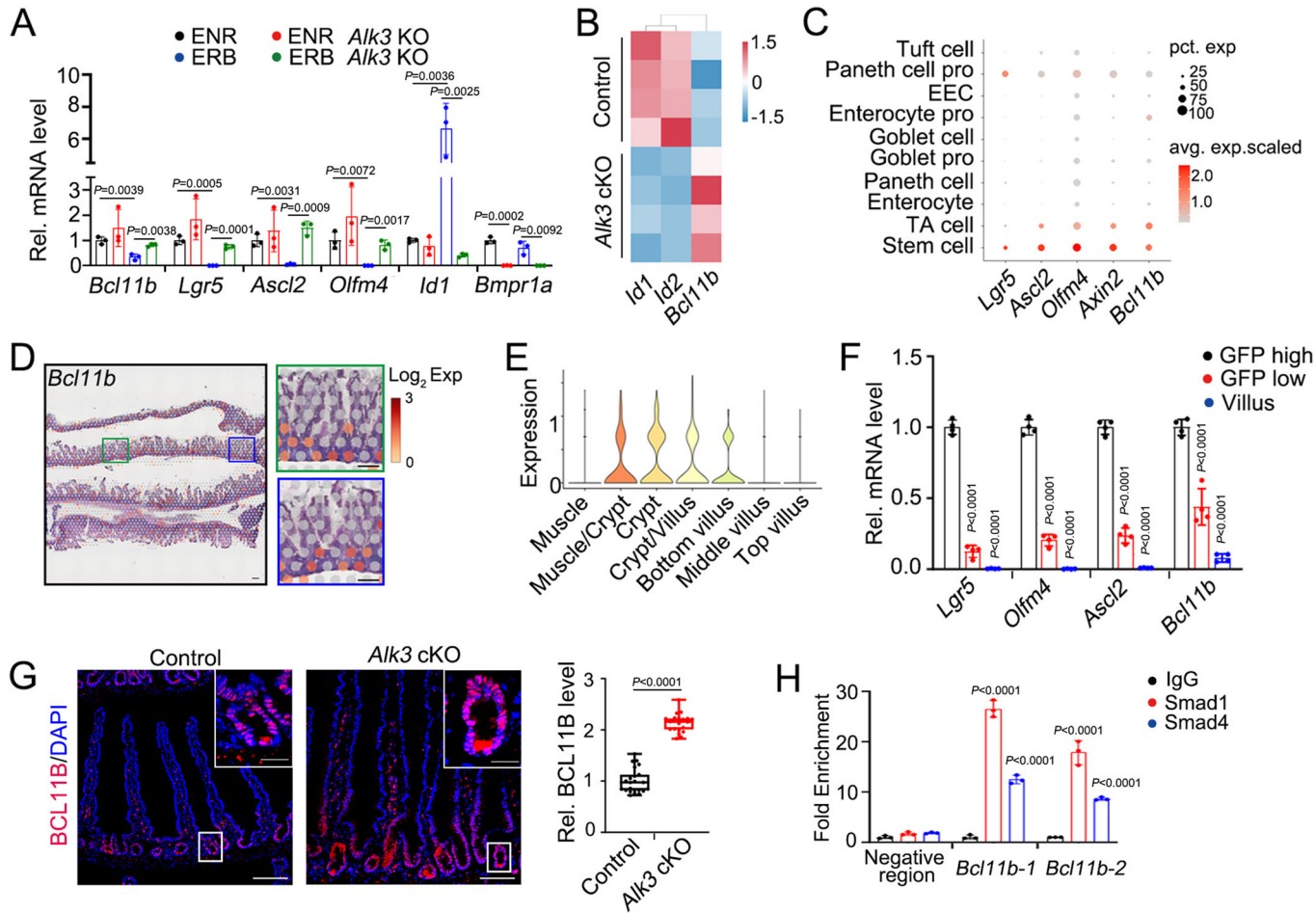

**Figure 1.  BMP inhibits *Bcl11b* expression in Lgr5$^+$ ISCs.**

(A) Organoids derived from the crypts of *Vil-CreERT2* or *Vil-CreERT2; Alk3$^{fl/fl}$* mice were treated with 4-hydroxytamoxifen (4-OHT) to delete *Alk3* for 24 h. Then the organoids were treated with ENR or ERB for 2 days and harvested for analysis of stem cell signature gene expression by qRT-PCR. E, EGF, 50 ng/mL; N, Noggin, 100 ng/mL; R, R-spondin 1, 500 ng/mL; B, BMP4, 20 ng/mL. $n$ = three independent experiments. (B) Heatmap of indicated gene expression in the intestinal epithelium of *Vil-CreERT2* (control) and *Vil-CreERT2;Alk3$^{fl/fl}$* (*Alk3* cKO) mice after tamoxifen injection. $n$ = 4 mice/group. GSE247044 was used for analysis. (C) scRNA-seq analysis showing the gene expression level (color scale) and expressing cells (point diameter) in each cell cluster of the intestinal epithelium. GSE186917 was used for analysis. (D) Hematoxylin and eosin (H&E) staining and integration of *Bcl11b* transcriptome onto visium datasets. Scale bar, 200 μm. (E) Expression pattern of *Bcl11b* transcriptome in different regions onto visium datasets. $n$ = 1 dataset. (F) The expression of *Lgr5, Olfm4, Ascl2* and *Bcl11b* were examined by qRT-PCR in Lgr5-high and -low cells sorted from *Lgr5-GFP-IRES-creERT2* mice through FACS and isolated villus. $n$ = 4 mice/group. (G) Immunofluorescence staining (left) and quantification (right) for BCL11B protein in intestine of *Vil-CreERT2* (control) and *Vil-CreERT2;Alk3$^{fl/fl}$* (*Alk3* cKO) mice. Each dot represents one image measurement, sampled across at least 21 tissue sections from $n$ = 3 mice/group. Scale bar, 100 μm (low magnification) and 25 μm (high magnification). The central line represents the median. The box spans from the 25th to the 75th percentile. Whiskers extend to the smallest and largest values. All points are presented. (H) ChIP-qPCR analysis of Smad1 or Smad4 binding at *Bcl11b* binding regions in Lgr5$^+$ ISCs. Fold change is shown as relative to normal-IgG control. $n$ = three independent experiments. Data represent mean ± SD (A, F, H). The data were analyzed by unpaired t test with Welch-correction (G) and Two-way ANOVA with Tukey's multiple comparisons test (A, F, H). The exact P value is displayed. Source data are available online for this figure.

characterized by a significant reduction in regenerated crypts and lower body weight (Fig. 3C,D; Appendix Fig. S2A). Consistently, cell proliferation was also hindered dramatically in *Bcl11b* cKO mice (Fig. 3E; Appendix Fig. S2B). In DSS-induced colitis model, *Bcl11b* knockout also led to more pronounced body weight loss, increased epithelium damage, as well as impaired cell proliferation (Fig. 3F,G; Appendix Fig. S2C). Moreover, *Bcl11b* overexpression rescued the impaired architecture and reduced *Lgr5* expression following irradiation injury in cultured organoids (Fig. 3H,I). These experiments suggest that Bcl11b is required for intestinal epithelium regeneration by maintaining Lgr5$^+$ ISCs.

## BCL11B activates Wnt/β-catenin signaling

To elucidate how Bcl11b acts in ISCs, we examined its effect on the regulatory signaling of ISCs. Bcl11b could enhance Wnt3a-induced Topflash-luciferase in HEK293T cells in a dose-dependent manner (Fig. 4A), while having no effect on BMP, TGF-β, TNF-α, Notch and Hippo signaling (Appendix Fig. S3A–E). It also enhanced Wnt reporter activity in different colorectal cancer cells (Fig. 4B; Appendix Fig. S3F,G). In agreement with the Wnt activation by Bcl11b, most of the examined Wnt target genes were down-regulated in *Bcl11b*-deficeint crypts (Fig. 4C) and *Bcl11b* knockout

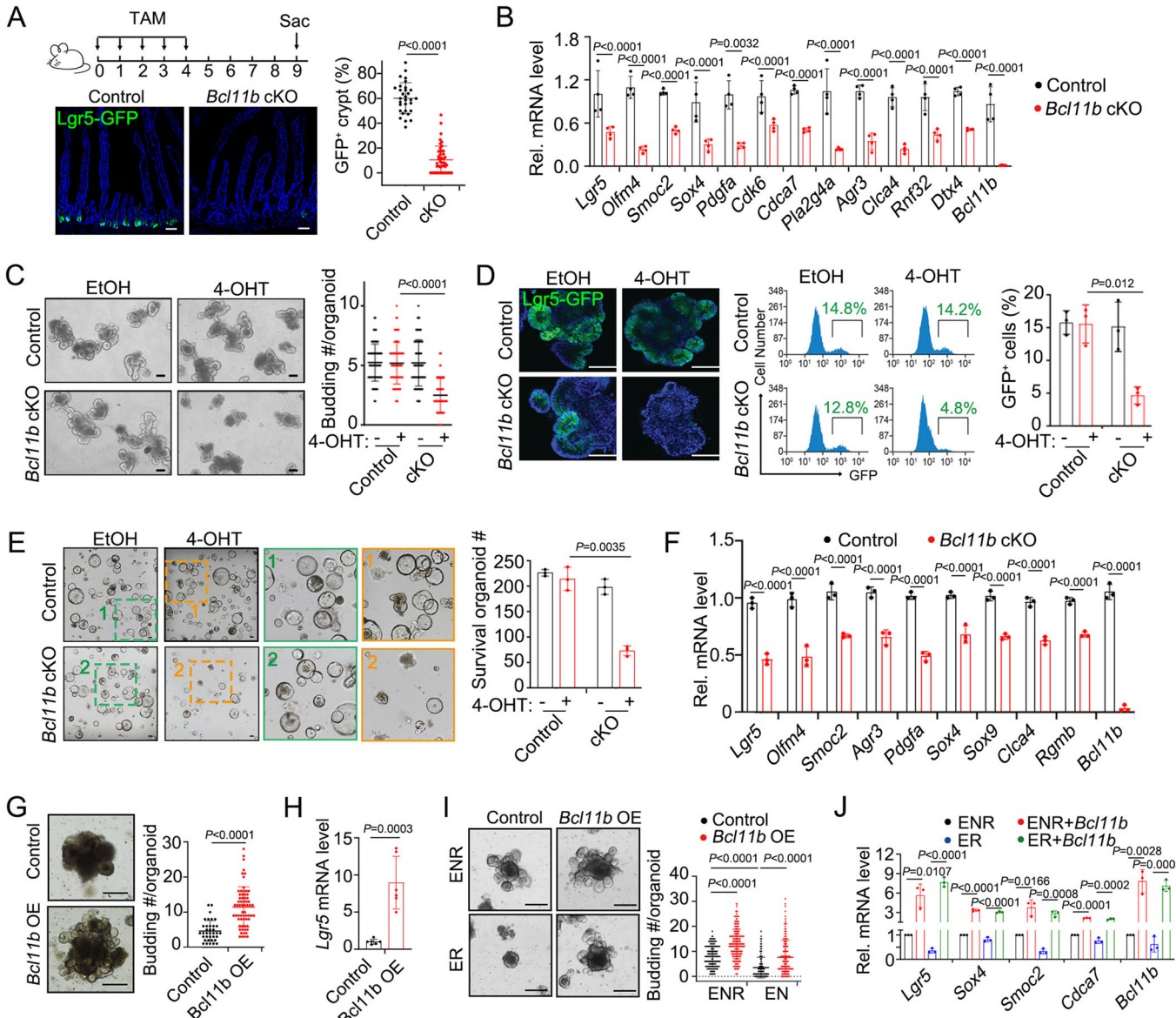

organoids (Appendix Fig. S3H). Additionally, Dox-induced *Bcl11b* overexpression in the intestinal organoids upregulated the expression of Wnt target genes, like *Axin2* and *Bmp3* (Appendix Fig. S3I).

Then we tested whether Bcl11b modulates stemness of Lgr5+ ISCs by activating Wnt signaling. When we cultured organoids with lower level of Rspo-1, the budding number of organoids decreased (Appendix Fig. S3J), indicating a lower stem cell activity. However, when Bcl11b was overexpressed, organoids cultured with low Rspo-1 were able to maintain the Lgr5+ ISCs activity (Appendix Fig. S3J). In line with the result in organoids, *Bcl11b* knockout attenuated the hyperproliferation and decreased crypt length induced by intravenously injection of adenovirus expressing Rspo-1 in vivo (Fig. 4D; Appendix Fig. S3K). These data collectively suggest that Bcl11b plays a role in ISC maintenance by activating Wnt signaling.

We then attempted to address how Bcl11b activates Wnt signaling. As Bcl11b is a nuclear protein (Senawong et al, 2003) and activated the Wnt reporter in *APC*-mutated colon cancer cells, we hypothesized that

it might work downstream of β-catenin. As expected, it further enhanced the reporter expression induced by the active β-catenin (SA-β-catenin) (Fig. 4E). We observed that ectopically overexpressed Bcl11b interacted with both endogenous β-catenin and TCF4 (Fig. 4F), and the Bcl11b-β-catenin interaction was confirmed by surface plasmon resonance (SPR) assay (Fig. 4G). However, Bcl11b did not interact with the Notch and Hippo signal downstream transcription factors HES1 or YAP (Appendix Fig. S4A), consistent with the reporter assay results (Appendix Fig. S3D,E). The interaction between ectopically overexpressed Bcl11b and β-catenin was enhanced by Wnt3a, although the Bcl11b-TCF4 interaction was unchanged (Appendix Fig. S4B,C). Furthermore, the β-catenin-TCF4 interaction was decreased in *Bcl11b* KO crypts (Fig. 4H). Overexpression of Bcl11b promoted the β-catenin-TCF4 interaction, which was further increased by Wnt3a (Appendix Fig. S4D). These results together indicate that Bcl11b specifically facilitates Wnt signaling by interacting with the β-catenin-TCF4 complex.

**Figure 2. Bcl11b is critical for Lgr5+ stem cell maintenance.**

(A) Representative images and quantification of Lgr5-GFP+ cells in the proximal jejunum of *Vil-CreERT2;Lgr5-GFP-IRES-creERT2* (control) and *Vil-CreERT2;Lgr5-GFP-IRES-creERT2;Bcl11b^{fl/fl}* (*Bcl11b* cKO) mice at 9d post tamoxifen injection (dpt). Nuclei were counter-stained with DAPI. Scale bar, 100 μm. n = 3 mice/group. At least 10 crypts per mice were counted. Each dot represents one crypt measurement. Line indicate mean ± SD. (B) *Vil-CreERT2;Lgr5-GFP-IRES-creERT2* (control) and *Vil-CreERT2;Lgr5-GFP-IRES-creERT2;Bcl11b^{fl/fl}* (*Bcl11b* cKO) mice were sacrificed 2 days after daily tamoxifen administration for 5 times, and crypts were isolated for analysis of stem cell signature gene expression by qRT-PCR. n = 4 mice/group. Each dot represents one mouse. (C) Organoids derived from the crypts of *Vil-CreERT2;Lgr5-GFP-IRES-creERT2* (control) and *Vil-CreERT2;Lgr5-GFP-IRES-creERT2;Bcl11b^{fl/fl}* (*Bcl11b* cKO) mice were treated with EtOH or 4-hydroxytamoxifen (4-OHT) for 48 h. Images were taken 2 days later (left), and the bud number of each organoid were counted from three independent experiments (right). Scale bar, 100 μm. Each dot represents one organoid. 75 organoids were measured from 3 independent experiments. (D) Organoids derived from the crypts of *Vil-CreERT2;Lgr5-GFP-IRES-creERT2* (control) and *Vil-CreERT2;Lgr5-GFP-IRES-creERT2;Bcl11b^{fl/fl}* (*Bcl11b* cKO) mice were treated with EtOH or 4-OHT for 48 h and then fixed for analysis of GFP expression 48 h later. Left panel shows the representative images, right panel shows the statistics result of GFP+ cells through FACS. Nuclei were counter-stained with DAPI. Scale bar, 100 μm. Each dot represents one experiment. n = three independent experiments. (E) The Lgr5+ ISCs of *Vil-CreERT2;Lgr5-GFP-IRES-creERT2* (control) and *Vil-CreERT2;Lgr5-GFP-IRES-creERT2;Bcl11b^{fl/fl}* (*Bcl11b* cKO) mice were sorted by FACS. About 5000 cells were seeded in each well for single cell culture upon EtOH or 4-OHT treatment. Six days later, the images were taken, and organoids were counted. Right panel shows the average organoids number of three independent experiments. Scale bar, 100 μm. Each dot represents one experiment. n = three independent experiments. (F) Organoids derived from the crypts of *Vil-CreERT2;Lgr5-GFP-IRES-creERT2* (control) and *Vil-CreERT2;Lgr5-GFP-IRES-creERT2;Bcl11b^{fl/fl}* (*Bcl11b* cKO) mice were treated with 4-OHT for 48 h before harvested for analyzing stemness-related gene expression with qRT-PCR. Each dot represents one experiment. n = 3 independent experiments. (G) Representative morphology images (left) and budding number (right) of organoids derived from the crypts of *R26-M2rtTA;Vil-CreERT2* mice at day 11 after 10 μM Doxycycline treatment, which were infected with Lentivirus to overexpress *Bcl11b*. Scale bar, 200 μm. Each dot represents one organoid. n = 36 organoids (Control) and 65 organoids (Bcl11b OE) were measured from 3 independent experiments. (H) *Lgr5* expression of organoids in (G). Each dot represents one experiment. n = 6 independent experiments. (I) Representative morphology images (left) and budding number (right) of organoids derived from the crypts of *R26-M2rtTA;Vil-CreERT2* mice. The organoids were infected with Bcl11b-expressing lentivirus and treated with 10 μM doxycycline for 10 days and deprived of Noggin for last 5 days. Scale bar, 200 μm. Each dot represents one organoid. At least 167 organoids were measured from 3 independent experiments. (J) Organoids were harvested for gene expression with qRT-PCR in (I). n = three independent experiments. Data represent mean ± SD (A–J). The data were analyzed by unpaired t test with Welch-correction (A, G, H), Two-way ANOVA with Tukey's multiple comparison test (B–F, I, J). The exact P value is displayed. Source data are available online for this figure.

As Bcl11b is a DNA binding protein (Li et al, 2010a), we wanted to identify its targets in ISCs by performing chromatin immuno-precipitation sequencing (ChIP-seq) (Dataset EV1). The canonical Bcl11b motif was successfully enriched (Fig. 4I). When comparing this ChIP-seq data with the targets of TCF4 and β-catenin (Schuijers et al, 2015b; Schuijers et al, 2014), many TCF4/β-catenin target genes were observed to be co-occupied by Bcl11b (Fig. 4I). Gene ontology (GO) analysis revealed that Bcl11b could regulate cell cycle, stem cell maintenance, cell proliferation and Wnt signaling (Fig. 4J). ChIP assay confirmed the binding of Bcl11b to a number of Wnt-target genes and ISCs signature genes (Fig. 4K; Appendix Fig. S4E), but not the Notch target genes (Appendix Fig. S4F). These data indicate that Bcl11b directly binds and likely activates Wnt target gene in the intestinal epithelium.

## Bcl11b maintains the chromatin accessibility by repressing the NuRD complex

To investigate how Bcl11b activates Wnt target gene expression, we conducted mass spectrometry analysis to identify Bcl11b-interacting proteins in cultured organoids. Biological processing analysis of these enriched proteins showed the candidates mainly participate in chromatin remodeling, regulation of cell fate specification, stem cell differentiation, as well as stem cell population maintenance (Appendix Fig. S5A; Dataset EV2). GO cellular component analysis of enriched Bcl11b-interacting proteins showed that the NuRD complex, a complex mediating epigenetic modification and chromatin remodeling, was enriched (Fig. 5A). The NuRD complex contains several proteins, including HDAC1, MBD2/3, RBBP4/7, GATAD2B and MTA2 (Leighton and Williams, 2020). The interaction of Bcl11b with NuRD complex components were confirmed by immunoprecipitation with ectopically overexpressed Bcl11b and SPR assay (Fig. 5B; Appendix Fig. S5B,C). These data indicate that Bcl11b may affect chromatin accessibility by modulating the function of the NuRD complex.

To address the role of Bcl11b in chromatin accessibility regulation, we performed assay for transposase accessible chromatin sequencing (ATAC-seq) and ChIP-seq for H3K27me3 and H3K27Ac in Lgr5+ ISCs of control or *Bcl11b* cKO mice. Interestingly, the ATAC-seq signal was reduced at promoter regions in *Bcl11b*-KO ISCs compared to control ISCs (Appendix Fig. S6A; Dataset EV3). Global functional enrichment analysis showed that the reduced ATAC signals were in Wnt signaling and cell cycle regulation (Appendix Fig. S6B). Notably, Bcl11b binding was abundant at regions where the ATAC-seq signal was lost in *Bcl11b*-cKO ISCs, specifically cluster 1 and cluster 2 (29.81% and 41.65% of which bind BCL11B, respectively) (Fig. 5C). GO analysis of genes in cluster 1 demonstrated that Wnt signaling pathway and stem cell population maintenance were enriched (Fig. 5D). Consistently, the ATAC-seq tracks showed lower signal at several Wnt target and ISCs signature gene loci in *Bcl11b* KO ISCs (Appendix Fig. S6C). Bcl11b binding sites were located in the genes with lower expression levels and lower ATAC-seq signal in *Bcl11b* KO ISCs (Fig. 5E; Appendix Fig. S6D), suggesting that Bcl11b might function as a transcriptional activator in ISCs. As H3K27Ac is a marker for active gene transcription (Kazakevych et al, 2017) and H3K27me3 is a marker for silent gene transcription (Saxena and Shivdasani, 2021), we did ChIP-seq for H3K27me3 and H3K27Ac in control and *Bcl11b* cKO ISCs. In line with the reduced expression and lower ATAC-seq signal, Bcl11b binding sites at Wnt target gene and ISC signature gene loci, including *Bmp3*, *Rnf32*, *Jun*, *Fzd7*, *Sox4*, *Cdca7*, showing decreased H3K27Ac modification or elevated H3K27me3 modification in KO versus control ISCs. Moreover, blocking the NuRD complex activity rescued impaired stemness induced by *Bcl11b*-deficiency, further confirming that Bcl11b regulates ISC maintenance through inhibiting the NuRD complex (Fig. 5F). Together, our data suggest that Bcl11b binds to and positively regulates Wnt target genes via inhibiting the NuRD complex-mediated epigenetic modification and chromatin remodeling.

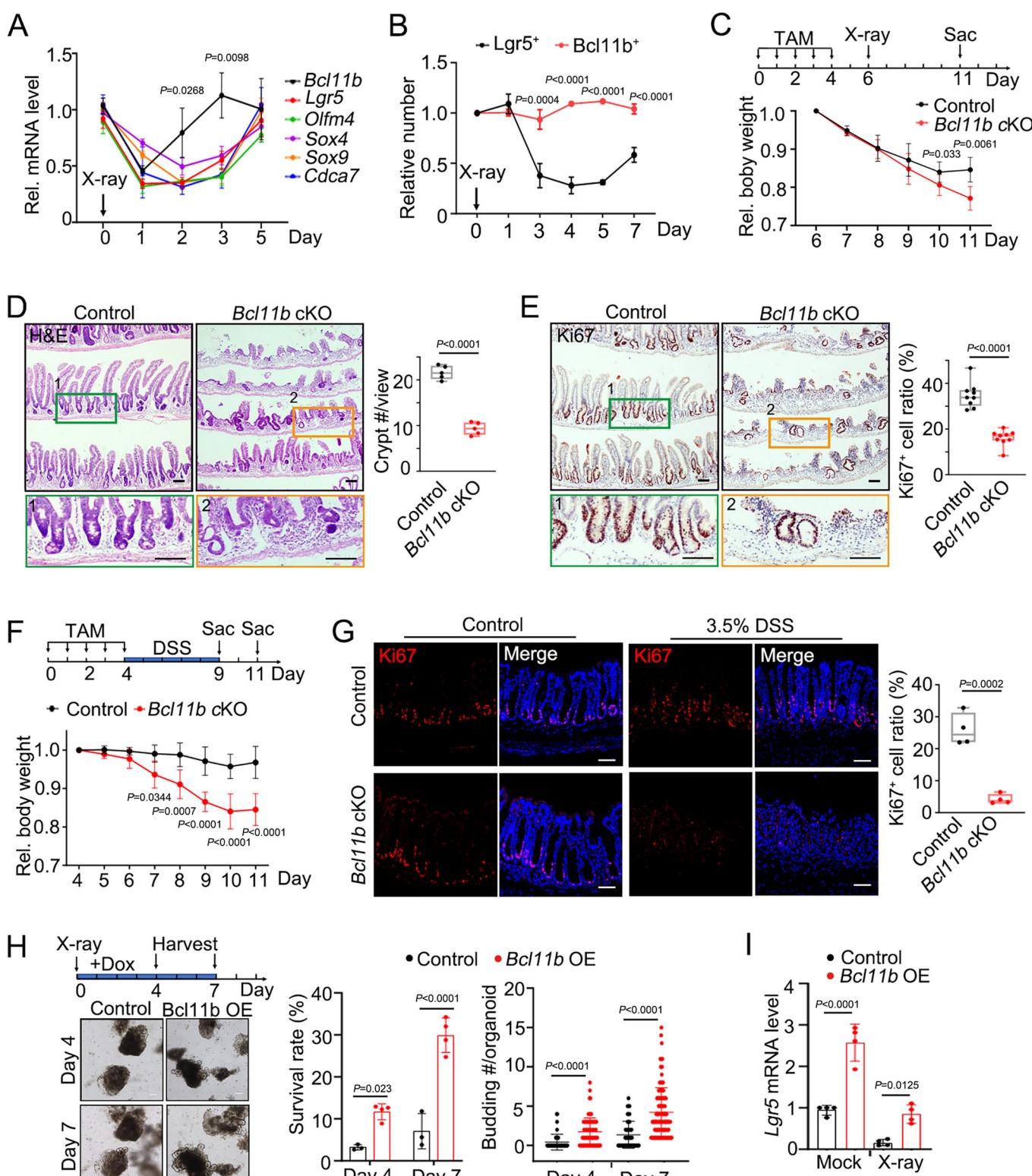

## Bcl11b promotes colorectal cancer formation

Since Lgr5[+] stem cells play important roles in colorectal cancer (CRC) development (Barker et al, 2009; Clevers, 2006; Nusse and Clevers, 2017; Shimokawa et al, 2017), we examined the role of

Bcl11b in this process. At the single cell transcription level, Bcl11b expression was higher in gut mucosal cancer samples from patients compared with normal tissues, especially in stem cell/TA cell clusters (Fig. 6A; Appendix Fig. S7A,B). Consistently, the two separate GSE datasets of endoscopically procured gut mucosal

**Figure 3. Bcl11b is required for intestinal epithelium regeneration after injury.**

(A) *Lgr5-GFP-IRES-creERT2* mice were exposed to 6 Gy X-ray radiation, and GFP$^+$ cells were isolated from crypts at different time after irradiation for gene expression. $n = 3$ mice/group. (B) *Lgr5-GFP-IRES-creERT2;Bcl11b-tomato* mice were exposed to 10 Gy X-ray radiation, and cells from crypts were analyzed at different time after irradiation through FACS. $n = 4$ mice/group. (C) Body weight change of *Vil-CreERT2;Lgr5-GFP-IRES-creERT2* (control) and *Vil-CreERT2;Lgr5-GFP-IRES-creERT2;Bcl11b$^{fl/fl}$* (*Bcl11b* cKO) mice after treated daily with tamoxifen for 5 times and exposed to 10 Gy X-ray radiation. $n = 5$ mice/group. (D, E) *Vil-CreERT2;Lgr5-GFP-IRES-creERT2* (control) and *Vil-CreERT2;Lgr5-GFP-IRES-creERT2;Bcl11b$^{fl/fl}$* (*Bcl11b* cKO) mice were treated with tamoxifen for 5 consecutive days, exposed to 10 Gy X-ray radiation 2 days later and then sacrificed 5 days later for analysis. Representative images of H&E (D) and Ki67 staining (E) were shown. Insert 1 and 2 were enlarged. Scale bars, 100 μm. $n = 5$ mice/group. Each dot represents one mouse (D). Each dot represents one field and at least 9 field were measured from 5 mice per group (E). The central line represents the median. The box spans from the 25th to the 75th percentile. Whiskers extend to the smallest and largest values. All points are presented. (F) Relative body weight change of *Vil-CreERT2;Lgr5-GFP-IRES-creERT2* (control) and *Vil-CreERT2;Lgr5-GFP-IRES-creERT2;Bcl11b$^{fl/fl}$* (*Bcl11b* cKO) mice after treated daily with tamoxifen for 5 times and fed with 3.5% DSS-containing water. $n = 5$ mice/group. (G) Immunofluorescence staining of Ki67 in *Vil-CreERT2;Lgr5-GFP-IRES-creERT2* (control) and *Vil-CreERT2;Lgr5-GFP-IRES-creERT2;Bcl11b$^{fl/fl}$* (*Bcl11b* cKO) mice with or without DSS treatment at day 11. The central line represents the median. The box spans from the 25th to the 75th percentile. Whiskers extend to the smallest and largest values. All points are presented. $n = 4$ mice/group. Scale bars, 100 μm. (H) Representative morphology images and quantification of organoids derived from the crypts of *R26-M2rtTA;Vil-CreERT2* mice at indicated time after treated with doxycycline and 8 Gy X-ray radiation. The organoids were infected with Bcl11b-expressing lentivirus. Scale bars, 100 μm. $n = 3$ independent experiments (Control) and 4 independent experiments (Bcl11b OE). Each dot represents one experiment (middle). Each dot represents one organoid (right) and at least 40 organoids were measured from independent experiments. (I) Stemness gene *Lgr5* expression of organoids in (H). $n =$ four independent experiments. Each dot represents one experiment. Data represent mean ± SD (A–C, F, H, I). The data were analyzed by unpaired t test with Welch-correction (D, E, G) and Two-way ANOVA with Tukey's multiple comparison test (A, B, C, F, H, I). The exact P value is displayed. Source data are available online for this figure.

biopsies from cancer patients showed Bcl11b expression was markedly increased in colon adenocarcinoma (COAD) and rectum adenocarcinoma (READ) compared with normal samples (Fig. 6B). In further support of these observations, Bcl11b signals were much higher in E-cadherin-marked intestinal epithelial cells of surgically resected full-thickness intestinal tissues from CRC patients than that in normal samples (Fig. 6C). Moreover, the nuclear localization of β-catenin (Appendix Fig. S7C) and the ratio of β-catenin$^+$ cells/BCL11B$^+$ cells were increased in CRC tissues (Fig. 6D). Furthermore, analysis of the TCGA dataset showed that Bcl11b was increased in *APC* mutated samples (Fig. 6E; Appendix Fig. S7D). Collectively, these data suggest that intestinal epithelial Bcl11b expression is correlated with colorectal cancer.

To functionally address the role of Bcl11b in tumor formation, we deleted *Bcl11b* in LoVo cells, a colon cancer cell line carrying *APC* mutation (Ilyas et al, 1997) (Appendix Fig. S7E). *Bcl11b* KO resulted in decreased cell proliferation (Fig. 6F), as well as reduced Topflash reporter activity (Appendix Fig. S7F). Moreover, *Bcl11b* KO attenuated the colony formation of LoVo cells, evidenced by fewer and smaller colony size (Fig. 6F; Appendix Fig. S7G). The xenograft transplantation assays showed *BCL11B* KO LoVo cells exhibited impaired tumor development, with a reduction of both tumor weight and tumor size, compared to parental cells (Fig. 6G; Appendix Fig. S7H). These data indicate that Bcl11b promotes colorectal cancer development through activating Wnt signaling.

## Discussion

Balance of BMP and Wnt signaling is crucial for Lgr5$^+$ ISCs maintenance and colon cancer development (Wang et al, 2022; Zhou et al, 2023). In this study, we report that a new Wnt activator Bcl11b plays a key role in Lgr5$^+$ ISCs maintenance and epithelium regeneration (Fig. 7). Mechanistically, Bcl11b activates Wnt signaling via facilitating the TCF4-β-catenin interaction and inhibiting NuRD complex function. As Bcl11b expression is suppressed by BMP signaling, our data revealed a new cross-talk between BMP and Wnt signaling in Lgr5$^+$ ISCs, in which BMP restrains Wnt signaling by inhibiting the Wnt activator *Bcl11b* expression.

We found that Bcl11b promotes Lgr5$^+$ ISCs by activating Wnt/β-catenin signaling. Bcl11b may have different functions in different tissues. It has been reported that Bcl11b is required for the early stage of T cell development and T cell identity maintenance by directly regulating target genes (Ikawa et al, 2010; Li et al, 2010a; Li et al, 2010b; Liu et al, 2010; Trujillo-Ochoa et al, 2023; Wakabayashi et al, 2003). Cai and colleagues reported that Bcl11b regulates the mammary epithelial stem cell quiescence, binds to the promoters of Wnt target genes and inhibits Wnt signaling in breast cancer (Bai et al, 2022; Cai et al, 2017). The mechanism underlying the tissue-specific functions of *Bcl11b* requires further investigation.

Bcl11b has been reported to repress intestinal epithelium regeneration and adenoma formation by inhibiting the Wnt/β-catenin pathway as the *Bcl11b$^{KO/+}$;Apc$^{min/+}$* mice developed more adenoma than the *Apc$^{min/+}$* mice (Sakamaki et al, 2015). The discrepancy between that study and ours may be owing to different genetic mice used: conventional *Bcl11b$^{+/-}$* mice were mainly utilized in Sakamaki et al's study, while we employed inducible intestinal epithelium-specific knockout mice. As Bcl11b is critical for T cell development, the observed effect of Bcl11b heterozygosity on tumor formation may possibly attribute to T cell deficiency. These authors reported that some of the Wnt target genes, including *Myc*, *Ccnd1* and *Jun*, were elevated in Lgr5$^+$ cells of *Lgr5-CreERT2;Bcl11b$^{fl/fl}$* mice two months after 4-OHT induced *Bcl11b* knockout. However, more Wnt target genes including *Axin2*, *Sp5*, *Lef1*, *Mmp7*, *Tnfrsf19* were decreased after *Bcl11b* knockout, which is consistent with our data. The precise function of *Bcl11b* in a specific tissue needs careful examination with proper models.

The crucial role of *Bcl11b* in sustaining Lgr5$^+$ ISCs is corroborated by the significant reduction of Lgr5$^+$ ISCs upon *Bcl11b* deletion. Surprisingly, *Bcl11b* deletion does not alter the overall architecture of the intestinal epithelium, nor does it affect the proliferation of crypt cells or the differentiation of Paneth and goblet cells (Appendix Fig. S1D). Consistent with our observation, impaired stem cells were not accompanied by reduced Ki67$^+$ cells (Baghdadi et al, 2022; Castillo-Azofeifa et al, 2023; Koppens et al, 2021; Li et al, 2020a; Sato et al, 2020). Indeed, depletion of Lgr5$^+$ ISCs can be compensated by the expansion of other stem cell

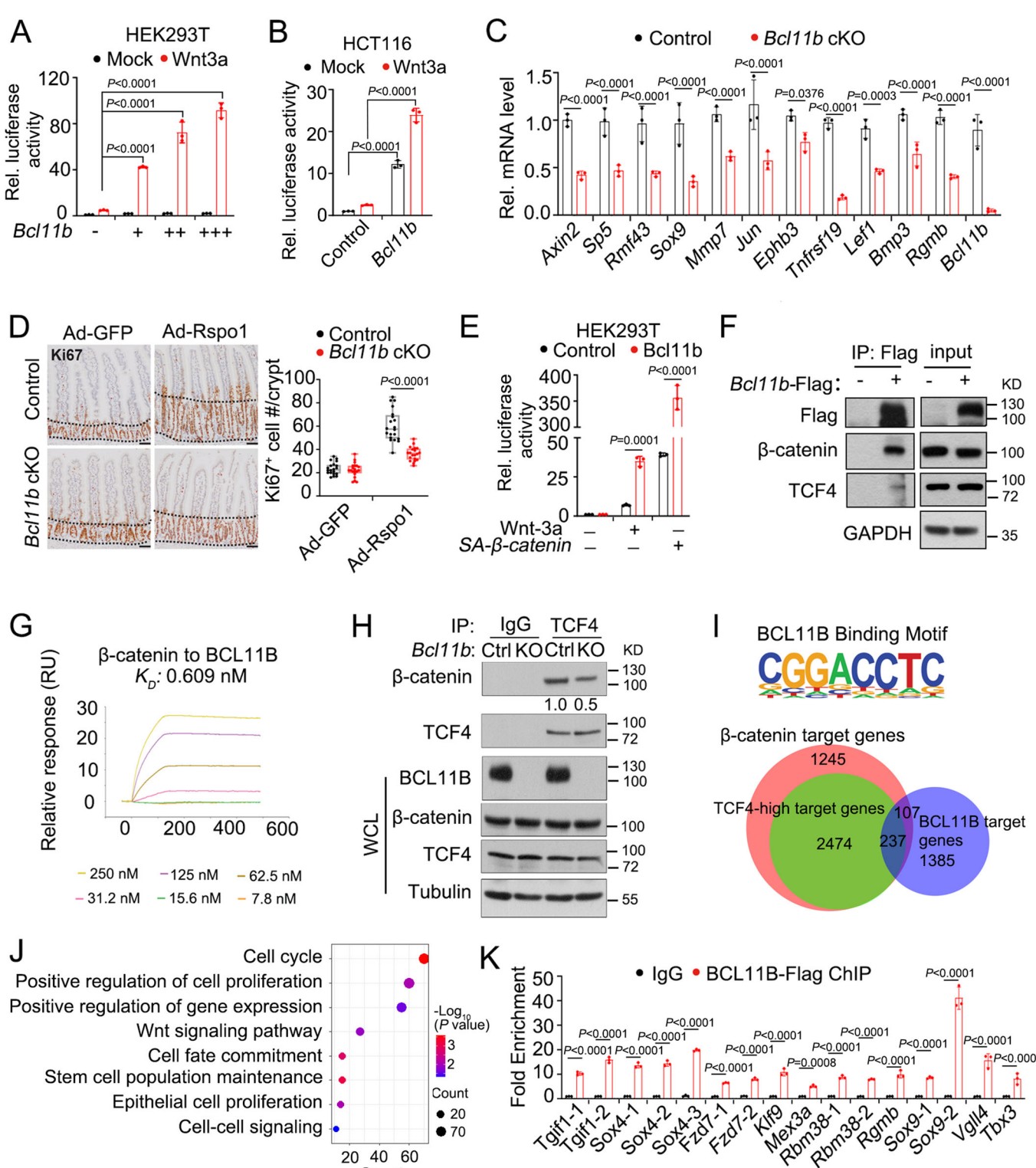

populations (Sangiorgi and Capecchi, 2008; Takeda et al, 2011; Tetteh et al, 2016; Tian et al, 2011; van Es et al, 2012). Although depletion of Lgr5+ ISCs does not disrupt intestinal homeostasis, Lgr5+ ISCs are indispensable for intestinal epithelium regeneration following injury (Barker et al, 2007; Clevers, 2013; Metcalfe et al,

2014; van der Flier and Clevers, 2009). Consistent with this, our data showed that Bcl11b is essential for intestinal epithelium regeneration after X-ray and DSS-induced injuries.

The interaction of Bcl11b with the NuRD complex in the intestinal epithelial cells is consistent with the recent study that

◀ **Figure 4. Bcl11b facilitates Wnt/β-catenin signaling.**

(A, B) HEK293T cells (A) or HCT116 cells (B) were transfected with different amounts of *Bcl11b* plasmid (+: 25, ++: 50 and +++: 100 ng/well in (A), 100 ng/well in (B)) together with Topflash-luciferase or Fopflash-Luciferase. The cells were treated with 20 ng/mL Wnt-3a for 18 h and then harvested for luciferase measurement. The ratio of Topflash-luciferase to Fopflash-Luciferase for each treatment was shown. $n$ = three independent experiments. (C) *Vil-CreERT2;Lgr5-GFP-IRES-creERT2* (control) and *Vil-CreERT2;Lgr5-GFP-IRES-creERT2;Bcl11b^fl/fl^* (*Bcl11b* cKO) mice were sacrificed 2 days after daily tamoxifen administration for 5 times, and crypts were isolated for analysis of stem cell signature gene expression by qRT-PCR. $n$ = 3 mice/group. (D) *Vil-CreERT2;Lgr5-GFP-IRES-creERT2* (control) and *Vil-CreERT2;Lgr5-GFP-IRES-creERT2;Bcl11b^fl/fl^* (*Bcl11b* cKO) mice were injected adenovirus expressing GFP or R-spondin 1 (Rspo-1) 1 day later after 5 times daily tamoxifen administration. The mice were sacrificed 5 days later for Ki67 staining of intestinal epithelium. The dashed lines mark the crypt region of intestinal epithelium. Scale bar, 100 μm. $n$ = 3 mice/group. Each dot represents one crypt measurement. The graph depicts 20 crypts measured on tissue sections from 3 mice. The central line represents the median. The box spans from the 25th to the 75th percentile. Whiskers extend to the smallest and largest values. All points are presented. (E) HEK293T cells were transfected with indicated plasmids together with Topflash-luciferase or Fopflash-Luciferase, and reporter assay was performed. $n$ = three independent experiments. (F) Organoids derived from the crypts of *R26-M2rtTA;Vil-CreERT2* mice were harvested for immunoprecipitation with anti-Flag antibody at day 3 after doxycycline treatment, which were infected with Bcl11b-expressing lentivirus. $n$ = three independent experiments. (G) Representative SPR binding profile of purified β-catenin to BCL11B protein. $n$ = three independent experiments. (H) *Vil-CreERT2;Lgr5-GFP-IRES-creERT2* (Ctr1) and *Vil-CreERT2;Lgr5-GFP-IRES-creERT2;Bcl11b^fl/fl^* (KO) mice were sacrificed 1 day after 5 times daily tamoxifen administration. Crypts were isolated for immunoprecipitation and immunoblotting with anti-TCF4 antibody or control IgG. Image J was used for quantitation of bands. $n$ = three independent experiments. (I) HOMER motif of BCL11B-Flag ChIP-seq. Venn diagram displaying the overlap of BCL11B, β-catenin, and TCF4 target genes. (J) Functional enrichment analysis of *Bcl11b* target genes. Differently expressed genes with $P < 0.05$ (Wald test) and absolute value of |log2FC| ≥ 1 were used for Gene ontology (GO) analysis. GO terms with $P < 0.05$ (Fisher's Exact test) were determined to be statistically significant. (K) ChIP-qPCR analysis of Bcl11b-Flag binding at regions near indicated gene promoters in intestinal organoids. Fold change is shown relative to normal-IgG control. Data represent mean ± SD (A–E, K). $n$ = 3 independent experiments. The data were analyzed by Two-way ANOVA with Tukey's multiple comparison test (A–E, K). The exact P value is displayed. Source data are available online for this figure.

Bcl11b and NuRD complex regulate T-cell development (Liao et al, 2023). Our data indicate that Bcl11b inhibits the NuRD complex, leading to increased chromatin accessibility and gene expression due to reduced H3K27me3 and elevated H3K27Ac. The HDAC1 inhibitors TSA and VPA were able to restore the phenotype caused by *Bcl11b* deficiency. Consistent with this, the NuRD component *Mbd3* deletion has been shown to induce increased expression of *Lgr5* (Aguilera et al, 2011). Although Bcl11b disrupts NuRD function is unclear and needs further investigation, the Bcl11b homolog Bcl11A has been shown to directly interact with RBBP4 (Moody et al, 2018). *BCL11A* and *BCL11B* share 67% and 55% identity at the amino acid level, especially the N-terminal peptide (residues 2–16) of BCL11B and BCL11A are very similar (Satterwhite et al, 2001), strongly suggesting that BCL11B may also interact with RBBP4.

The low BMP and high Wnt signaling activities at the bottom of crypts is critical for Lgr5⁺ ISC maintenance, and different mechanisms may be involved in the balance of these two pathways. We have reported that BMP signaling could counteract Wnt signaling to suppress the expression of the stem cell signature genes in Lgr5⁺ ISCs (Qi et al, 2017). Therefore, BMP suppresses the Wnt activator Bcl11b expression, which provides another way to coordinate BMP and Wnt signaling in regulation of Lgr5⁺ ISCs. Understanding of the cross-talk between different niche factor-mediated signaling pathways in ISCs will help us to find therapeutic strategies for treatment of related diseases.

# Methods

### Reagents and tools table

| Reagent/Resource | Reference or Source | Identifier or Catalog Number/ RRID |
|---|---|---|
| **Experimental Models** | | |
| HEK293 cells (*H. sapiens*) | ATCC | CRL-1573 |
| HEK293T cells (*H. sapiens*) | ATCC | CRL-3216 |

| Reagent/Resource | Reference or Source | Identifier or Catalog Number/ RRID |
|---|---|---|
| NMuMG cells (*H. sapiens*) | ATCC | CRL-1636 |
| LoVo cells (*H. sapiens*) | ATCC | CCL-229 |
| NIH3T3 cells (*H. sapiens*) | ATCC | CRL-16586 |
| HEK293FT cells (*H. sapiens*) | Thermo Fisher Scientific | R70007 |
| HCT116 cells (*H. sapiens*) | ATCC | CCL-247 |
| Caco-2 cells (*H. sapiens*) | ATCC | HTB-37 |
| *Lgr5-GFP-IRES-CreERT2* mice (*M. musculus*) | Jackson Laboratory | 008875 |
| *Bcl11B^fl/fl^* mice (*M. musculus*) | Li et al, 2010b | N/A |
| *Bcl11b-tomato* mice (*M. musculus*) | Li et al, 2010b | N/A |
| *Vil-CreERT2* mice (*M. musculus*) | Ireland et al, 2004 | N/A |
| *R26-M2rtTA* mice (*M. musculus*) | Jackson Laboratory | 006965 |
| **Recombinant DNA** | | |
| pLVX-TRE3G | Clontech | 631193 |
| pLVX-TRE3G-bsd-Bcl11b-Flag | This paper | N/A |
| pETMBP.3C-*Rbbp4* | This paper | N/A |
| pETMBP.3C-*Bcl11b* | This paper | N/A |
| pETMBP.3C-*Mbd3* | This paper | N/A |
| pETMBP.3C-*β-catenin* | This paper | N/A |
| pETMBP.3C-*Hdac1* | This paper | N/A |
| px458 | Addgene | 48138 |
| px459 | Addgene | 49139 |
| px458-sg*Bcl11b*-#1 | This paper | N/A |
| px459-sg*Bcl11b*-#2 | This paper | N/A |
| px459-sg*Bcl11b*-#3 | This paper | N/A |
| pGL3-BRE-luciferase | This paper | N/A |
| pGL3-CAGA-luciferase | This paper | N/A |
| pGL3-NF-κB-luciferase | This paper | N/A |

| Reagent/Resource | Reference or Source | Identifier or Catalog Number/ RRID |
|---|---|---|
| Topflash-luciferase | This paper | N/A |
| Fopflash-luciferase | This paper | N/A |
| pGa981-Rbpj-promoter-luciferase | This paper | N/A |
| pGL3-CYR61-promoter-luciferase | Zhang et al, 2024 | N/A |
| **Antibodies** | | |
| Rabbit anti-KI67 | Abcam | ab15580; AB_443209 |
| Mouse anti-E-cadherin | BD Biosciences | 610182; AB_397581 |
| Mouse anti-β-catenin | Sigma | C7207; AB_476865 |
| Rabbit anti-Bcl11b | Proteintech | 55414-1-AP; AB_11182609 |
| Rabbit anti-lysozyme | Dako | F0372 |
| Goat anti-Rabbit IgG, HRP | Invitrogen | 31460; AB_228341 |
| Rabbit anti-HA | Santa Cruz | sc-7392; AB_627809 |
| Mouse anti-Myc | Cell Signaling Technology | 2276; AB_331783 |
| Mouse anti-β-catenin | Santa Cruz | sc-7963; AB_626807 |
| Mouse anti-tubulin | Proteintech | 66031-1-Ig; AB_11042766 |
| Rabbit anti-Tcf4 | Cell Signaling Technology | 2569; AB_2199816 |
| Rabbit anti-Rbbp4 | Santa Cruz | sc-373873; AB_10918458 |
| Rabbit anti-HDAC1 | Cell Signaling Technology | 34589; AB_2756821 |
| Rabbit anti-Mbd3 | Santa Cruz | sc-166319; AB_2139753 |
| Goat anti-Lamin B | Santa Cruz | sc-6217; AB_648158 |
| Mouse anti-Flag | Sigma | F3165; AB_259529 |
| Mouse anti-GAPDH | ZSGB-Bio | TA-08; AB_2747414 |
| Mouse IgG polyclonal antibody | Sigma | 12-371; AB_145840 |
| Rabbit anti-Hes1 | Cell Signaling Technology | 11988S; AB_2728766 |
| Mouse anti-YAP | Santa Cruz | sc-101199; AB_1131430 |
| Mouse anti-FLAG | Sigma | F1804; AB_262044 |
| Donkey anti-mouse IgG | Invitrogen | A-31571; AB_162542 |
| Mouse anti-H3K27me3 | Abcam | ab6002; AB_305237 |
| Rabbit anti-H3K27Ac | Cell Signaling Technology | 8173P; AB_10949503 |
| Rabbit anti-Smad1 | Cell Signaling Technology | 6944; AB_10858882 |
| Rabbit anti-Smad4 | Purified by our lab (Fei et al, 2010) | N/A |

| Reagent/Resource | Reference or Source | Identifier or Catalog Number/ RRID |
|---|---|---|
| **Oligonucleotides and other sequence-based reagents** | | |
| PCR primers | This study | Appendix Table S1 |
| Bcl11b-gRNA#1: 5′-GCACGGCTTCCGCATCTACC-3′ | This study | N/A |
| Bcl11b-gRNA#2: 5′-GACGCACGGGCAGATCGGCA-3′ | This study | N/A |
| Bcl11b-gRNA#3: 5′-GTGCTCGGACGACGTGGCGA-3′ | This study | N/A |
| **Chemicals, Enzymes and other reagents** | | |
| DMEM | Thermo Fisher Scientific | 11965092 |
| McCoy's 5A (Modified) medium | Thermo Fisher Scientific | 16600082 |
| MEM | Thermo Fisher Scientific | 11095080 |
| Advanced DMEM/F12 | Thermo Fisher Scientific | 12634028 |
| Fetal bovine serum | Hyclone | SH20074.03 |
| Penicillin/Streptomycin | Thermo Fisher Scientific | 15140122 |
| GlutaMAX | Thermo Fisher Scientific | 35050061 |
| N2 | Thermo Fisher Scientific | 17502048 |
| B27 | Thermo Fisher Scientific | 17504044 |
| N-acetylcysteine | Sigma | A9165-25G |
| TrypLE | Gibco | 12604021 |
| 4-OH tamoxifen | Sigma | H7904 |
| Polyetherimide | Polysciences | PT-101-01N |
| EDTA | Beyotime | ST066 |
| Matrigel | BD Biosciences | 356234 |
| G418 | Gibco | 11811 |
| EGF | Novoprotein | C029 |
| Noggin | OrganRagen | 807-NOG |
| R-spondin1 | OrganRagen | 861-RS1 |
| CHIR-99021 | Selleck | S2924 |
| Blebbistatin | Selleck | S7099 |
| Puromycin | Gibco | A1113803 |
| Blasticidin S | Selleck | S7419 |
| Doxycycline | Selleck | S4163 |
| Protease inhibitor cocktail | Roche | 4693124001 |
| PMSF | Beyotime | ST506 |
| Polybrene | macgene | MC032 |
| Lenti-Concentin Virus Precipitation Solution | ExCell Biology | EMB810A-1S |
| Hematoxylin and eosin | Beyotime | C0105 |

| Reagent/Resource | Reference or Source | Identifier or Catalog Number/ RRID |
|---|---|---|
| Alcian blue | Baso | BA4087B |
| 2X KAPA HIFI HotStart polymerase | KAPA | KK2601 |
| Dynabeads Protein A/G immunoprecipitation Kit | Invitrogen | 10007D |
| NovoNGS CUT&Tag 3.0 High-Sensitivity Kit | Novoprotein | H259-YH01 |
| TruePrep DNA library Prep Kit V2 for illumine | Vazyme | TD202/TD502 |
| RNeasy Mini Kit | QIAGEN | 74104 |
| **Software** | | |
| Image J | https://imagej.net/ij/ | |
| GraphPad Prism10 | https://www.graphpad-prism.cn/ | |
| CytExpert | https://www.beckman.com/flow-cytometry/research-flow-cytometers/cytoflex/software | |
| IGV (2.6.2) | https://igv.org/ | |
| MACS2 (v2.2.5) | https://pypi.org/project/MACS2/ | |
| Biorender | https://www.biorender.com/ | |
| DAVID | http://david.adcc.ncifcrf.gov/ | |
| **Other** | | |
| X-ray irradiator | Radsource | RS2000 |
| Olympus FV3000 confocal microscope | Olympus | FV3000 |
| High Content Screening System Opera Phenix | PerkinElmer | Opera Phenix |
| CytoFlex LX | Beckman | CytoFlex LX |
| MoFlo XDP | Beckman | MoFlo XDP |
| LightCycler 480 | Roche | LightCycler 480 |
| Biacore 8K plus | Cytiva | Biacore 8K plus |

## Methods and protocols

### Mice

*Lgr5-GFP-IRES-CreERT2* mice were obtained from Jackson Laboratory (008875). *Bcl11B^{fl/fl}* and *Bcl11b-tomato* mice were kindly provided by Dr. Pengtao Liu, Wellcome Trust Sanger Institute (Li et al, 2010b), and *Vil-CreERT2* mice were a gift from Dr. Sylvie Robine, Institut Curie-CNRS (Ireland et al, 2004). All mice were back-crossed into the C57BL/6 genetic background for at least 10 generations. Both male and female mice ranging from 2 to 4 months old in age were used. No statistical method was used to predetermine sample size. Generally, we used at least three mice per genotype in each experiment. None of the animals was excluded from experiment. For Cre induction, mice were intraperitoneally injected with 100 μL tamoxifen in sunflower oil at 20 mg/mL for 5 consecutive days. For irradiation, mice received a single dose of abdominal X-ray radiation (6 or 10 Gy) with X-ray irradiator (Radsource, RS2000) and were then analyzed at different time

points in Laboratory Animal Resources Center of Tsinghua University. For DSS treatment, mice were given 3.5% DSS (molecular weight 36,000–50,000 Da; MP Biomedicals) in drinking water over a period of 5 days and then provided with normal drinking water. For adenovirus injection, the full length human Rspo-1 was cloned to generate adenovirus, and $5 \times 10^8$ pfu virus was injected to each mouse through tail vein injection. Procedures were approved by the Institutional Animal Care and Use Committee of Tsinghua University and were in compliance with all relevant ethical regulations (19-CYG1.G23-1). All specific-pathogen-free mice used in experiments were socially housed under a 12 h light-dark cycle with free access to food and water, with 23–25 °C and 50–56% humidity in the laboratory animal center of Tsinghua University.

### Cell lines

HEK293 (ATCC, CRL-1573), HEK293T (ATCC, CRL-3216), NMuMG (ATCC, CRL-1636), LoVo (ATCC, CCL-229), NIH3T3 (ATCC, CRL-16586), HEK293FT (Thermo Fisher Scientific, R70007) cells were cultured at 37 °C with 5% $CO_2$ in Dulbecco's modified Eagle medium (DMEM) (Thermo Fisher Scientific, 11965092) supplemented with 10% fetal bovine serum (FBS, Hyclone, SH20074.03) and 1% penicillin and streptomycin. HEK293FT cells were cultured under the selection with G418 (500 μg/mL, Gibco, 11811). HCT116 (ATCC, CCL-247) cells were cultured at 37 °C with 5% $CO_2$ in McCoy's 5A (Modified) medium (Thermo Fisher Scientific, 16600082) supplemented with 10% fetal bovine serum and 1% penicillin and streptomycin. Caco-2 (ATCC, HTB-37) cells were cultured at 37 °C with 5% $CO_2$ in Minimum Essential Medium (MEM) (Thermo Fisher Scientific, 11095080) supplemented with 20% fetal bovine serum and 1% penicillin and streptomycin.

### Isolation of intestinal crypts and organoid culture

Intestinal crypts were isolated and cultured as previously described (Zhao et al, 2015). Briefly, mouse intestine was cut longitudinally and washed three times with cold PBS. Villi were carefully scraped away, and small pieces (5–10 mm) of intestine were incubated in 2 mM EDTA in PBS for 30 min on ice. These pieces were then vigorously suspended in cold PBS, and the mixture was passed through 70 μm cell strainer (BD Biosciences). The crypt fraction was enriched through centrifugation (3 min at 300–400 × g). Then the crypts were embedded in Matrigel (BD Biosciences) and seeded on 48- or 24-well plates. After polymerization, crypt culture ENR medium (Advanced DMEM/F12 containing EGF (50 ng/mL, Novoprotein, C029), Noggin (100 ng/mL, OrganRagen, 807-NOG) and R-spondin1 (500 ng/mL, OrganRagen, 861-RS1), Penicillin/Streptomycin, GlutaMAX-I, N2, B27 and N-acetylcysteine (Sigma-Aldrich)) was added and refreshed every 2 days. For passaging, the organoids embedded in Matrigel were suspended in 1 ml cold PBS and pelleted by centrifugation (3 min at 300–400 × g). The pelleted organoids were embedded in fresh Matrigel and seeded on a plate followed by addition of culture medium. To induce *Bcl11b* knockout in organoids, 500 nM 4-OH-tamoxifen (4-OHT) was added to the culture medium. For the GFP expression analysis and Olfm4 in situ hybridization, organoids were harvested 2 days after 4-OHT removal. For single cell culture, Lgr5-GFP⁺ cells sorted from FACS was cultured in the expansion

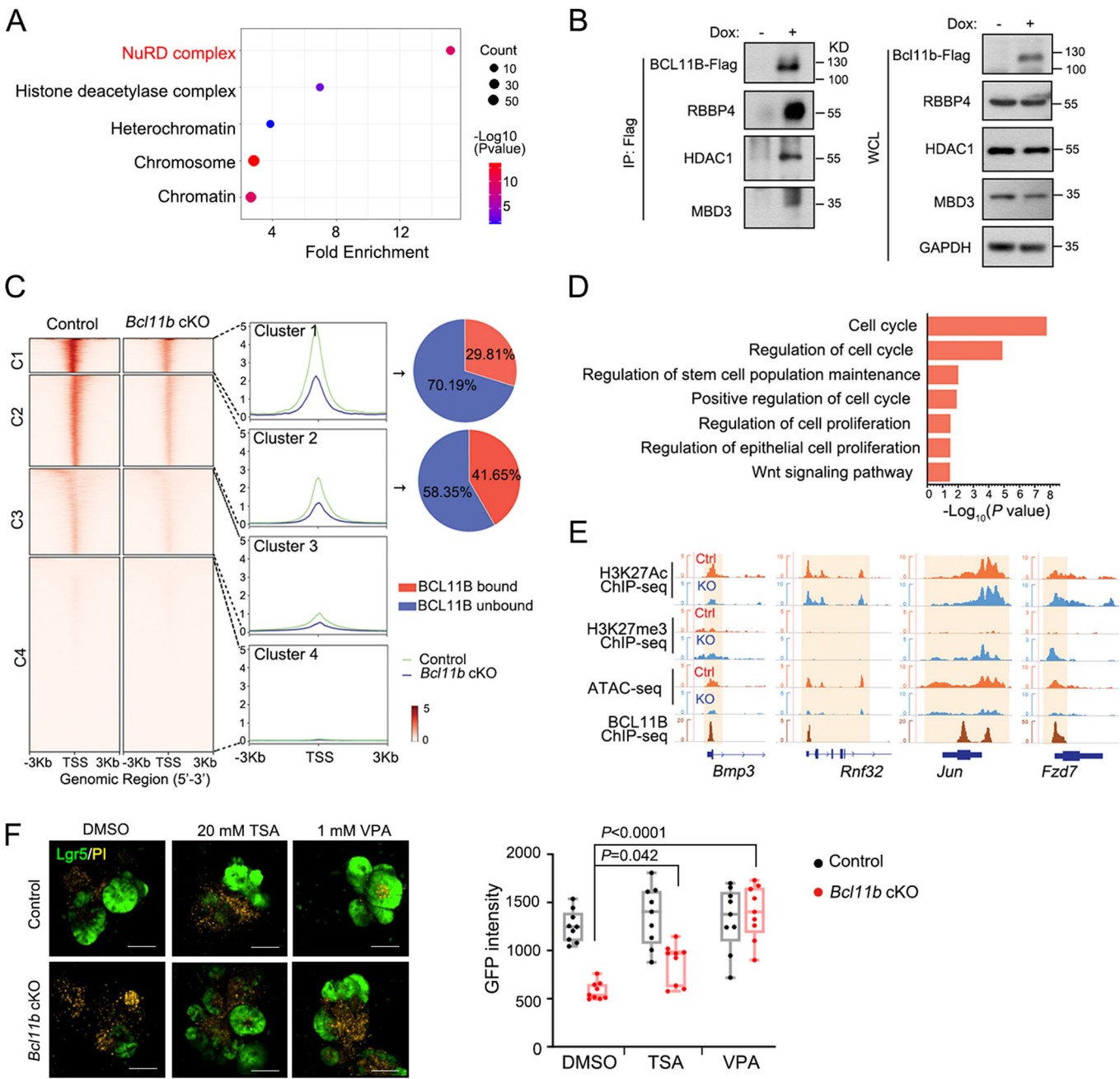

Figure 5. Bcl11b activates Wnt-target gene expression by modulating NuRD complex activity.

(A) Gene ontology analysis of top Bcl11b-interacting proteins in cellular component by DAVID. Proteins that are significantly enriched in the IP group with log2FoldChange > 1.95 and P.adjusted <0.00025 (empirical Bayes adjustment for t-test) were subjected to Gene Ontology enrichment analysis. GO terms with $P < 0.05$ (Fisher's Exact test) were determined to be statistically significant. (B) Organoids derived from the crypts of *R26-M2rtTA;Vil-CreERT2* mice were infected with Bcl11b-expressing lentivirus, treated with doxycycline for 3 days and then harvested for immunoprecipitation with anti-Flag antibody and immunoblotting with indicated antibodies. (C) ATAC-seq profiles of Lgr5+ ISCs in *Vil-CreERT2;Lgr5-GFP-IRES-creERT2* (control) and *Vil-CreERT2;Lgr5-GFP-IRES-creERT2;Bcl11b^{fl/fl}* (*Bcl11b* cKO) mice were divided into four clusters according to their chromatin dynamics. The relative strength of ATAC signals upon *Bcl11b* loss in each cluster is shown in the middle. (D) Functional enrichment analysis of the genes whose promoter showed lower ATAC signals from the Cluster 1 in (C). GO terms with $P < 0.05$ (Fisher's Exact test) were determined to be statistically significant. (E) Integrative Genomics Viewer (IGV) tracks display ChIP-seq reads and ATAC-seq reads along the indicated genes. Blue reads are from KO Lgr5+ ISCs and red reads from control. The y axis represents the CPM (count per million) of genes. The yellow boxes of the tracks depict the binding sites. ChIP-seq and ATAC-seq tracks show Bcl11b bind to indicated genes. $n = 3$ mice per group. (F) Organoids derived from the crypts of *Vil-CreERT2;Lgr5-GFP-IRES-creERT2* (control) and *Vil-CreERT2;Lgr5-GFP-IRES-creERT2;Bcl11b^{fl/fl}* (*Bcl11b* cKO) mice were treated with 4-OHT for 48 h and then with 20 nM TSA or 1 mM VPA for another 48 h before analysis of GFP expression. The left panel shows the representative images (green represent Lgr5+ cells, yellow represent PI signal), and the right panel shows the statistics of GFP+ cells intensity. Scale bar, 100 μm. $n = 3$ independent experiments. Each dot represents one image measurement. At least 9 images were measured from 3 independent experiments. The central line represents the median. The box spans from the 25th to the 75th percentile. Whiskers extend to the smallest and largest values. All points are presented. The data were analyzed by Two-way ANOVA with Tukey's multiple comparison test (F). The exact P value is displayed. Source data are available online for this figure.

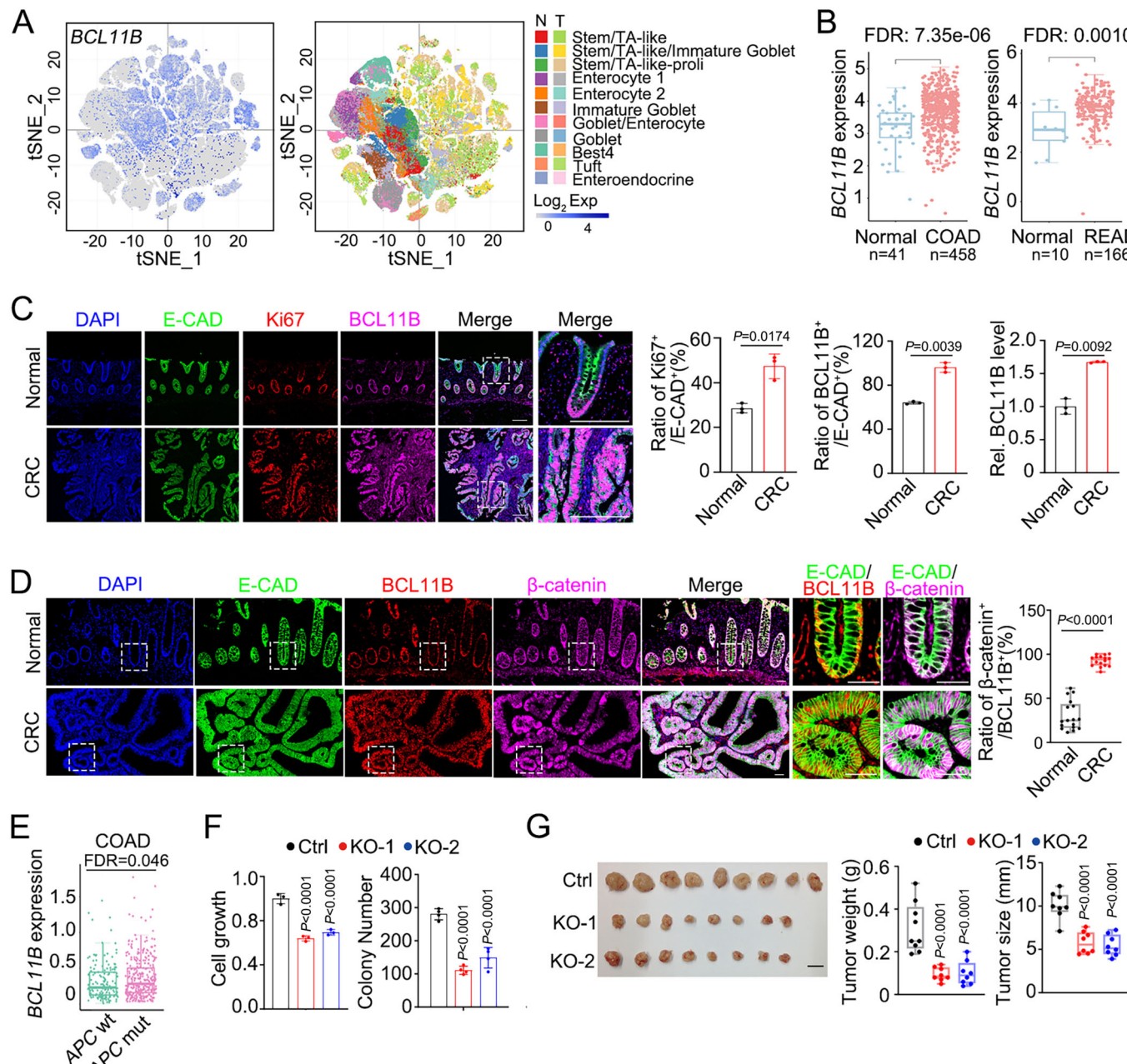

**Figure 6.  Bcl11b promotes colorectal cancer formation.**

(**A**) t-SNE plot with clustering results and *Bcl11b* expression in epithelial cells from health control (Normal, N) and colorectal cancer (Tumor, T) patients based on scRNA-seq data (GSE178341). (**B**) *Bcl11b* expression in COAD, READ tissues and normal tissues revealed by analysis of TCGA dataset. The central line represents the median. The box spans from the 25th to the 75th percentile (IQR). Whiskers extend to the smallest and largest values within 1.5 times the IQR. Points beyond are outliers. (**C**) Immunofluorescence staining (left) and quantification (right) of E-cadherin, Ki67, BCL11B in normal and CRC samples. $n = 3$ tissue/group. Scale bar, 200 μm. One dot represents one tissue sample. (**D**) Immunofluorescence staining (left) and quantification (right) of E-cadherin, BCL11B and β-catenin in normal and CRC samples. $n =$ three tissues/group. Each dot represents one crypt. At least 16 crypts on tissue sections from 3 different tissues. The central line represents the median. The box spans from the 25th to the 75th percentile. Whiskers extend to the smallest and largest values. All points are presented. (**E**) Analysis of TCGA dataset revealed the expression of *Bcl11b* in COAD tissues with *APC* mutation ($n = 324$) or wild-type *APC* ($n = 157$). The central line represents the median. The box spans from the 25th to the 75th percentile (IQR). Whiskers extend to the smallest and largest values within 1.5 times the IQR. Points beyond are outliers. (**F**) Control and *Bcl11b* knockout LoVo cells were seeded with 2000 cells per well, and cell growth was measured at day 4 (left panel). About 2000 control and *Bcl11b* knockout LoVo cells were subjected to colony formation assay, and the colony numbers were counted 2 weeks later (right panel). $n =$ three independent experiments (left) and four independent experiments (right). (**G**) Control and *Bcl11b* knockout LoVo cells ($10^7$ cells) were subcutaneously injected into the nude mice. The mice were sacrificed 15 days later to examine the tumor size. The central line represents the median. The box spans from the 25th to the 75th percentile. Whiskers extend to the smallest and largest values. All points are presented. Scale bar, 1 cm. $n = 9$ mice/group. Each dot represents one tumor. Data represent mean ± SD (**B**–**G**). The data were analyzed by unpaired t test with Welch-correction (**C**, **D**) and One-way ANOVA with Tukey's multiple comparison test (**F**, **G**). *$P < 0.05$, **$P < 0.01$ and ***$P < 0.001$. The exact $P$ value is displayed. Source data are available online for this figure.

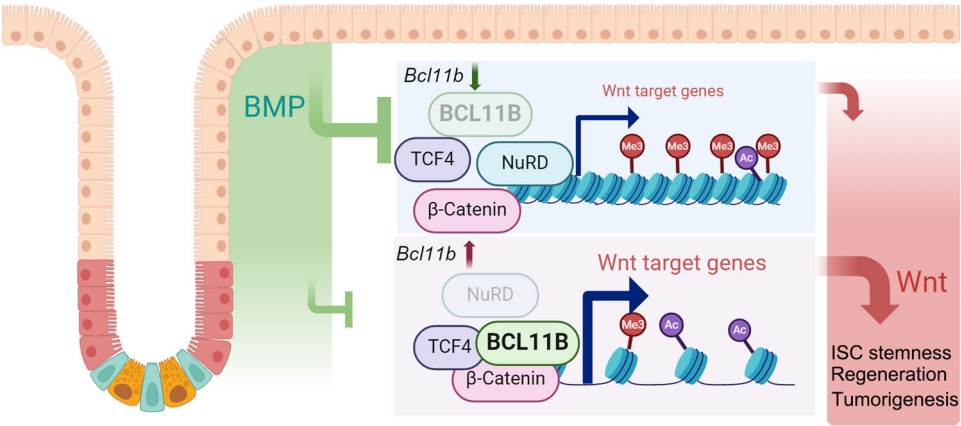

**Figure 7. A Schematic overview of the role of Bcl11b in cross-talk of Wnt and BMP signaling.**

BMP and Wnt signaling exhibit a opposite gradient pattern along the villus-crypt axis. BMP suppresses the expression of Bcl11b, which promotes Wnt signaling. At the bottom of crypt, Bcl11b expression is high due to low BMP activity. Thus, elevated Bcl11b enhances the β-catenin-TCF4 interaction and inhibits the NuRD activity, thereby increasing chromatin accessibility and activating Wnt target genes transcription. Consequently, Wnt activation maintains ISC stemness and promotes epithelium regeneration and tumor formation.

medium (crypt culture medium plus 2.5 μM CHIR-99021, 10% Wnt-3a conditional medium and 6.67 μM Blebbistatin).

### Virus production and organoid infection

Lentivirus was used for overexpressing *Bcl11b* in the organoids. Firstly, we modified pLVX-TRE3G vector (Clontech, 631193). Puromycin resistance gene was replaced with the blasticidin S resistance gene, named pLVX-TRE3G-bsd. Mouse Bcl11b cDNAs were constructed into the pLVX-TRE3G-bsd vector fused with Flag tag.

Lentivirus was produced in HEK293FT. The 10cm-dish cells were transfected with 4.3 μg of the plasmids pLVX-TRE3G-bsd expressing Bcl11b, 4.3 μg Vsvg and 6.43 μg Δ8.9, using Polyetherimide (PEI, Polysciences, PT-101-01N), and cultured with G418-free medium. Three days post-transfection, the supernatant was passed through a 0.45-μm filter, added lenti-Concentin Virus Precipitation Solution (ExCell Biology, EMB810A-1S) and refrigerated at 4 °C overnight. After $1500 \times g$ centrifugation at 4 °C for 30 min, lentiviral pellet was resuspended in 250 μL of infection medium (the ENR medium containing 6.67 μM blebbistatin, 2.5 μM CHIR-99021, 10% Wnt3a conditional medium and 10 μg/mL polybrene (Macgene, MC032)).

Before virus infection, organoids, derived from *R26-M2rtTA;Vil-CreERT2* mice, were cultured with the expansion medium plus 10 mM nicotinamide for 2 days. Then, the organoids were digested with TrypLE (Gibco, 12604021) for 5 min at 37 °C and resuspended with infection medium containing virus. Add 10 μL Matrigel into each well of 48-well plate and incubate at 37 °C for 10 min to solidify Matrigel. Add 250 μL of infection medium containing cells and virus on the solidified Matrigel and incubate overnight at 37 °C. Next day, remove the infection medium and wash the virus with warm PBS. Then, overlay 10 μL Matrigel and culture the organoids with the ENR medium containing 6.67 μM blebbistatin, 2.5 μM CHIR-99021 and 10% Wnt3a conditional medium. Two days after infection, change the medium with ENR plus 2.5 μg/mL blasticidin

S (Selleck, S7419). At 2–3 days post infection, protein expression was induced with 10 μM doxycycline (Selleck, S4163).

### Immunofluorescence and immunohistochemistry

To examine GFP expression, intestine isolated from mice was washed with cold PBS for several time, fixed with 4% formaldehyde solution and embedded in OCT (Sakura) after dehydrating in 30% sucrose overnight. The sections were prepared with freezing microtome (Leica) followed by observation with Olympus FV3000 confocal microscope with 20× magnification setting on the Olympus 20×/0.3 air objective. To monitor GFP expression in cultured organoids, isolated organoids were first fixed for 1 h with 4% paraformaldehyde at room temperature. The sections or organoids were then incubated with 4′,6-diamidino-2-phenylindole (DAPI) at room temperature for 1 h. For immunohistochemistry, formalin-fixed, paraffin-embedded intestine sections (5 μm) were de-paraffinized in isopropanol and graded alcohols, followed by antigen retrieval, and endogenous peroxidase quenched by $H_2O_2$. Sections were then blocked with 5% serum for 30 min and incubated overnight with primary antibody at 4 °C. Primary antibodies for immunohistochemistry were rabbit anti-Ki67 (Abcam, ab15580, RRID: AB_443209, 1:100), mouse anti-E-cadherin (BD, 610182, RRID: AB_397581, 1:300), mouse anti-β-catenin (Sigma, C7207, RRID: AB_476865, 1:100), rabbit anti-Bcl11b (Proteintech, 55414-1-AP, RRID: AB_11182609, 1:500) and rabbit anti-lysozyme (Dako, F0372, 1:200). Secondary horseradish peroxidase-conjugated anti-rabbit antibody (Invitrogen, 31460, RRID: AB_228341, 1:200) was added for 2 h followed by development with DAB substrate (Dako) according to the manufacturer's recommendations. For hematoxylin-eosin or alcian blue staining, the rehydrated slices were stained with the hematoxylin and eosin (Beyotime, C0105) or alcian blue (Baso, BA4087B) for 2 min and then washed. Hematoxylin-eosin staining and immunohistochemistry images were captured using Magscanner machine (KF-PRO-120-HI, KFBIO) with 20× magnification

setting on the 20×/0.75 air objective and analyzed using KF-viewer software. A fluorescence kit based on TSA technology (Recordbio Biological Technology, Shanghai, China) was employed to stain the BCL11B in the sections following the manufacturer's instructions. Bcl11b-stained images were captured using Olympus EVIDENT VS200 with 20× magnification setting on the Olympus 20×/0.3 air objective, and analyzed using OlyVIA (Olympus). To monitor GFP expression in cultured organoids in vitro, immunofluorescence images were captured using High Content Screening System (Opera Phenix, PerkinElmer) with 10× magnification setting on the 10×/0.3 air objective, and analyzed using Harmony 4.9 software. The 488 nm laser and 568 nm laser were used to observe GFP and PI signaling.

### Flow cytometry

Fresh intestinal crypts or cultured organoids derived from indicated mice were incubated in TrypLE (Gibco, 12604021) for 20 min at 37 °C to obtain single-cell suspension. For cultured organoids, the organoids embedded in Matrigel were first suspended in cold PBS after removal of medium and were pelleted by centrifugation (3 min at 300–400 × $g$). The dissociated cells were passed through 40 μm cell strainer (BD) and single Lgr5-GFP high cells were analyzed (CytoFlex LX, Beckman) or sorted by flow cytometry (MoFlo XDP, Beckman).

### In situ hybridization

In situ hybridization was done as previously described (Qi et al, 2017). Briefly, samples were then dehydrated and embedded in paraffin, sectioned and processed to in situ hybridization. The sections were de-waxed, rehydrated, pretreated and hybridized overnight for 12 h at 65 °C with digoxigenin-labeled RNA probes. After wash, sections were incubated in blocking solution for 2 h followed by incubation with alkaline phosphatase-conjugated anti-digoxigenin (1:2000; Roche) overnight at 4 °C. The sections were then incubated with AP substrate (Roche). The Olfm4 probes were generated through in vitro transcription with in vitro transcription kit (Roche).

### qRT–PCR

Total RNA was extracted with RNeasy Mini Kit (Qiagen) and cDNA was prepared using Revertra Ace (Toyobo). qRT–PCR was performed with TransStart Green qPCR SuperMix (Transgen Biotech) in triplicates on a LightCycler 480 (Roche) with Gapdh as the reference gene. Data were analyzed according to the ΔCT method. Primer sequences are listed in Appendix Table S1.

### Immunoblotting and immunoprecipitation

Protein lysates were prepared from intestinal epithelium or organoids. Immunoblotting was performed as previously described (Gao et al, 2008). Co-Immunoprecipitation were performed with Dynabeads Protein A/G immunoprecipitation Kit (10007D, Invitrogen) as the manufacturer's instructions. Transfer 10 μL Dynabeads Protein A and 10 μL Dynabeads Protein G to a tube per sample, place on magnet and remove supernatant. Beads were resuspended in 200 μL Binding&Washing Buffer containing 1 μg anti-Flag or indicated antibody per sample and incubated 4 h with rotation at 4 °C, placed on magnet and washed with Ab Binding&Washing Buffer. Organoids derived from the crypts of *R26-M2rtTA;Vil-CreERT2* mice were infected with *Bcl11b*-

expressing lentivirus, treated with doxycycline for 3 days. Cells were transfected with plasmids as indicated in figure legend using PEI. An entire 24-well plate doxycycline-induced organoids or one well of 6-well plate HEK293FT cells were harvested by PBS and lysed in 300 μL ice-cold IP lysis buffer (10 mM Tris-HCl, pH 7.5, 100 mM NaCl, 0.5% NP-40, 1 mM EDTA, 1 mM PMSF, 50 mM NaF, 2 mM $Na_3VO_4$, protease inhibitor cocktails (Roche, 04693132001)) for one hour. After centrifugation at 13,000 rpm for 10 min, 30 μL supernatant was collected for input, and the rest of the supernatant was immunoprecipitated with specific antibodies-binding Dynabeads with rotation overnight at 4 °C. The immunoprecipitants were washed three times with PBS. The Dynabeads-antibody-antigen complex was gently resuspended in 30 μL loading buffer and analyze by immunoblotting or performed following LC-MS experiments.

The following primary antibodies were used: rabbit anti-Bcl11b (Proteintech, 55414-1-AP, RRID: AB_11182609, 1:500), rabbit anti-HA (Santa Cruz, sc-7392, RRID: AB_627809, 1:1000), mouse anti-Myc (Cell Signaling Technology, 2276, RRID: AB_331783, 1:1000), mouse anti-β-catenin (Santa Cruz, sc-7963, RRID: AB_626807, 1:300), mouse anti-tubulin (Proteintech, 66031-1-Ig, RRID: AB_11042766, 1:20000), rabbit anti-Tcf4 (Cell Signaling Technology, 2569, RRID: AB_2199816, 1:1000), rabbit anti-Rbbp4 (Santa Cruz, sc-373873, RRID: AB_10918458, 1:1000), rabbit anti-HDAC1 (Cell Signaling Technology, 34589, RRID: AB_2756821, 1:1000), rabbit anti-Mbd3 (Santa Cruz, sc-166319, RRID: AB_2139753, 1:1000), goat anti-Lamin B (Santa Cruz, sc-6217, RRID: AB_648158, 1:1000), mouse anti-Flag (Sigma, F3165, RRID: AB_259529, 1:40,000), mouse anti-GAPDH (ZSGB-Bio, TA-08, RRID:AB_2747414, 1:10,000), mouse IgG polyclonal antibody (Sigma-Aldrich, 12-371, RRID: AB_145840), rabbit anti-Hes1 (Cell Signaling Technology, 11988S, RRID: AB_2728766, 1:1000) and mouse anti-YAP (Santa Cruz, sc-101199, RRID: AB_1131430, 1:1000).

### LC-MS/MS protein identification

To identify the interacting proteins with BCL11B, LC-MS/MS were performed following immunoprecipitation experiments. Immunoprecipitation was performed using two entire 24-well plate doxycycline-induced Bcl11b expressing organoids as one sample with mouse IgG polyclonal antibody (Sigma-Aldrich, 12-371, RRID: AB_145840) as control and anti-FLAG (Sigma, F1804, RRID: AB_262044) antibody for Flag-BCL11B using Dynabeads Protein A/G as above described in the immunoblotting and immunoprecipitation method. Specifically, the immunoprecipitants-beads conjugations were washed five times with PBS and then twice with 50 mM $NH_4HCO_3$. The beads were resuspended with 100 μL 50 mM $NH_4HCO_3$. On-beads digestion and LC-MS/MS analysis was performed in the Protein Chemistry and Proteomics Facility at Tsinghua University. The beads-pulldown proteins were reduced by Tris (2-carboxyethyl) phosphine and alkylated by chloroacetamide, then digested by trypsin for overnight. The supernatant was removed and collected into fresh, labeled tubes. Beads were washed twice in 100 μL of 0.1% formic acid and the supernatants were pooled. Peptide supernatants were dried by a speedvac, and re-dissolved by 20 μL of 0.1% formic acid for mass spectrometry analysis.

Samples were analyzed using a Vanquish Neo UHPLC system (Thermo Fisher Scientific), directly interfaced with an Orbitrap Astral mass spectrometer (Thermo Fisher Scientific. Peptides were loaded to a trap column (300 μm × 0.5 cm C18, PepMap™ Neo,

Thermo Fisher Scientific) with a max pressure of 1500 bar using mobile phase A (0.1% formic acid in $H_2O$), then separated on an analytical column (150 μm × 15 cm C18, ES906, Thermo Fisher Scientific) with a gradient of 4–60% mobile phase B (80% acetonitrile and 0.1% formic acid) at a flow rate of 800 nL/min for 23 min. The MS analysis was operated in data-independent acquisition (DIA) mode, with one full scan (380–980 *m/z*, Resolution = 240,000 at 200 *m/z*, analyzed by Orbitrap) at automatic gain control (AGC) of 5e6 with maximum injection time (IT) 5 ms, followed by DIA ms2 scan (150–2000 *m/z*, analyzed by Astral) at AGC of 5e5 with IT 3 ms and 25% of HCD collision energy. 390–980 *m/z* were divided into 300 DIA isolation windows, and the loop count is controlled by time of 0.6 s.

All proteomic data were searched against the mouse proteome (uniprot reviewed sequences downloaded 14 April 2023) using directDIA approach with Spectronaut software (version 18) following manufacture instructions. During the pulsar search setting, the search parameters were as follows: (i) full tryptic specificity was required; (ii) PSM FDR, peptide FDR and protein group FDR were all 0.01; (iii) Minimum number of amino acids allowed for a peptide is 7; (iv) up to two missed tryptic cleavage sites were allowed; (v) carbamidomethylation (C) were set as the fixed modifications; (vi) the correction factor of precursor ion and fragment ion mass tolerances were set at 1; and (vii) variable protein modifications were allowed for methionine oxidation and protein N-terminal acetylation. Detected peptides and proteins were controlled in 1% FDR by q-value. Peak ms1 area was used to quantify protein group. Precursor and protein posterior Error Probability (PEP) cutoff were 0.2 and 0.75, respectively.

The different abundance analysis was conducted using R (version 4.3.1). All proteins with missing values were filtered and the remaining data were log2 transformed for heteroscedasticity correction and normalization of data distribution. Following data preprocessing, the limma package (version 3.56.2) was used to identify differentially abundant proteins between the IgG and IP contrast groups (van Ooijen et al, 2018). The design formula was set using *model.matrix* function. *lmFit* function fits the linear model and calculates the t-statistic for each protein. *eBayes* function conducts the empirical Bayes test for t-statistic adjustment and returns log2FoldChange, *P* value, and adjusted *P* value with Benjamini–Hochberg correction. Proteins that are significantly enriched in the IP group (log2FoldChange > 1.95 and P.adjusted <0.00025) were subjected to Gene Ontology enrichment analysis.

### Protein production and purification

All genes (*Rbbp4*, *Bcl11b*, *Mbd3*, *β-catenin,* and *Hdac1*) were PCR-amplified and cloned into the pETMBP.3C vector to produce MBP-tag-fused recombinant proteins. Recombinant proteins were expressed in *E. coli* BL21-CodonPlus (DE3) with induction by 1 mM IPTG for 16 h at 18 °C. Following induction, *E. coli* cells were resuspended in binding buffer (50 mM Tris-Cl, pH 7.9, 1 M NaCl, and 10 mM imidazole), lysed with a high-pressure homogenizer and sedimented at 18,000 rpm for 30 min. The supernatant lysates were purified on Amylose Resin (NEB). After extensive washing with binding buffer, proteins were eluted with MBP elution buffer (50 mM Tris-Cl, pH 7.9, 1 M NaCl, and 10 mM maltose for MBP-tagged proteins), then purified with a HiPrep 26/60 Sephacryl S-200 HR column (17-1195-01; GE Healthcare) on an AKTA purifier (GE Healthcare), and eluted with PBS.

### Surface plasmon resonance (SPR) assay

The binding kinetics of β-catenin, RBBP4, MBD3 and HDAC1 to Bcl11b were analyzed by SPR using Biacore 8 K plus (Cytiva) at room temperature (25 °C). Bcl11b was immobilized on the chip CM5 surface in 10 mM sodium acetate buffer (pH = 5.0), and other proteins were applied as the analyte via serial dilution. Proteins used were exchanged into PBS buffer and serially diluted with PBST (PBS containing 0.05% Tween-20) as indicated in figure legends, sequentially injected into the chip at a flow rate of 30 μL/min for 2 min, and then allowed to dissociate. The binding curve at zero concentration of protein was subtracted as a blank from each experimental curve. After each cycle, a shot injection of 3 mM NaOH was used to regenerate the sensor surface. Data were analyzed, and kinetic constants were estimated using Biacore insight evaluation software (Cytiva).

### Generation of Bcl11b knockout cells

pSpCas9(BB)-2A-GFP (px458) and pSpCas9(BB)-2A-Puro (px459) plasmids were purchased from Addgene (Addgene plasmid 48138 and 49139). Three gRNAs targeting exon4 of human Bcl11b were designed using the online software (http://crispr.mit.edu) from the Feng Zhang Laboratory. gRNA #1 was cloned into px458 and the other two gRNAs (gRNA#2 and gRNA#3) were cloned into px459 using the BbsI restriction enzyme sites. LoVo cells were transfected with either gRNA pair 1 (gRNA#1plus gRNA#2) or gRNA pair 2 (gRNA#1 plus gRNA#3) using Lipofectamine 2000 (Life Technologies). After 24 h, cells were changed to fresh medium containing 1 μg/ml puromycin. Two days after puromycin selection, GFP+ cells were picked through FACS and cultured in a 96 well plate as single clones. Genomic DNA was purified from individual clones for PCR genotyping. Then immunoblotting was used to further confirm the knockout. The pair 1 gRNAs induce 1803bp deletion in the exon4 and pair2 gRNAs induce 1467 bp deletion in exon4. The Bcl11b KO-1 was generated by pair 1 gRNAs and Bcl11b KO-2 was generated by pair 2 gRNAs. The gRNA oligonucleotides were as follows: gRNA#1: 5'-GCACGGCTTCCG-CATCTACC-3', gRNA#2: 5'-GACGCACGGGCAGATCGGCA-3', gRNA#3: 5'-GTGCTCGGACGACGTGGCGA-3'.

### Reporter assay

Cells were transfected with various plasmids as indicated, and at 12 h post-transfection, the cells were treated with growth factors. One day later, the cells were harvested, and luciferase activities were measured by aluminometer (Berthold Technologies). pGL3-BRE-luciferase vector, pGL3-CAGA-luciferase vector, pGL3-NF-κB-luciferase vector, Topflash-luciferase vector, Fopflash-luciferase vector were generated to monitor BMP, TGF-β, NF-κB and Wnt signaling, respectively. pGa981-Rbpj-promoter-luciferase and pGL3-CYR61-promoter-luciferase (Zhang et al, 2024) were used to monitor Notch and Hippo signaling, respectively. Reporter activity was normalized to the co-transfected Renilla. For the Topflash reporter, the activity was normalized to Fopflash. Experiments were repeated at least three times.

### ATAC-seq

For miniATAC-seq, 5000 ISCs sorted at day 9 after tamoxifen injection in control and *Bcl11b*-cKO mice (*n* = 3 mice per group), respectively. The library was constructed with TruePrep DNA library Prep Kit V2 for illumine (Vazyme, TD202 and TD502). Amplified library was purified using AMPure beads and sequenced by the Illumina Novaseq 6000.

### In situ chromatin immunoprecipitation sequencing (ChIP-seq) and CUT&Tag ChIP-seq

Alive cells were sorted from Bcl11b-Flag overexpressing organoids treated with doxycycline for 5 days, then in situ ChIP-seq procedure was performed based on previous report (Wang et al, 2019). Briefly, 500,000 cells were fixed with 0.25% formaldehyde and 80% methanol. Cells were incubated with mouse IgG polyclonal antibody (Sigma-Aldrich 12-371, RRID: AB_145840), anti-Flag (Sigma, F1804, RRID: AB_262044) antibody at 4 °C overnight and secondary antibody (Invitrogen, A-31571, RRID: AB_162542) for 10 min. Cells were incubated with 12 µg/mL MEA/B-PAT at 4 °C for 1 h to bind antibody. Then genome DNA was fragmented with reaction buffer (10 mM TAPS-NaOH, 5 mM MgCl$_2$, 1X cocktail and 0.01% Digitonin), EDTA was added to the mixture to stop the fragmentation reaction. Cell lysate was resuspended with Lysis buffer (10 mM Tris-HCl pH = 8.5, 0.05% SDS, 0.1 mg/ml Proteinase K) and incubated at 65 °C for 1 h to release DNA fragments and then incubated at 85 °C for 15 min to deactivate Proteinase K. 1.8% Triton X-100 was added before incubation at 37 °C for 1 h to quench SDS in the reaction. For ChIP-seq, DNA was directly used as template, amplification was performed with 2X KAPA HIFI HotStart polymerase (KAPA, KK2601). The library was purified with 1.4X AMPure beads and sequenced with the Illumina Novaseq 6000. For CUT&Tag, 100,000 ISCs were sorted into 500 uL PBS at day 9 after tamoxifen injection in control and Bcl11b-cKO mice ($n = 3$ mice per group), respectively. For H3K27me3 and H3K27Ac ChIP, ISCs were lysed by the NovoNGS CUT&Tag 3.0 High-Sensitivity Kit (Novoprotein, H259-YH01), and the lysates were incubated with H3K27me3 (Abcam, ab6002, RRID: AB_305237, 1:50) and H3K27Ac (Cell Signaling Technology, 8173P, RRID: AB_10949503, 1:50) antibody at 4 °C overnight. Smad1 and Smad4 ChIP was performed as previously described (Qi et al, 2017). Briefly, the organoids were derived from Lgr5-EGFP-IRES-creERT2 mice and cultured with ER medium for 4 h. Then, 1,000,000 Lgr5$^+$ ISCs were sorted from cultured organoids as each sample. The cells were crosslinked with 1% formaldehyde for 10 min at room temperature and performed the following lysis and procedure according to the manufacturer's instructions (iDeal ChIP-seq kit, Diagenode) using anti-Smad1 (CST,6944, RRID: AB_10858882) and anti-Smad4 (purified by our lab) (Fei et al, 2010). For ChIP-qPCR, the immunoprecipitated DNA was analyzed using real-time PCR with a lightCycle-480 system, as descried detail in the qRT–PCR part. ChIP-qPCR experiments included 3 replicates and primers were available in Appendix Table S1.

### Bioinformatic analysis of ATAC-seq and ChIP-seq data

HISAT2 was used to align the sequences to the mouse genome and generate bam files. After deprived of PCR duplicates using Picard tools, Deeptools (3.3.1) bamCoverage (CPM normalized and extended reads) was used to generate bigwig files from bam files. MACS2 (v2.2.5) was used for peak calling and to generate bed files from aligned reads. The HOMER (v4.10.0) annotatePeaks.pl was used to annotate the peaks (Dataset EV1 and EV3). Visualization the binding peaks of bigwig files using IGV (2.6.2). For ATAC-seq, our parameter settings are: p-value < 0.05, shift is −100, extsize is 200 and nomodel, the remaining parameters are default. For ChIP-seq, the IP samples are the treatment group, and the IgG samples as control group. We choose the peaks with log$_2$(fold enrichment) > 3

and q value < 0.01 as the binding peaks of Bcl11b, the remaining parameters are default. The HOMER (v4.10.0) annotatePeaks.pl was used to annotate the peaks. Visualization the binding peaks of bigwig files using IGV (2.6.2). For ATAC-seq, the heatmap profiles of the control group and Bcl11b knockout peaks were drew by plotHeatmap of Deeptools. The quantification plots of the different clusters of control and Bcl11b knockout were summarized by plotProfile of Deeptools. For in situ ChIP-seq, the motif of Bcl11b was called by findMotifsGenome.pl of HOMER.

### Gene ontology (GO) analysis

Genes with $P < 0.05$ and absolute value of $|\log_2 FC| \geq 1$ were used for Gene ontology (GO) analysis, which was performed using the web tool: DAVID (http://david.adcc.ncifcrf.gov/). GO terms with $P < 0.05$ were determined to be statistically significant. The Venn diagrams were generated by bioVenn (http://www.biovenn.nl/).

### Single cell RNA sequencing (scRNA-seq) analysis

A scRNA-seq dataset (GSE186917) (Liu et al, 2023) of control mouse intestinal epithelium was utilized in our research. We detected Bcl11b expression in the original processing results of the wild-type scRNA-seq library, and visualized Bcl11b expression in the original UMAP map. Additionally, the scRNA-seq data (GSE178341) (Pelka et al, 2021) of intestinal epithelium was utilized to observe Bcl11b expression, and visualized in the 10x single cell loupe cell browser and Single Cell Portal (https://singlecell.broadinstitute.org/single_cell).

### Spatial RNA-seq and analysis

Mouse intestines were cleaned with PBS and cut longitudinally. Tissues were quickly embedded in Optimal Cutting Temperature compound (OCT, Sakura Tissue-TEK) on dry ice and stored at −80 °C. For spatial RNA-seq experiment, OCT blocks were cut with a pre-cooled cryostat at 10 µm thickness, and sections were transferred to the capture regions on the Visium 10X Genomics slides. Before the process of the formal spatial RNA-seq protocol, several frozen sections were subjected to RNA extraction (Qiagen, 74104). To get optimized time for permeabilization, the tissue optimization experiment (10X Genomics, 1000193) was performed and the fluorescence was captured by microscopy (Nikon), determining 3 min as tissue permeabilization time. Spatial RNA-seq was performed following the manufacturer's instruction (10X Genomics, 1000184). The experimental slide was fixed and stained with H&E, and images were captured (KEYENCE). The sequence libraries were processed according to manufacturer's instructions (10X Genomics, Visium Spatial Transcriptomic).

The sequencing data generated from the spatial RNA-seq experiment were aligned onto H&E-stained images using Space Ranger v.1.0.0, a toolkit developed by 10X Genomics. Subsequently, the aligned UMI count matrices and spatial images of the small and large intestine samples were imported into R using the Seurat R package for further analysis. SCTransform function implemented in Seurat was employed for comprehensive data normalization, gene expression scaling, and regression of mitochondrial gene expression percentage and cell cycle stage effects to ensure the removal of batch effects and improved the accuracy of downstream analyses. In order to integrate the spatial information obtained from the spatial RNA-seq data with the single-cell RNA-seq data, we utilized the TransferData function of Seurat. By leveraging the single-cell RNA-seq data as an

anchor set, this method enabled us to predict corresponding clusters for each spatial spot. Each spot was then assigned the cluster with the highest score, providing valuable insights into the transcriptional heterogeneity and spatial organization of the tissue.

### Gene expression analysis from TCGA datasets

Utilizing GDCRNATools v1.16.6 in R v4.2.1, TCGA gene expression count data and clinical data were acquired. Transcript counts were normalized through the TMM normalization method, followed by differential expression analysis using limma (Ritchie et al, 2015). The ggpubr v0.4.0 package was employed to generate the boxplot, and the stat_compare_means function from the ggpubr R package was utilized to determine statistical significance.

### Statistics

No statistical methods were used to predetermine sample size. The experiments were not randomized, and the investigators were not blinded to allocation during experiments and outcome assessment. All experiments were carried out with at least three biological replicates, and each experiment was independently repeated at least three times. Data shown in column graphs represent the mean ± S.D., as indicated in figure legends. The normality of all samples was tested using the Shapiro–Wilk test. Parametric tests were chosen when data was normally distributed based on Shapiro–Wilk Test. Normally distributed data were analyzed using parametric t-test with Welch's correction, one-way ANOVA with Tukey's multiple comparisons test, or two-way ANOVA analysis with Tukey's multiple comparisons test was used to compare difference as indicated in figure legends. Statistical analysis was performed with GraphPad Prism10 or Image J software, and schematic overview was created with BioRender.com. For immunoblotting, immunohistochemistry, in situ hybridization and immunofluorescence, representative images were shown.

## Data availability

The ATAC-seq and ChIP-seq (in situ ChIP-seq, CUT&Tag-seq) data generated in this study are publicly available through the Gene Expression Omnibus (GEO) with the accession code GSE255138. The bulk RNA-seq data GSE247044 was used to analyze the expression of *Bcl11b* upon *Alk3* deletion. The scRNA-seq data GSE186917 and GSE178341 were used to explore the expression pattern of *Bcl11b* (Liu et al, 2023; Pelka et al, 2021). The microscopy images of this paper are collected in the BioImage Archive: 10.6019/S-BIAD1299. All codes that the main steps of the analysis and data are available from the corresponding author under request.

The source data of this paper are collected in the following database record: biostudies:S-SCDT-10_1038-S44318-024-00276-1.

## Peer review information

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

## Acknowledgements

We thank Drs. Pengtao Liu and Sylvie Robine for mice. We thank Dr. Kexin Kang for protein purification, Dr. Aibin He and Min Liu (Peking University) for the advice on in situ ChIP-seq. We are also grateful to Dr. Hongshuang Li and Dr. Hui Zhang (The State Key Laboratory of Membrane Biology, Tsinghua University) for assistance on microscope and FACS, Zi Yang (Protein Chemistry and Proteomics Facility at Technology Center for Protein Sciences of Tsinghua University) for assistance on SPR assay, and Dr. Yuling Chen (Protein Chemistry and Proteomics Facility at Tsinghua University) for LC-MS/MS protein identification. We thank the laboratory animal resources center of Tsinghua University for generating and handling of mice. We thank Dr. Xiaohua Yan for pcDNA3.1-YAP and pGL3.0-Cyr61-promoter-luciferase plasmids, Dr. Xiaoyu Hu and Dr. Liu Yang for anti-Hes1 antibody. This work was supported by grants from the National Natural Science Foundation of China (92354306 to Yuan Liu and 31988101 to Y-GC), the National Key Research and Development Program of China (2023YFA1800603), the Natural Science Foundation of Jiangxi Province (20224ACB209001), Beijing Science and Technology Plan (Z231100007223006) and Shenzhen Medical Research Fund (B2302022) to Y-GC.

## Author contributions

**Yehua Li**: Conceptualization; Resources; Data curation; Formal analysis; Supervision; Validation; Investigation; Visualization; Methodology; Writing—original draft; Project administration; Writing—review and editing. **Xiaodan Wang**: Data curation; Software; Formal analysis; Visualization; Methodology. **Meimei Huang**: Resources; Data curation; Formal analysis; Validation; Investigation; Visualization; Methodology; Writing—original draft; Writing—review and editing. **Xu Wang**: Resources; Software; Formal analysis; Investigation; Visualization; Methodology; Writing—review and editing. **Chunlin Li**: Software; Formal analysis. **Siqi Li**: Formal analysis; Investigation; Visualization; Methodology. **Yuhui Tang**: Investigation; Methodology. **Shicheng Yu**: Software; Formal analysis. **Yalong Wang**: Resources; Investigation. **Wanglu Song**: Software; Formal analysis. **Wei Wu**: Conceptualization; Resources; Supervision. **Yuan Liu**: Conceptualization; Resources; Data curation; Software; Formal analysis; Funding acquisition; Validation; Investigation; Visualization; Methodology; Writing—original draft; Writing—review and editing. **Ye-Guang Chen**: Conceptualization; Supervision; Funding acquisition; Writing—original draft; Project administration; Writing—review and editing.

Source data underlying figure panels in this paper may have individual authorship assigned. Where available, figure panel/source data authorship is listed in the following database record: biostudies:S-SCDT-10_1038-S44318-024-00276-1.

## Disclosure and competing interests statement

The authors declare no competing interests.

