## [Peer Review File · The EMBO Journal]

BMP balances Wnt signaling via the Bcl11b-regulated NuRD complex to maintain intestinal stem cells

Yehua Li, Xiaodan Wang, Meimei Huang, Xu Wang, Chunlin Li, Siqi Li, Yuhui Tang, Shicheng Yu, Yalong Wang, Wanlu Song, Wei Wu, Yuan Liu, and Ye-Guang Chen

Corresponding authors: Ye-Guang Chen (ygchen@tsinghua.edu.cn) , Yuan Liu (liu-yuan@mail.tsinghua.edu.cn)

Review Timeline:

Submission Date:	1st Apr 24
Editorial Decision:	7th May 24
Appeal:	21st May 24
Editorial Decision:	7th Jun 24
Revision Received:	1st Aug 24
Editorial Decision:	17th Sep 24
Revision Received:	22nd Sep 24
Accepted:	2nd Oct 24

Editor: Ieva Gailite

Transaction Report:

Dear Dr. Chen,

Thank you for submitting your manuscript for consideration by The EMBO Journal. We have now received three reviewer reports on your manuscript, which are included below for your information. Based on these comments, we unfortunately had to conclude that the study is not a sufficiently strong candidate for publication in The EMBO Journal.

As you can see, while the reviewers find the topic per se of interest, reviewers #1 and #2 also indicate multiple substantial concerns with the experimental approach and the used controls, and they find that the proposed model of Bcl11b action is not sufficiently supported by the data. These concerns are further echoed by reviewer #3. Given these opinions from good experts in the research field and since the raised concerns affect the central conclusions of the study and substantial experimental work with an uncertain outcome would be needed to address them, I am afraid that we cannot offer further proceedings towards publication in The EMBO Journal.

Thank you in any case for the opportunity to consider this manuscript. I regret that I could not communicate more positive news this time, but I nevertheless hope that you will find our reviewers' comments helpful for further improvement of the manuscript.

Yours sincerely,

Ieva Gailite

Referee #1:

Li et al. report that Bcl11b, a BMP-SMAD repressed gene, plays a role in the intestinal stem cell niche and they propose it acts to repress the NuRD complex that inhibits gene expression. Acute knockout of Bcl11b decreases stemness and expression of intestinal stem cell genes, although these acute knockout experiments may have a seriously flawed control. Acute knockout of Bcl11b impairs recovery of the gut from radiation and DSS injury. Forced over-expression of Bcl11b enhances TOPFLASH, a plasmid based reporter assay. Addressing mechanism, they find that over-expressed Bcl11b pulls down β -catenin and interact with β -catenin in SPR. Confusingly, Bcl11b IP does not find β -catenin nor TCF4 in a MS study. Over-expressed Bcl11b binds to >6000 genes, i.e., a third of the genes, so it is hard to know what to make of that. It dilutes the claim that Bcl11b specifically interacts with TCF4/ β -catenin. IP of over-expressed Bcl11b confirms previous reports of interaction with the NuRD complex components, and KO of Bcl11b decreases chromatin accessibility in half the genome as far as I can tell, by ATAC-seq. The evidence that Bcl11b is specifically involved in colon cancer is circumstantial; the knockout slows growth, but it's not clear this is specific to APC-mutant cancers.

Overall, the authors present a very large amount of data that suggest a role for Bcl11b in regulation of the NuRB complex, uneven data that Bcl11b directly interacts with Tcf4 and β -catenin, and mouse data for an important role for Bcl11b in intestinal stem cells, but with a caveat that a key control may be absent.

Major Issues:

Acute activation of Cre causes DNA damage and stem cell toxicity, and it is not at all clear this was controlled for in their mouse studies. This is a well-established problem. See "Genome Toxicity and Impaired Stem Cell Function after Conditional Activation of CreERT2 in the Intestine", pmid: 30449703. The authors must clearly state the genotype of the controls in Fig 2E and 2F, and Fig 3D-G, Fig 4C-D. The authors must test if the results in these experiments are due to Cre toxicity versus Bcl11b excision.

Fig 2E: I expected four conditions - control and Bcl11b ko, +/- tamoxifen. The data presented do not show any control for tamoxifen effect on stem cells, which is a known problem (pmid: 23415913).

The fold activation of TOPFLASH in HEK293 cells is surprisingly large, yet the reporter is plasmid based. Very few genes have

this kind of effect on Wnt signaling. And then, HCT116 cells have mutant active β -catenin already, but expression of Bcl11b alone has a minimal effect. This is confusing and not consistent with the simple model. Is NuRD important for regulating simple promoters on plasmids? What happens to endogenous Wnt target genes in these cells?

ChIP-seq performed with over-expressed FLAG-Bcl11b is going to be full of over-expression artifacts due to protein binding to irrelevant low-affinity promoters. This artifact is reflected in the over 6000 Bcl11b target genes (Figs 4I-K, S4D-E, tables S2). I do not find this helpful; the experiment should be repeated with endogenous expression levels of Bcl11b, or removed.

Minor concerns that should be addressed:

Fig 1A: ERN, or ENR? Label figure and legend consistently.

Please define ENR and ERB in results or legend of Fig 1A. Neither is defined in the manuscript. I'm guessing the ERB has BMP in it?

Fig 1A should have a broken scale so we can see Bcl11b more clearly.

Fig 1H: There is no description of the ChIP-qPCR antibodies or primers or methods in the manuscript.

The Flag-Bcl11b-expressing lentivirus is not described adequately. I don't see how it is dox regulated. What is the transduction efficiency? Is there selection? Please expand.

In figure 2 it shows acute KO of Bcl11b causes a marked decrease in stem cell markers and stemness. In contrast, in fig 4D, acute KO of Bcl11b has no effect on the number of Ki67+ cells. Please address these apparently contradictory results.

Please comment on this prior report: Bcl11b SWI/SNF-complex subunit modulates intestinal adenoma and regeneration after γ -irradiation through Wnt/ β -catenin pathway <https://academic.oup.com/carcin/article/36/6/622/276606>

The authors do LC-MS/MS of Flag-Bcl11b IP from organoids. This seems quite challenging with tiny amounts of protein. Please report the number of organoids, how they were harvested, lysed, and the protein yield prior to IP.

The analysis of the MS data is not well described. What filtering for SEQUEST quality, number of peptides, adjusted P value was used? Neither of the MS datasets find β -catenin nor TCF7. Why is this?

Referee #2:

General summary and opinion: This study by Li et al is the description of a comprehensive data set that shows that Bcl11b, a gene that is downregulated upon BMP pathway activation (for example upon treatment with endogenous BMP ligands), act as a transcriptional regulator of WNT target genes. It is nicely shown that Bcl11b-deficient cells have lower expression of WNT target genes, whereas Bcl11b-overexpressed cells have higher Wnt target gene expression. Mechanistically, the authors on the one hand propose a direct interaction of Bcl11b with WNT components B-Cat/TCF4 for Wnt pathway specific aspects, on the other hand suggest a more general transcriptional regulation of Bcl11b, through modulating of the NuRD complex. Furthermore, the mechanistic work is supported by sturdy in vivo data showing that Bcl11b is important for reparative mechanisms in the intestine (radiation of small intestine, and using DSS-colitis in the colon), as well as potentially relevant for tumorigenesis. Indeed showing an important physiological role for Bcl11b in the intestinal stem cell system. Overall, the study is comprehensive and technically sound, and sufficient replicates etc. was used. The data is also presented clearly.

Specific major concerns:

- BMP regulating Bcl11b expression: The strongest evidence that this is a direct regulatory mechanism actually only comes from Fig. 1H (Smad occupancy at gene). All the data before just shows that Bcl11b behaves similar to 'stem cell' genes, and the changes (increase/decrease) can be a consequence of a change in # of stem cells (like in the Alk3 KO (similar to their previous study), that will get an expansion of the stem cell zone). The authors could do a short term (1/4/8h or so) BMP4 treatment to show rapid repression (preferably prior to changes in Wnt genes, as this would be a consequence of the changed Bcl11b expression such as the authors propose), which would support direct target gene hypothesis. As is, there's not sufficient data to support that Bcl11b is a bona fide BMP-repressed gene.

Fig. 1H is further a bit confusing as this is ChIP data from sorted Lgr5 cells (which should not have active BMP signaling, nor is this a cell where Bcl11b is repressed (see rest of the figure!)). It would be helpful to have Smad1/4 negative genes as a control qPCR to show that this is specific for the site.

- Fig 2 and throughout (also the methods). The Lgr5 mouse line is a mosaic line where only ~30-40% expresses GFP. In the methods it is described that random parts were assessed for GFP levels during microscopy, however, this could lead to randomly determining positive vs negative GFP sites. Fig. 2A shows complete absence, but this is perhaps more likely due to

negative patch (as the qPCR shows just a 50% reduction in 2B. Related to this figure, the authors suggest no changes were found in Paneth cells. However, changes in PCs take longer (4-6 weeks after deletion) to be observed. Original Sato Paneth cell 'niche' paper (Sato et al 2011, Nature) shows deletion of certain genes takes 6 weeks to observe reduction of Paneth cells. I recommend deleting Bcl11b and look 4-6 weeks later.

- Specificity of Bcl11b in regulating Wnt: ChIP-seq identified ~7000 genes in ISCs to be under the control of Bcl11b. To me, this is basically half if not more of all genes expressed in ISCs (genome = 20k genes, ~12-15k expressed in specific cell types). As ISCs are high Wnt condition, it's likely to then find strong overlap. To make this specific, the authors should pick something that's also high in ISCs (such as Notch-Hes1 and its target genes), and show that Bcl11b does not regulate those genes (or similar line of reasoning). To me, a more plausible conclusion is that Bcl11b is just important for gene expression, facilitating whatever signaling takes place (high Wnt in ISCs in this case). Don't get me wrong, I really like the paper, I just find that this specificity is in contrast to a gene that seems to control gene expression so broadly. Furthermore, such 'aspecific' hypothesis also fits better with the proteomics data (that didn't identify Bcat/TCF4, but found more broad partners in the nucleus), rather than the OE interaction data with B-cat (again a negative other relevant TF would be good as a negative control).

Of note, my first 2 points are important to address experimentally, or verify method-technically. My 3rd point is definitely open to interpretation, but feel should be addressed as the authors currently claim a Wnt-specific role and this is not fully supported by the data with a lack of negative controls. The lack of BMP/TGFb/TNF reporter line activity is not sufficient, rather a Yap or Hes1 reporter would indeed be more relevant to ISC signaling state.

Minor concerns:

- Please provide high magnification of Bcl11b staining (Fig. 1G) to see stem-cell / Paneth cell zone.
- Throughout: please when using statistical analysis of low n, one cannot assume normal distribution (so use non-parametric tests)
- Please use a mouse = 1n, example: Fig. 2A has many data points, but I don't think these are different mice (just individual crypts). Same for Fig. 2C/2G/2I, n= a well not 1 organoid, and preferably from different mice.
- Ensure X-ray was used (normally it's gamma-radiation machines that is used in patients for therapy, not the X-ray to image). That said, x-ray is gamma-radiation type so if tuneable this could have been used.
- Wording: please make sure to state when using ectopic overexpression of a plasmid to determine its role, or when looking at interactions (IP). Compared to looking at endogenous levels. Similar; OE and tagged IP experiments certainly are proof that proteins can interact, but also not proof that they are in a complex under normal conditions or in ISCs.

Referee #3:

The manuscript by Li and team explores the role of Bcl11b in intestinal stem cells. In figure 1, the authors use multiple lines of evidence to demonstrate that Bcl11b is repressed by BMP signaling and enriched in stem and progenitor cells in the crypt. In figure 2, Bcl11b is shown to be required for a full number of Lgr5+ ISCs both in vivo and in organoid models. Gain and loss of function systems show that Bcl11b expression correlates with ISC marker expression. In figure 3, irradiation and DSS models are used to show that Bcl11b is important for the intestine to resist damage from these treatments. In figure 4, cell culture assays and co-IP experiments point to interactions between Bcl11b and TCF and Bcatenin. ChIPseq analysis shows overlap between Bcl11b and TCF4 and Bcatenin binding sites, further supporting a role in regulating the WNT signaling pathway. In figure 5, the authors dig deeper into how Bcl11b may be regulating genes and find that BCL11B interacts with components of the NURD complex, and that loss of BCL11B leads to changes in ATAC and H3K27ac and H3K27me3 that correspond to reduced WNT target gene expression in the BCL11B mutant. Finally, figure 6 provides evidence that BCL11B supports cancer growth, as might be expected based upon its support of the WNT pathway.

This work is generally well-executed and provides novel insights into how the WNT pathway is regulated in normal intestinal homeostasis and in colon cancer. The authors frequently use both gain and loss of function genetic approaches and state-of-the-art techniques to make their discoveries. While there are some important concerns to be addressed to improve the rigor and readability of the work, the work will certainly make exciting contributions to the field.

Concerns:

The data haven't been made available to the reviewers and the number of replicates for many experiments is unclear. (for example, ChIPseq)

Please indicate where the datasets arise from in the figures and figure legends. For example, figure 1C was from a previous publication, but it's not so easy to find the data source. It would help readers to have a more direct connection to the data.

Please address this paper that finds opposite results in a mouse model of colon cancer:

<https://pubmed.ncbi.nlm.nih.gov/25827435/>

Cre-only controls should be used in the regeneration assays to test for off-target effects. It is not clear if these controls are in place.

It does not appear that there is a description of the overexpression construct. This is one example of several in the methods section that appears to lack key experimental details and should be improved so that others can try and reproduce the work.

** As a service to authors, EMBO Press provides authors with the possibility to transfer a manuscript that one journal cannot offer to publish to another EMBO publication or the open access journal Life Science Alliance launched in partnership between EMBO Press, Rockefeller University Press and Cold Spring Harbor Laboratory Press. The full manuscript and if applicable, reviewers' reports, are automatically sent to the receiving journal to allow for fast handling and a prompt decision on your manuscript. For more details of this service, and to transfer your manuscript please click on Link Not Available. **

Dear Dr. Gailite,

Thank you for reviewing our manuscript “BMP balances Wnt signaling via the Bcl11b-regulated NuRD complex to maintain Lgr5+ intestinal stem cells”. We are very grateful to the reviewers for their insightful comments and suggestions. We have carefully read the comments and feel that both Reviewer #2 and Reviewer #3 are positive for our work. Reviewer #1 raised the concerns including the control design and experimental approach. We can address all the concerns raised by the reviewers (see our rebuttal for details). We think that our work provides a novel insight on in intestinal stem cell biology, as stated by Reviewer 2 (Indeed showing an important physiological role for Bcl11b in the intestinal stem cell system. Overall, the study is comprehensive and technically sound, and sufficient replicates etc. was used. The data is also presented clearly) and Reviewer 3 (This work is generally well-executed and provides novel insights into how the WNT pathway is regulated in normal intestinal homeostasis and in colon cancer.... the work will certainly make exciting contributions to the field).

We politely disagree with Reviewer #1's concerns about our controls. As we have used mouse genetic manipulations to study intestinal stem cells for many years and published more than a dozen of papers (for instance: Zhao et al, 2015; Qi et al, 2017; Li et al, 2018; Li et al, 2019; Liu et al, 2023a; Liu et al, 2023b; Wang et al, 2024), we were aware of the possible stem cell toxicity induced by Cre activation. For this reason, we injected mice with the same amount of tamoxifen as the control to eliminate any non-specific effects of tamoxifen or Cre activation. For the organoid experiment, both control and Bcl11b cKO-derived organoids were treated with 4-hydroxytamoxifen (4-OHT) to avoid the toxicity of tamoxifen on stem cells. The detailed mice genotypes were described in the figure legend of Figure 2A. Since the control genotypes were the same throughout the entire manuscript, we did not repeatedly mention this every time. Meanwhile, we have also addressed this point in the Methods section. Reviewer #1 seems to miss the legend of Figure 2A and the description in the Methods. Thus, we do not think that Reviewer #1 judged our work objectively based on the assumption that we used wrong controls. To enhance the clarity, we have stated this in the text and labeled the detailed genotypes in legends for each figure panel.

Both Reviewer #1 and Reviewer #2 raised concerns about the ChIP-seq result. We have tested various Bcl11b antibodies, but none of them were suitable for ChIP-seq assay, particularly with the limited availability of organoids. We agree that conducting this assay with overexpressed Bcl11b is not a perfect approach. However, we respectfully do not agree with Reviewer #1's opinion that the binding of Bcl11b on the detected target genes were over-expression artifacts. Although background may increase after Bcl11b

over-expression, the result can also give us information about Bcl11b binding targets, when the specific signal is high enough. As the reviewers suggested, we have re-analyzed the ChIP-seq data with more restricted parameters to remove the low signals and included the result in the revised manuscript. By doing so, we identified 1863 Bcl11b binding sites in 1729 genes, a number similar to the Bcl11b targets reported in breast cancer by Cai's group (Bai et al, 2022). Among the Bcl11b targets, about 1/5 are also Wnt targets, which supports our conclusion that Bcl11b promotes Wnt signaling. If necessary, we will redo ChIP-seq assay by reducing Flag-Bcl11b expression in Bcl11b KO organoids to mimic the endogenous protein level.

Reviewer #1 and Reviewer #3 referred the previous published paper "Bcl11b SWI/SNF-complex subunit modulates intestinal adenoma and regeneration after γ -irradiation through Wnt/ β -catenin pathway" (2015, Carcinogenesis, 36(6):622-31). In this paper, Sakamaki et al. reported that Bcl11b represses intestinal epithelium regeneration and adenoma formation through inhibiting the Wnt/ β -catenin pathway. 1) They observed conventional *Bcl11b*^{KO/+}; *Apc*^{min/+} mice developed more adenoma than the *Apc*^{min/+} mice. As Bcl11b is critical for T cell development, the observed effect of Bcl11b heterozygosity on tumor formation may be attributed to T cell development deficiency, rather than direct effect of Bcl11b heterozygous deletion in the intestinal epithelium. And their reported promotion effect of Bcl11b heterozygosity on intestinal epithelium regeneration following radiation-induced injury was very subtle. Thus, their data cannot specifically address the role of intestinal epithelial Bcl11b in tumor formation and regeneration after injury. 2) To address the inhibitory effect of Bcl11b on Wnt signaling, they reported the inhibitory role of Bcl11b on Wnt signaling using the Wnt reporter Topflash-luciferase in CaCo2, SW480 and 293FT cells. However, they did not show the Topflash/Fopflash ratio. Fopflash is widely used as a negative control to examine Wnt signaling, and normalization to Fopflash is important to accurately indicate Wnt activation. In their results, Bcl11b also inhibited control Fopflash in SW480 cells, while Fopflash signal was too low in CaCo2 cells. In 293FT cells, just Topflash result was shown, therefore it is hard to know whether Bcl11b also inhibits Fopflash. Our results showed the ratio of Topflash/Fopflash, which is more accurate. 3) They reported that some of the Wnt target genes, including *Myc*, *Ccnd1* and *Jun*, were elevated in Lgr5+ cells from Lgr5-CreERT2; *Bcl11b*^{f/f} mice two months after 4-OHT induced Bcl11b knockout. However, in their cDNA microarray data, more Wnt target genes, including *Axin2*, *Sp5*, *Lef1*, *Mmp7* and *Tnfrsf19*, decreased after Bcl11b knockout, the later of which is consistent with our data. Therefore, we believe that our model is appropriate for examination of the role of intestinal epithelial Bcl11b.

We sincerely apologize for missing some experimental details. We have now added these in the revised manuscript. Attached is a point-to-point response for the reviewers' comments in the rebuttal.

I hope that you can reconsider your decision and give us a chance to revise the manuscript.

Sincerely yours,

Ye-Guang Chen

Referee #1:

Li et al. report that Bcl11b, a BMP-SMAD repressed gene, plays a role in the intestinal stem cell niche and they propose it acts to repress the NuRD complex that inhibits gene expression. Acute knockout of Bcl11b decreases stemness and expression of intestinal stem cell genes, although these acute knockout experiments may have a seriously flawed control. Acute knockout of Bcl11b impairs recovery of the gut from radiation and DSS injury. Forced over-expression of Bcl11b enhances TOPFLASH, a plasmid based reporter assay. Addressing mechanism, they find that over-expressed Bcl11b pulls down β -catenin and interact with β -catenin in SPR. Confusingly, Bcl11b IP does not find β -catenin nor TCF4 in a MS study. Over-expressed Bcl11b binds to >6000 genes, i.e., a third of the genes, so it is hard to know what to make of that. It dilutes the claim that Bcl11b specifically interacts with TCF4/ β -catenin. IP of over-expressed Bcl11b confirms previous reports of interaction with the NuRD complex components, and KO of Bcl11b decreases chromatin accessibility in half the genome as far as I can tell, by ATAC-seq.

The evidence that Bcl11b is specifically involved in colon cancer is circumstantial; the knockout slows growth, but it's not clear this is specific to APC-mutant cancers.

Overall, the authors present a very large amount of data that suggest a role for Bcl11b in regulation of the NuRB complex, uneven data that Bcl11b directly interacts with Tcf4 and β -catenin, and mouse data for an important role for Bcl11b in intestinal stem cells, but with a caveat that a key control may be absent.

Response: Thank you for insightful comments. As addressed below, we have clarified all of your concerns by adding more data, re-calculating the sequencing data, making clear labeling, etc. We have also modified the manuscript accordingly. We hope these improvements would address your concerns.

Major Issues:

Acute activation of Cre causes DNA damage and stem cell toxicity, and it is not at all clear this was controlled for in their mouse studies. This is a well-established problem. See "Genome Toxicity and Impaired Stem Cell Function after Conditional Activation of CreERT2 in the Intestine", PMID: 30449703. The authors must clearly state the genotype of the controls in Fig 2E and 2F, and Fig 3D-G, Fig 4C-D. The authors must test if the results in these experiments are due to Cre toxicity versus Bcl11b excision.

Response: We have used mouse genetic manipulations to study intestinal stem cells for many years and published more than a dozen of papers (Li et al, 2019; Li et al, 2018; Liu et al, 2023a; Liu & Chen, 2020; Qi & Chen, 2015; Qi et al, 2017; Wang & Chen, 2018; Wang et al, 2024; Zhao et al, 2015), we were aware of the possible stem cell toxicity induced by Cre activation. For this reason, we usually injected mice with the same amount of tamoxifen as the control to eliminate any non-specific effects of tamoxifen or Cre activation. The detailed genotype was clearly stated in the figure

legend of Figure 2A. Since the control genotypes were the same throughout the entire manuscript, including Fig 2E and 2F, and Fig 3D-G, Fig 4C-D, we did not repeatedly mention this every time. Meanwhile, we have also addressed this point in the Methods section. To enhance its clarity, we have included the description of control genotype in the text and legends for each figure panel, as per your suggestion.

Fig 2E: I expected four conditions - control and Bcl11b ko, +/- tamoxifen. The data presented do not show any control for tamoxifen effect on stem cells, which is a known problem (pmid: 23415913).

Response: We appreciate the reviewer for suggestions. Indeed, we have performed many controls, just as the reviewer suggested. In all the organoid experiments, we used organoids from Vil-CreERT2;Lgr5-GFP-IRES-creERT2 (control) mice treated with the same amount of 4-hydroxytamoxifen (4-OHT) to avoid the toxicity of tamoxifen on stem cells. As shown in Response Figure 1, we have implemented the controls as suggested by the reviewer. To keep conciseness and emphasize key points, we just omitted the EtOH (ethanol) group from the submitted manuscript and focused on the 4-OHT group to assess its effect on stem cells. We have now modified the new Figure 2 and included all the EtOH controls.

Figure for reviewers removed

The fold activation of TOPFLASH in HEK293 cells is surprisingly large, yet the reporter is plasmid based. Very few genes have this kind of effect on Wnt signaling. And then, HCT116 cells have mutant active β -catenin already, but expression of Bcl11b alone has a minimal effect. This is confusing and not consistent with the simple model. Is NuRD important for regulating simple promoters on plasmids? What happens to endogenous Wnt target genes in these cells?

Response: We found that Bcl11b, but not other candidates, greatly promoted Wnt reporter expression (Response Figure 2). Thanks to your suggestion, we realized that the TOPFLASH result of HCT116 was mis-labeled by messing up Wnt3a and Bcl11b in figure labeling. Sorry for the mistake. From Response Figure 3 that shows the correct labelling, we can see that Bcl11b over-expression greatly increases the Wnt reporter activity, while Wnt3a shows moderate enhancement, which is consistent with the fact that HCT116 cells harbor mutant active β -catenin. We have now corrected the new Figure 4B.

Figure for reviewers removed

Figure for reviewers removed

ChIP-seq performed with over-expressed FLAG-Bcl11b is going to be full of over-expression artifacts due to protein binding to irrelevant low-affinity promoters. This artifact is reflected in the over 6000 Bcl11b target genes (Figs 4I-K, S4D-E, tables S2). I do not find this helpful; the experiment should be repeated with endogenous expression levels of Bcl11b, or removed.

Response: Thanks for the insightful comments. We have tested various Bcl11b antibodies, but none of them were suitable for ChIP-seq assay, particularly with the limited organoid materials. The high number of Bcl11b target genes could be due to overexpressed Flag-Bcl11b in our experiment. We re-analyzed the ChIP-seq data with more restricted parameters by choosing the peaks with $\log_2(\text{fold enrichment}) > 3$ and $q \text{ value} < 0.01$ as the binding peaks of Bcl11b, and the remaining parameters were default. The HOMER (v4.10.0) `annotatePeaks.pl` was used to annotate the peaks. The reanalysis identified 1863 Bcl11b binding sites in 1729 genes, a number similar to the Bcl11b targets reported in breast cancer by Cai's group (Bai *et al.*, 2022) (new table S2). GO analysis of these sites still revealed that Bcl11b regulates cell cycle, stem cell maintenance, positive regulation of cell proliferation and Wnt signaling (Response Figure 4, new Fig. 4I-J).

Figure for reviewers removed

Minor concerns that should be addressed:

Fig 1A: ERN, or ENR? Label figure and legend consistently. Please define ENR and ERB in results or legend of Fig 1A. Neither is defined in the manuscript. I'm guessing the ERB has BMP in it? Fig 1A should have a broken scale so we can see Bcl11b more clearly.

Response: Sorry for the confusion. We have labeled them as ENR in the new Figure 1A. As described in the figure legend of Figure 1A, B is the abbreviation of BMP4. Also, we have labeled all abbreviations of culture medium more clearly in figure legends. In addition, we change the y axis as you suggested (Response Figure 5, new Figure 1A).

Figure for reviewers removed

Fig 1H: There is no description of the ChIP-qPCR antibodies or primers or methods in the manuscript.

Response: ChIP-qPCR antibodies were the same as those used in the ChIP-seq. For Smad1 and Smad4 ChIP, the experiments were performed as previously described (Qi *et al.*, 2017). The primers were listed in Table S1 as mentioned in the qRT-PCR method. For clarification, we have described these in the ChIP part again.

The Flag-Bcl11b-expressing lentivirus is not described adequately. I don't see how it is dox regulated. What is the transduction efficiency? Is there selection? Please expand.

Response: Sorry for missing the information. We have made a detailed description about lentivirus in page 18:

Virus production and organoid infection. Lentivirus was used for overexpressing Bcl11b in the organoids. Firstly, we modified pLVX-TRE3G vector (Clontech, 631193). Puromycin resistance gene was replaced with the blasticidin S resistance gene, named pLVX-TRE3G-bsd. Mouse Bcl11b cDNAs were constructed into the pLVX-TRE3G-bsd vector fused with Flag tag.

Lentivirus was produced in HEK293FT, which were cultured in Dulbecco's

modified Eagle's medium (DMEM, Invitrogen, 31053028) supplemented with 10% fetal bovine serum (FBS, Hyclone, SH20074.03) under the selection with G418 (500 µg/mL, Gibco, 11811). The 10cm-dish cells were transfected with 4.3 µg of the plasmids pLVX-TRE3G-puro expressing Bcl11b, 4.3 µg Vsvg and 6.43 µg Δ8.9, using Polyetherimide (PEI, Polysciences, PT-101-01N), and cultured with G418-free medium. Three days post-transfection, the supernatant was passed through a 0.45-µm filter, added lenti-Concentin Virus Precipitation Solution (ExCell Biology, EMB810A-1S) and refrigerated at 4 °C overnight. After 3000g centrifugation at 4 °C for 30 min, retroviral pellet was re-suspended in 250 µL of infection medium (the ENR medium containing 6.67 µM blebbistatin, 2.5 µM CHIR-99021, 10% Wnt3a conditional medium and 10 µg/mL polybrene (Macgene, MC032)).

Before virus infection, organoids, derived from *R26-M2rtTA; Vil-CreERT2* mice, were cultured with the expansion medium plus 6.67 µM blebbistatin, 2.5 µM CHIR-99021 and 10 mM nicotinamide for 2 days. Then, the organoids were digested with TrypLE (Gibco, 12604021) for 5 min at 37 °C and re-suspended with infection medium containing virus. Add 10 µL Matrigel into each well of 48-well plate and incubate at 37 °C for 10 min to solidify Matrigel. Add 250 µL of infection medium containing cells and virus on the solidified Matrigel and incubate overnight at 37 °C. Next day, remove the infect medium and wash the virus with warm PBS. Then, overlay 10 µL Matrigel and culture the organoids with the ENR medium containing 6.67µM blebbistatin, 2.5 µM CHIR-99021 and 10% Wnt3a conditional medium. Two days after infection, change the medium with ENR plus 2.5 µg/mL blasticidin S (Selleck, S7419). At 2-3 days post infection, protein expression was induced with 10 µM doxycycline (Selleck, S4163).

In figure 2 it shows acute KO of Bcl11b causes a marked decrease in stem cell markers and stemness. In contrast, in fig 4D, acute KO of Bcl11b has no effect on the number of Ki67+ cells. Please address these apparently contradictory results.

Response: Many studies have demonstrated that impacts on Lgr5+ cells do not necessarily affect Ki67+ cells (Baghdadi *et al*, 2022; Koppens *et al*, 2021; Li *et al*, 2020; Sato *et al*, 2020). Our study further showed that the stemness and proliferation programs can be uncoupled (Liu *et al*, 2023b). Similarly in this study, we found that Bcl11b KO impaired Lgr5+ ISC stemness without dramatically affecting cell proliferation (Figure S1C and Figure 4D). We have added this in the Discussion (page 15).

Please comment on this prior report: Bcl11b SWI/SNF-complex subunit modulates intestinal adenoma and regeneration after γ-irradiation through Wnt/β-catenin pathway <https://academic.oup.com/carcin/article/36/6/622/276606>

Response: In the mentioned paper, Sakamaki et al. reported that Bcl11b represses intestinal epithelium regeneration and adenoma formation through inhibiting the Wnt/β-catenin pathway. The discrepancy may be owing to different mice used in the studies: conventional Bcl11b^{+/-} mice were mainly utilized in their study, while we employed inducible intestinal epithelium-specific knockout mice. They observed conventional *Bcl11b*^{KO/+}; *Apc*^{min/+} mice developed more adenoma than the *Apc*^{min/+} mice.

Given that Bcl11b is critical for T cell development, the observed effect of Bcl11b heterozygosity on tumor formation may possibly attribute to T cell deficiency. These authors reported that some of the Wnt target genes, including Myc, Ccnd1 and Jun, were elevated in Lgr5+ cells from Lgr5-CreERT2;Bcl11b^{f/f} mice two months after 4-OHT induced Bcl11b knockout. However, more Wnt target genes including Axin2, Sp5, Lef1, Mmp7, Tnfrsf19 decreased after Bcl11b knockout, which is consistent with our data. We have discussed the mentioned work in the Discussion (page 14).

The authors do LC-MS/MS of Flag-Bcl11b IP from organoids. This seems quite challenging with tiny amounts of protein. Please report the number of organoids, how they were harvested, lysed, and the protein yield prior to IP.

Response: As you mentioned, it was indeed challenging to perform LC-MS/MS with tiny amounts of protein. Organoids derived from the crypts of R26-M2rtTA;Vil-CreERT2 mice were infected with Bcl11b-expressing lentivirus. After blasticidin S selection, the organoids were treated with doxycycline for 3 days. All organoids from an entire 24-well plate were used as one sample and harvested with PBS. Then, the organoids precipitation was lysed in 300 μ L ice-cold lysis buffer (10 mM Tris-HCl, pH 7.5, 100 mM NaCl, 0.5% NP-40, 1 mM EDTA, 1mM PMSF, 50 mM NaF, 2 mM Na₃VO₄, protease inhibitor cocktails (04693132001, Roche)) for 1 hr. After centrifugation at 13,000 rpm for 10 min, 30 μ L supernatant was collected for preclearing, and the rest of the supernatant was immunoprecipitated with anti-Flag antibody-binding Dynabeads with rotation overnight at 4 °C, and subjected to LC-MS/MS analysis. We have added these details in the Methods (page 22).

The analysis of the MS data is not well described. What filtering for SEQUEST quality, number of peptides, adjusted P value was used? Neither of the MS datasets find β -catenin nor TCF7. Why is this?

Response: Sorry for missing the information. We have made a detailed description about LC-MS/MS protein identification in page 23:

LC-MS/MS protein identification. Enzymatic digestion and LC-MS/MS analysis were performed in the Protein Chemistry and Proteomics Facility at Tsinghua University. For SDS-PAGE samples, the gels were washed with solution (25mM NH₄HCO₃, 50% chloroacrylonitrile) in 37°C and washed twice with acetonitrile, and then dehydrate the gel through vacuum centrifuge concentrator. For beads-pulldown samples, the immunoprecipitants-beads conjugations were washed five times with PBST and five times with 50 mM NH₄HCO₃. Both gel sample and beads samples were reduced by Tris(2-carboxyethyl)phosphine and alkylated by chloroacetamide, then digested with trypsin for overnight and terminated with 1 μ L of 10% trifluoroacetic acid. After centrifuged with a low speed, the supernatant were removed into a new tube. Then, add extraction solution (50% acetonitrile, 0.1% formic acid) into the colloidal particles to extract twice. Combine the supernatant and extraction solution, and concentrate it to 10 μ L volume. Add 20 μ L 0.1% formic acid to re-dissolve and prepare for mass spectrometer. The resulting tryptic peptides were analyzed using an UltiMate™ 3000 RSLCnano system (Thermo Fisher Scientific), directly interfaced

with an Q Exactive HF-X mass spectrometer. The MS/MS data were searched against the selected database using the SEQUEST search engine in Proteome Discoverer 2.3 software. The search criteria were as follows: full tryptic specificity was required; two missed cleavages were allowed; carbamidomethylation (C) were set as the fixed modifications; oxidation (M) and acetylation (protein N terminal) were set as the variable modification; precursor ion mass tolerances were set at 10 ppm for all MS acquired in an Orbitrap mass analyzer, and the fragment ion mass tolerance was set at 20 mmu for all MS2 spectra acquired. During the database search, false discovery rate (FDR) was set to 0.01 for peptides filter. Protein FDR confidence includes high (FDR<0.01), medium (FDR<0.05) and low (FDR>0.05), respectively. Besides, Exp. q-value, also named q-value, was used to restrict the best score threshold. q-values are less than 0.01 for high confidence hits and 0.05 for medium confidence hits. Following proteome discoverer (PD), search results were assigned to two groups (IgG and IP), and IgG served as the negative control. To determine the protein abundance of IP compared with IgG, abundance was utilized for candidate filter and determined by peak area, and relative protein quantification was calculated as the median of all possible pairwise peptides ratios in the IP sample by the IgG sample. The proteins were filtered for abundance ratio >2 and peptide number >1 (new table S4). The top 27 proteins with ratio IP/IgG >2 were used for STRING (cn.string-db.org) or GO analysis.

Actually, β -catenin was found in the Bcl11b-interacting proteins using the Protein A-Sepharose 4B conjugate, although without the abundant ratio (We highlighted this protein in the full list of new table S4 (line 505)). Nonetheless, the interaction between Bcl11b and β -catenin was detected by immunoblotting and *in vitro* by SPR assay using purified proteins (Fig. 4F-G). In addition, the endogenous β -catenin-Tcf4 interaction was decreased in Bcl11b KO crypts, also supporting the promoting role of Bcl11b in Wnt signaling (figure 4H). Functionally, Bcl11b KO decreased crypt length induced by adenovirus-mediated expression of Rspo-1 (Fig 4D).

Referee #2:

General summary and opinion: This study by Li et al is the description of a comprehensive data set that shows that Bcl11b, a gene that is downregulated upon BMP pathway activation (for example upon treatment with endogenous BMP ligands), act as a transcriptional regulator of WNT target genes. It is nicely shown that Bcl11b-deficient cells have lower expression of WNT target genes, whereas Bcl11b-overexpressed cells have higher Wnt target gene expression. Mechanistically, the authors on the one hand propose a direct interaction of Bcl11b with WNT components B-Cat/TCF4 for Wnt pathway specific aspects, on the other hand suggest a more general transcriptional regulation of Bcl11b, through modulating of the NuRD complex. Furthermore, the mechanistic work is supported by sturdy in vivo data showing that Bcl11b is important for reparative mechanisms in the intestine (radiation of small intestine, and using DSS-colitis in the colon), as well as potentially relevant for tumorigenesis. Indeed showing an important physiological role for Bcl11b in the intestinal stem cell system. Overall, the study is comprehensive and technically sound, and sufficient replicates etc. was used. The data is also presented clearly.

Response: Thank you for your positive comments.

Specific major concerns:

- BMP regulating Bcl11b expression: The strongest evidence that this is a direct regulatory mechanism actually only comes from Fig. 1H (Smad occupancy at gene). All the data before just shows that Bcl11b behaves similar to 'stem cell' genes, and the changes (increase/decrease) can be a consequence of a change in # of stem cells (like in the Alk3 KO (similar to their previous study), that will get an expansion of the stem cell zone). The authors could do a short term (1/4/8h or so) BMP4 treatment to show rapid repression (preferably prior to changes in Wnt genes, as this would be a consequence of the changed Bcl11b expression such as the authors propose), which would support direct target gene hypothesis. As is, there's not sufficient data to support that Bcl11b is a bona fide BMP-repressed gene.

Fig. 1H is further a bit confusing as this is ChIP data from sorted Lgr5 cells (which should not have active BMP signaling, nor is this a cell where Bcl11b is repressed (see rest of the figure!)). It would be helpful to have Smad1/4 negative genes as a control qPCR to show that this is specific for the site.

Response: Thank you for your great suggestion We will perform the BMP4 time course as you suggested to see whether Bcl11b expression would decrease earlier than Wnt target genes.

The ChIP data of Smad1 and Smad4 were from both Lgr5 high and low cells, not just Lgr5 high cells. As Bcl11b is absent in Lgr5 negative cells, they were not used for this assay. The detailed method was added in the text (page 28). In brief, the organoids derived from Lgr5-EGFP-IRES-creERT2 mice were cultured with ER medium for 4 hrs. Then, 1,000,000 Lgr5⁺ ISCs were sorted from the organoids for each sample. We agree that BMP signaling is very low in Lgr5 high cells, but it is active in Lgr5 low cells where Bcl11b is repressed. To confirm the specific binding, we have included the

Smad1/4-unbound region as a negative control, as you suggested (Response Figure 6, new Fig. 1H).

Figure for reviewers removed

- Fig 2 and throughout (also the methods). The Lgr5 mouse line is a mosaic line where only ~30-40% expresses GFP. In the methods it is described that random parts were assessed for GFP levels during microscopy, however, this could lead to randomly determining positive vs negative GFP sites. Fig. 2A shows complete absence, but this is perhaps more likely due to negative patch (as the qPCR shows just a 50% reduction in 2B).

Response: As the reviewer pointed out, indeed only partial ISCs can be labeled with GFP in this line. Therefore, beside the image data in Figure 2A, we also used FACS to quantify the Lgr5+ cells. The data in Figure S1D showed that Lgr5+ cells decreased from 7% to 2%. As the intensity of GFP was also decreased, this decrease may result in the phenomenon that GFP+ cells seemed to disappear completely in the images. Also, qPCR is very sensitive and could detect a very low level of Lgr5 mRNA in a cell in Figure 2B. Therefore, we measured the Lgr5 expression from various angles, and all the results indicated that Bcl11b depletion resulted in ISC decrease.

Related to this figure, the authors suggest no changes were found in Paneth cells. However, changes in PCs take longer (4-6 weeks after deletion) to be observed. Original Sato Paneth cell 'niche' paper (Sato et al 2011, Nature) shows deletion of certain genes takes 6 weeks to observe reduction of Paneth cells. I recommend deleting Bcl11b and look 4-6 weeks later.

Response: Thank you for pointing this out, and we will measure Paneth cell number in a long-term time point as you suggested.

- Specificity of Bcl11b in regulating Wnt: ChIP-seq identified ~7000 genes in ISCs to be under the control of Bcl11b. To me, this is basically half if not more of all genes expressed in ISCs (genome = 20k genes, ~12-15k expressed in specific cell types). As ISCs are high Wnt condition, it's likely to then find strong overlap. To make this specific, the authors should pick something that's also high in ISCs (such as Notch-Hes1 and its

target genes), and show that Bcl11b does not regulate those genes (or similar line of reasoning). To me, a more plausible conclusion is that Bcl11b is just important for gene expression, facilitating whatever signaling takes place (high Wnt in ISCs in this case). Don't get me wrong, I really like the paper, I just find that this specificity is in contrast to a gene that seems to control gene expression so broadly. Furthermore, such 'aspecific' hypothesis also fits better with the proteomics data (that didn't identify Bcat/TCF4, but found more broad partners in the nucleus), rather than the OE interaction data with B-cat (again a negative other relevant TF would be good as a negative control).

Response: Thank you for the insightful comments. We have re-analyzed our ChIP-seq data with more restricted parameters and identified 1863 Bcl11b binding sites in 1729 genes, a number similar to the Bcl11b targets reported in breast cancer cells by Cai's group (Bai *et al.*, 2022).

Actually, β -catenin was found in the Bcl11b-interacting proteins using the Protein A-Sepharose 4B conjugate, although without the abundant ratio (We highlighted this protein in the full list of new table S4 (line 505)). Nonetheless, the interaction between Bcl11b and β -catenin was detected by immunoblotting and *in vitro* by SPR assay using purified proteins (Fig. 4F-G). In addition, the endogenous β -catenin-Tcf4 interaction was decreased in Bcl11b KO crypts, also supporting the promoting role of Bcl11b in Wnt signaling (figure 4H). Functionally, Bcl11b KO decreased crypt length induced by adenovirus-mediated expression of Rspo-1 (Fig 4D). Together with those data from intestinal epithelium-specific Bcl11b KO mouse model, these results indicate that Bcl11b promotes Wnt signaling. We will add other relevant TFs as negative controls, such as Yap or Hes1, as you suggested.

Of note, my first 2 points are important to address experimentally, or verify method-technically. My 3rd point is definitely open to interpretation, but feel should be addressed as the authors currently claim a Wnt-specific role and this is not fully supported by the data with a lack of negative controls. The lack of BMP/TGFb/TNF reporter line activity is not sufficient, rather a Yap or Hes1 reporter would indeed be more relevant to ISC signaling state.

Response: Thank for your suggestion. We will examine Bcl11b's effect on the Hippo and Notch reporters.

Minor concerns:

- Please provide high magnification of Bcl11b staining (Fig. 1G) to see stem-cell / Paneth cell zone.

Response: Per your suggestion, it has been shown (Response Figure 7, new Fig. 1G).

Figure for reviewers removed

- Throughout: please when using statistical analysis of low n, one cannot assume normal distribution (so use non-parametric tests)

Response: Thanks for your comments. We have carefully re-analyzed our data. Parametric tests were chosen when data was normally distributed based on Shapiro-Wilk test. Normally distributed data were analyzed using parametric t-test with Welch's correction, one-way ANOVA with Tukey's multiple comparisons test, two-way ANOVA with Tukey's multiple comparisons test or Sidak's multiple comparisons test. Specific tests and significance levels are indicated in corresponding figure legends.

- Please use a mouse = 1n, example: Fig. 2A has many data points, but I don't think these are different mice (just individual crypts). Same for Fig. 2C/2G/2I, n= a well not 1 organoid, and preferably from different mice.

Response: Thanks for your suggestions. We have included how many fields/crypts/organoids were analyzed from three biological repeats in corresponding figure legends.

- Ensure X-ray was used (normally it's gamma-radiation machines that is used in patients for therapy, not the X-ray to image). That said, x-ray is gamma-radiation type so if tuneable this could have been used.

Response: We used a Rs2000 irradiator (Radsource) to do the X-ray irradiation in Laboratory Animal Resources Center of Tsinghua University. The X-ray from Rs2000 irradiator is equal to the gamma-Ray from Cs137. We have added the information in the Methods section.

- Wording: please make sure to state when using ectopic overexpression of a plasmid to determine its role, or when looking at interactions (IP). Compared to looking at endogenous levels. Similar; OE and tagged IP experiments certainly are proof that proteins can interact, but also not proof that they are in a complex under normal conditions or in ISCs.

Response: We have checked and changed these expressions in the text to ensure the accuracy.

Referee #3:

The manuscript by Li and team explores the role of Bcl11b in intestinal stem cells. In figure 1, the authors use multiple lines of evidence to demonstrate that Bcl11b is repressed by BMP signaling and enriched in stem and progenitor cells in the crypt. In figure 2, Bcl11b is shown to be required for a full number of Lgr5+ ISCs both in vivo and in organoid models. Gain and loss of function systems show that Bcl11b expression correlates with ISC marker expression. In figure 3, irradiation and DSS models are used to show that Bcl11b is important for the intestine to resist damage from these treatments. In figure 4, cell culture assays and co-IP experiments point to interactions between Bcl11b and TCF and Bcatenin. ChIPseq analysis shows overlap between Bcl11b and TCF4 and Bcatenin binding sites, further supporting a role in regulating the WNT signaling pathway. In figure 5, the authors dig deeper into how Bcl11b may be regulating genes and find that BCL11B interacts with components of the NURD complex, and that loss of BCL11B leads to changes in ATAC and H3K27ac and H3K27me3 that correspond to reduced WNT target gene expression in the BCL11B mutant. Finally, figure 6 provides evidence that BCL11B supports cancer growth, as might be expected based upon its support of the WNT pathway.

This work is generally well-executed and provides novel insights into how the WNT pathway is regulated in normal intestinal homeostasis and in colon cancer. The authors frequently use both gain and loss of function genetic approaches and state-of-the-art techniques to make their discoveries. While there are some important concerns to be addressed to improve the rigor and readability of the work, the work will certainly make exciting contributions to the field.

Response: Thank you for your positive comments.

Concerns:

The data haven't been made available to the reviewers and the number of replicates for many experiments is unclear. (for example, ChIPseq)

Response: Sorry for missing the information. We have now included it to allow reviewers' examination:

To review GEO accession GSE255138: Go to <https://www.ncbi.nlm.nih.gov/geo/query/acc.cgi?acc=GSE255138>. Enter token wpyhgayulvitfcp into the box.

Please indicate where the datasets arise from in the figures and figure legends. For example, figure 1C was from a previous publication, but it's not so easy to find the data source. It would help readers to have a more direct connection to the data.

Response: Sorry about it. We indicated the data source in the data availability statement. To make it clearer, we have indicated the source in the legend now. Fig 1C is from GSE186917, Fig 1B from GSE247044, Fig 6A and S7A-B from GSE178341.

Please address this paper that finds opposite results in a mouse model of colon cancer: <https://pubmed.ncbi.nlm.nih.gov/25827435/>

Response: In the mentioned paper, Sakamaki et al. reported that Bcl11b represses intestinal epithelium regeneration and adenoma formation through inhibiting the Wnt/ β -catenin pathway. The discrepancy may be owing to different mice used in the studies: conventional Bcl11b^{+/-} mice were mainly utilized in their study, while we employed inducible intestinal epithelium-specific knockout mice. They observed conventional *Bcl11b*^{KO/+}; *Apc*^{min/+} mice developed more adenoma than the *Apc*^{min/+} mice. Given that Bcl11b is critical for T cell development, the observed effect of Bcl11b heterozygosity on tumor formation may possibly attribute to T cell deficiency. These authors reported that some of the Wnt target genes, including Myc, Ccnd1 and Jun, were elevated in Lgr5⁺ cells from Lgr5-CreERT2;Bcl11b^{fl/fl} mice two months after 4-OHT induced Bcl11b knockout. However, more Wnt target genes including Axin2, Sp5, Lef1, Mmp7, Tnfrsf19 decreased after Bcl11b knockout, which is consistent with our data. We have discussed the mentioned work in the Discussion (page 14).

Cre-only controls should be used in the regeneration assays to test for off-target effects. It is not clear if these controls are in place.

Response: Thank you for pointing it out. Indeed, all control mice we used were referred to Vil-CreERT2;Lgr5-GFP-IRES-creERT2 mice that were injected with the same amount of tamoxifen to avoid off-target and possible toxicity-induced phenotypes.

It does not appear that there is a description of the overexpression construct. This is one example of several in the methods section that appears to lack key experimental details and should be improved so that others can try and reproduce the work.

Response: Thanks for suggestions. We have now added all the experimental details (pages 18, 22, 23 and 28). In particular, the Flag-Bcl11b-expressing lentivirus was described in page 18.

Reference

- Baghdadi MB, Ayyaz A, Coquenlorge S, Chu B, Kumar S, Streutker C, Wrana JL, Kim TH (2022) Enteric glial cell heterogeneity regulates intestinal stem cell niches. *Cell Stem Cell* 29: 86-100 e106
- Bai H, Lin M, Meng Y, Bai H, Cai S (2022) An improved CUT&RUN method for regulation network reconstruction of low abundance transcription factor. *Cell Signal* 96: 110361
- Koppens MAJ, Davis H, Valbuena GN, Mulholland EJ, Nasreddin N, Colombe M, Antanaviciute A, Biswas S, Friedrich M, Lee L *et al* (2021) Bone Morphogenetic Protein Pathway Antagonism by Grem1 Regulates Epithelial Cell Fate in Intestinal Regeneration. *Gastroenterology* 161: 239-254 e239
- Li Q, Sun Y, Jarugumilli GK, Liu S, Dang K, Cotton JL, Xiol J, Chan PY, DeRan M, Ma L *et al* (2020) Lats1/2 Sustain Intestinal Stem Cells and Wnt Activation through TEAD-Dependent and Independent Transcription. *Cell Stem Cell* 26: 675-692 e678
- Li Y, Liu Y, Chiang YJ, Huang F, Li Y, Li X, Ning Y, Zhang W, Deng H, Chen YG (2019) DNA Damage Activates TGF-beta Signaling via ATM-c-Cbl-Mediated Stabilization of the Type II Receptor TbetaRII. *Cell Rep* 28: 735-745 e734
- Li Y, Liu Y, Liu B, Wang J, Wei S, Qi Z, Wang S, Fu W, Chen YG (2018) A growth factor-free culture system underscores the coordination between Wnt and BMP signaling in Lgr5(+) intestinal stem cell maintenance. *Cell Discov* 4: 49
- Liu L, Wang Y, Yu S, Liu H, Li Y, Hua S, Chen YG (2023a) Transforming Growth Factor Beta Promotes Inflammation and Tumorigenesis in Smad4-Deficient Intestinal Epithelium in a YAP-Dependent Manner. *Adv Sci (Weinh)* 10: e2300708
- Liu Y, Chen YG (2020) Intestinal epithelial plasticity and regeneration via cell dedifferentiation. *Cell Regen* 9: 14
- Liu Y, Huang M, Wang X, Liu Z, Li S, Chen YG (2023b) Segregation of the stemness program from the proliferation program in intestinal stem cells. *Stem Cell Reports* 18: 1196-1210
- Qi Z, Chen YG (2015) Regulation of intestinal stem cell fate specification. *Sci China Life Sci* 58: 570-578
- Qi Z, Li Y, Zhao B, Xu C, Liu Y, Li H, Zhang B, Wang X, Yang X, Xie W *et al* (2017) BMP restricts stemness of intestinal Lgr5+ stem cells by directly suppressing their signature genes. *Nat Commun* 8: 13824
- Sato T, Ishikawa S, Asano J, Yamamoto H, Fujii M, Sato T, Yamamoto K, Kitagaki K, Akashi T, Okamoto R *et al* (2020) Regulated IFN signalling preserves the stemness of intestinal stem cells by restricting differentiation into secretory-cell lineages. *Nat Cell Biol* 22: 919-926
- Wang S, Chen YG (2018) BMP signaling in homeostasis, transformation and inflammatory response of intestinal epithelium. *Sci China Life Sci* 61: 800-807
- Wang Y, Huang M, Mu X, Song W, Guo Q, Zhang M, Liu Y, Chen YG, Ge L (2024) TMED10-mediated unconventional secretion of IL-33 regulates intestinal epithelium differentiation and homeostasis. *Cell Res* 34: 258-261
- Zhao B, Qi Z, Li Y, Wang C, Fu W, Chen YG (2015) The non-muscle-myosin-II heavy

chain Myh9 mediates colitis-induced epithelium injury by restricting Lgr5+ stem cells. *Nat Commun* 6: 7166

Dear Ye-Guang,

Thank you for contacting me with a preliminary revision plan for your manuscript. I find your responses generally reasonable, and I have therefore further discussed your plan with the original reviewer #1. I am glad to say that he/she found your response convincing and indicated that it would address most of their main concerns.

Therefore, I would like to invite you to submit a revised manuscript in response to the reviewers' comments. Please note that we will ultimately require strong support from the reviewers for publication here.

We generally allow three months as standard revision time. As a matter of policy, competing manuscripts published during this period will not negatively impact on our assessment of the conceptual advance presented by your study. However, please contact me as soon as possible upon publication of any related work to discuss the appropriate course of action. Should you foresee a problem in meeting this three-month deadline, please let us know in advance to arrange an extension.

When preparing your letter of response to the referees' comments, please bear in mind that this will form part of the Review Process File and will therefore be available online to the community. For more details on our Transparent Editorial Process, please visit our website: <https://www.embopress.org/page/journal/14602075/authorguide#transparentprocess>. Please also see the attached instructions for further guidelines on preparation of the revised manuscript.

Please feel free to contact me if you have any further questions regarding the revision. Thank you for the opportunity to consider your work for publication. I look forward to receiving the revised manuscript.

With best regards,

Ieva

We realize that it is difficult to revise to a specific deadline. In the interest of protecting the conceptual advance provided by the work, we recommend a revision within 3 months (5th Sep 2024). Please discuss the revision progress ahead of this time with the editor if you require more time to complete the revisions.

Summary of Figure Changes in the Revised Manuscript

Revised version	Previous version	Description
Figure 1A	Figure 1A	Replace the ERN label with ENR Change y axis with a broken scale
Figure 1G	Figure 1G	Add high magnification of Bcl11b staining
Figure 1H	Figure 1H	Add Smad1/4 negative region as negative control
Figure 2C-E	Figure 2C-E, S1E	Add two control groups; combine previous S1E with 2D Re-analyze the significant test
Figure 4B	Figure 4B	Correct the figure labeling
Figure 4I-J	Figure 4I-J	Replaced with the re-analyzed data using more restricted parameters
Figure 5A	Figure 5A	Replaced with the new data
Figure 5B-F	Figure 5C-G	Number change
Figure 6C	Figure 6C	Re-analyze the significant test
Figure S1B		New data: Change of Bcl11b and Id1 expression upon short-term BMP treatment
Figure S1D	Figure S1C	Replaced with LYZ staining of small intestinal sections from Control and Bcl11b cKO mice 4 weeks after Tamoxifen injection
Figure S1G	Figure S1G	Add two control groups
Figure S1C, S1E-F	Figure S1B, S1D, S1F	Number change
Figure S3D-E		New data: The effect of Bcl11b on Notch and Hippo signaling examined by reporter assay
Figure S3F-K	Figure S3D-I	Number change
Figure S4A		New data: Immunoprecipitation showing Bcl11b could not interact with YAP or HES1 as negative control
Figure S4B-D	Figure S4A-C	Number change
Figure S4F		New data: IGV tracks displaying CHIP-seq reads along the Notch target genes Hey1 and Heyl in the intestinal organoids
Figure S5A	Figure S5A	Replaced with the new data
Table S1	Table S1	Add the primer seq of Smad1/4 negative region
Table S2	Table S2	Replaced with the re-analyzed data using more restricted parameters
Table S4	Table S4	Replaced with the new LC-MS/MS data

Referee #1:

Li et al. report that Bcl11b, a BMP-SMAD repressed gene, plays a role in the intestinal stem cell niche and they propose it acts to repress the NuRD complex that inhibits gene expression. Acute knockout of Bcl11b decreases stemness and expression of intestinal stem cell genes, although these acute knockout experiments may have a seriously flawed control. Acute knockout of Bcl11b impairs recovery of the gut from radiation and DSS injury. Forced over-expression of Bcl11b enhances TOPFLASH, a plasmid based reporter assay. Addressing mechanism, they find that over-expressed Bcl11b pulls down β -catenin and interact with β -catenin in SPR. Confusingly, Bcl11b IP does not find β -catenin nor TCF4 in a MS study. Over-expressed Bcl11b binds to >6000 genes, i.e., a third of the genes, so it is hard to know what to make of that. It dilutes the claim that Bcl11b specifically interacts with TCF4/ β -catenin. IP of over-expressed Bcl11b confirms previous reports of interaction with the NuRD complex components, and KO of Bcl11b decreases chromatin accessibility in half the genome as far as I can tell, by ATAC-seq.

The evidence that Bcl11b is specifically involved in colon cancer is circumstantial; the knockout slows growth, but it's not clear this is specific to APC-mutant cancers.

Overall, the authors present a very large amount of data that suggest a role for Bcl11b in regulation of the NuRB complex, uneven data that Bcl11b directly interacts with Tcf4 and β -catenin, and mouse data for an important role for Bcl11b in intestinal stem cells, but with a caveat that a key control may be absent.

Response: Thank you for insightful comments. As addressed below, we have clarified all of your concerns by adding more data, re-calculating the sequencing data, making clear labeling, etc. We have also modified the manuscript accordingly. We hope these improvements would address your concerns.

Major Issues:

Acute activation of Cre causes DNA damage and stem cell toxicity, and it is not at all clear this was controlled for in their mouse studies. This is a well-established problem. See "Genome Toxicity and Impaired Stem Cell Function after Conditional Activation of CreERT2 in the Intestine", PMID: 30449703. The authors must clearly state the genotype of the controls in Fig 2E and 2F, and Fig 3D-G, Fig 4C-D. The authors must test if the results in these experiments are due to Cre toxicity versus Bcl11b excision.

Response: As we have used mouse genetic manipulations to study intestinal stem cells for many years and published more than a dozen of papers (Li et al, 2019; Li et al, 2018; Liu et al, 2023a; Liu & Chen, 2020; Qi & Chen, 2015; Qi et al, 2017; Wang & Chen, 2018; Wang et al, 2024; Zhao et al, 2015), we were aware of the possible stem cell toxicity induced by Cre activation. For this reason, we usually injected mice with the same amount of tamoxifen as the control to eliminate any non-specific effects of tamoxifen or Cre activation. The detailed genotype was clearly stated in the figure

legend of Figure 2A. Since the control genotypes were the same throughout the entire manuscript, including Fig 2E-F, Fig 3D-G and Fig 4C-D, we did not repeatedly mention this each time. Meanwhile, we have also addressed this point in the Methods section.

To enhance clarity, we have included the description of the control genotype in the revised manuscript (page 6, lines 117-119) and legends for each figure panel, as per your suggestion.

Fig 2E: I expected four conditions - control and *Bcl11b* ko, +/- tamoxifen. The data presented do not show any control for tamoxifen effect on stem cells, which is a known problem (pmid: 23415913).

Response: We appreciate the reviewer for suggestions. Indeed, we have performed many controls, just as the reviewer suggested. In all the organoid experiments, we used organoids from *Vil-CreERT2;Lgr5-GFP-IRES-creERT2* (control) mice treated with the same amount of 4-hydroxytamoxifen (4-OHT) to avoid the toxicity of tamoxifen on stem cells. As shown in Response Figure 1 (new Figure 2C-E and S1G), we have implemented the controls as suggested by the reviewer. To maintain conciseness and emphasize key points, we just omitted the EtOH (ethanol) group from the previous submitted manuscript and focused on the 4-OHT group to assess its effect on stem cells. We have now modified the new figure 2C-E and S1G and included all the EtOH controls.

Figure for reviewers removed

Vil-CreERT2;Lgr5-GFP-IRES-creERT2;Bcl11b^{fl/fl} (Bcl11b cKO) mice were treated with EtOH or 4-OHT for 48h and then fixed for analysis of GFP expression 48h later. Left panel shows the representative images, middle panel shows the representative FACS results, right panel shows the statistics result of GFP⁺ cells through FACS. Nuclei were counter-stained with DAPI. Scale bar, 100 μ m. Each dot represents one experiment. n= three independent experiments. (C) The Lgr5⁺ ISCs of *Vil-CreERT2;Lgr5-GFP-IRES-creERT2* (control) and *Vil-CreERT2;Lgr5-GFP-IRES-creERT2;Bcl11b^{fl/fl}* (Bcl11b cKO) mice were sorted by FACS. About 5000 cells were seeded in each well for single cell culture upon EtOH or 4-OHT treatment for two days. 6 days later, the images were taken, and organoids were counted. Left panel shows the representative images, middle panel shows the average number of organoids of three independent experiments, right panel shows the expression of *Bcl11b* by qRT-PCR. Each dot represents one experiment. Scale bar, 100 μ m. The data were analyzed by Two-way ANOVA with Tukey's multiple comparison test. *P<0.05, **P<0.01 and ***P<0.001.

The fold activation of TOPFLASH in HEK293 cells is surprisingly large, yet the reporter is plasmid based. Very few genes have this kind of effect on Wnt signaling. And then, HCT116 cells have mutant active β -catenin already, but expression of Bcl11b alone has a minimal effect. This is confusing and not consistent with the simple model. Is NuRD important for regulating simple promoters on plasmids? What happens to endogenous Wnt target genes in these cells?

Response: We found that Bcl11b, but not other BMP target genes, significantly promoted Wnt reporter expression (Response Figure 2). Thanks to your suggestion, we realized that the TOPFLASH result of HCT116 was mis-labeled by messing up Wnt3a and Bcl11b in figure labeling. Sorry for the mistake. In Response Figure 3, Bcl11b over-expression greatly increased the Wnt reporter activity, similar to that in HEK293T (Figure 4A). As HCT116 cells harbor mutant active β -catenin, Wnt3a shows moderate enhancement. We have now corrected the error in new Figure 4B.

Figure for reviewers removed

n=three independent experiments.

Figure for reviewers removed

ChIP-seq performed with over-expressed FLAG-Bcl11b is going to be full of over-expression artifacts due to protein binding to irrelevant low-affinity promoters. This artifact is reflected in the over 6000 Bcl11b target genes (Figs 4I-K, S4D-E, tables S2). I do not find this helpful; the experiment should be repeated with endogenous expression levels of Bcl11b, or removed.

Response: Thanks for the insightful comments. We have tested various Bcl11b antibodies, but none of them were suitable for ChIP-seq assay, particularly with the limited organoid materials. The high number of Bcl11b target genes could be due to overexpressed Flag-Bcl11b in our experiment. We re-analyzed the ChIP-seq data with more restricted parameters by choosing the peaks with $\log_2(\text{fold enrichment}) > 3$ and $q \text{ value} < 0.01$ as the binding peaks of Bcl11b, and the remaining parameters were default. The HOMER (v4.10.0) `annotatePeaks.pl` was used to annotate the peaks. The reanalysis identified 1863 Bcl11b binding sites in 1729 genes, a number similar to the Bcl11b targets reported in breast cancer by Cai's group (Bai *et al*, 2022) (new table S2). GO analysis of these sites still revealed that Bcl11b regulates cell cycle, stem cell maintenance, positive regulation of cell proliferation and Wnt signaling (Response Figure 4, new Figure 4I-J).

Figure for reviewers removed

Minor concerns that should be addressed:

Fig 1A: ERN, or ENR? Label figure and legend consistently. Please define ENR and ERB in results or legend of Fig 1A. Neither is defined in the manuscript. I'm guessing the ERB has BMP in it? Fig 1A should have a broken scale so we can see Bcl11b more clearly.

Response: Sorry for the confusion. We have labeled them as ENR in the new Figure 1A. As described in the figure legend of Figure 1A, B is the abbreviation of BMP4. Also, we have clarified all abbreviations of culture medium in figure legends. In addition, we change the y axis as you suggested (Response Figure 5, new Figure 1A).

Figure for reviewers removed

Fig 1H: There is no description of the ChIP-qPCR antibodies or primers or methods in the manuscript.

Response: ChIP-qPCR antibodies were the same as those used in the ChIP-seq. For Smad1 and Smad4 ChIP, the experiments were performed as previously described (Qi *et al.*, 2017). The primers were listed in Table S1 as mentioned in the qRT-PCR method. For clarification, we have described these details in the revised Methods section (page 29, lines 625-634).

The Flag-Bcl11b-expressing lentivirus is not described adequately. I don't see how it is dox regulated. What is the transduction efficiency? Is there selection? Please expand.

Response: Sorry for missing the information. We have made a detailed description about lentivirus in the revised Methods (pages 18-19):

Virus production and organoid infection. Lentivirus was used for overexpressing Bcl11b in the organoids. Firstly, we modified pLVX-TRE3G vector (Clontech, 631193). Puromycin resistance gene was replaced with the blasticidin S resistance gene, named pLVX-TRE3G-bsd. Mouse Bcl11b cDNAs were constructed into the pLVX-TRE3G-bsd vector fused with Flag tag.

Lentivirus was produced in HEK293FT, which were cultured in Dulbecco's modified Eagle's medium (DMEM, Invitrogen, 31053028) supplemented with 10% fetal bovine serum (FBS, Hyclone, SH20074.03) under the selection with G418 (500 µg/mL, Gibco, 11811). The 10cm-dish cells were transfected with 4.3 µg of the plasmids pLVX-TRE3G-puro expressing Bcl11b, 4.3 µg Vsvg and 6.43 µg Δ8.9, using Polyetherimide (PEI, Polysciences, PT-101-01N), and cultured with G418-free medium. Three days post-transfection, the supernatant was passed through a 0.45-µm filter, added lenti-Concentin Virus Precipitation Solution (ExCell Biology, EMB810A-1S) and refrigerated at 4 °C overnight. After 1500g centrifugation at 4 °C for 30 min, lentiviral pellet was re-suspended in 250 µL of infection medium (the ENR medium containing 6.67 µM blebbistatin, 2.5 µM CHIR-99021, 10% Wnt3a conditional medium and 10 µg/mL polybrene (Macgene, MC032)).

Before virus infection, organoids, derived from *R26-M2rtTA;Vil-CreERT2* mice, were cultured with the expansion medium plus 10 mM nicotinamide for 2 days. Then, the organoids were digested with TrypLE (Gibco, 12604021) for 5 min at 37 °C and re-suspended with infection medium containing virus. Add 10 µL Matrigel into each well of 48-well plate and incubate at 37 °C for 10 min to solidify Matrigel. Add 250 µL of infection medium containing cells and virus on the solidified Matrigel and incubate overnight at 37 °C. Next day, remove the infection medium and wash the virus with warm PBS. Then, overlay 10 µL Matrigel and culture the organoids with the ENR medium containing 6.67µM blebbistatin, 2.5 µM CHIR-99021 and 10% Wnt3a conditional medium. Two days after infection, change the medium with ENR plus 2.5 µg/mL blasticidin S (Selleck, S7419). At 2-3 days post infection, protein expression was induced with 10 µM doxycycline (Selleck, S4163).

In figure 2 it shows acute KO of Bcl11b causes a marked decrease in stem cell markers and stemness. In contrast, in fig 4D, acute KO of Bcl11b has no effect on the number of Ki67+ cells. Please address these apparently contradictory results.

Response: Many studies have demonstrated that impacts on Lgr5⁺ cells do not necessarily affect Ki67⁺ cells (Baghdadi *et al*, 2022; Koppens *et al*, 2021; Li *et al*, 2020; Sato *et al*, 2020). Our study further showed that the stemness and proliferation programs can be uncoupled (Liu *et al*, 2023b). Similarly in this study, we found that Bcl11b KO impaired Lgr5⁺ ISC stemness without dramatically affecting cell proliferation (new Figure S1D and Figure 4D). We have added this in the Discussion (page 15, lines 315-320).

Please comment on this prior report: Bcl11b SWI/SNF-complex subunit modulates intestinal adenoma and regeneration after γ-irradiation through Wnt/β-catenin

pathway <https://academic.oup.com/carcin/article/36/6/622/276606>

Response: In the mentioned paper, Sakamaki et al. reported that Bcl11b represses intestinal epithelium regeneration and adenoma formation through inhibiting the Wnt/ β -catenin pathway. The discrepancy may be owing to different mice used in the studies: conventional *Bcl11b*^{+/-} mice were mainly utilized in their study, while we employed inducible intestinal epithelium-specific knockout mice. They observed conventional *Bcl11b*^{KO/+}; *Apc*^{min/+} mice developed more adenoma than the *Apc*^{min/+} mice. Given that Bcl11b is critical for T cell development, the observed effect of Bcl11b heterozygosity on tumor formation may possibly attribute to T cell deficiency. These authors reported that some of the Wnt target genes, including *Myc*, *Ccnd1* and *Jun*, were elevated in Lgr5⁺ cells of *Lgr5-CreERT2*; *Bcl11b*^{fl/fl} mice two months after 4-OHT induced *Bcl11b* knockout. However, more Wnt target genes including *Axin2*, *Sp5*, *Lef1*, *Mmp7*, *Tnfrsf19* were decreased after Bcl11b knockout, which is consistent with our data. We have discussed the mentioned work in the Discussion section (pages 14-15, lines 299-311).

The authors do LC-MS/MS of Flag-Bcl11b IP from organoids. This seems quite challenging with tiny amounts of protein. Please report the number of organoids, how they were harvested, lysed, and the protein yield prior to IP.

Response: As you mentioned, it was indeed challenging to perform LC-MS/MS with tiny amounts of protein. Organoids derived from the crypts of *R26-M2rtTA*; *Vil-CreERT2* mice were infected with Bcl11b-expressing lentivirus. After blasticidin S selection, the organoids were treated with doxycycline for 3 days. All organoids from two entire 24-well plates were used as one sample and harvested with PBS. Then, the organoids pellet was lysed in 300 μ L ice-cold lysis buffer (10 mM Tris-HCl, pH 7.5, 100 mM NaCl, 0.5% NP-40, 1 mM EDTA, 1mM PMSF, 50 mM NaF, 2 mM Na₃VO₄, protease inhibitor cocktails (Roche, 04693132001)) for 1 hr. After centrifugation at 13,000 rpm for 10 min, 30 μ L supernatant was collected for input, and the rest of the supernatant was immunoprecipitated with IgG or anti-Flag antibody-binding Dynabeads with rotation overnight at 4 °C, and subjected to LC-MS/MS analysis. We have added these details in the Methods (pages 23-26).

The analysis of the MS data is not well described. What filtering for SEQUEST quality, number of peptides, adjusted P value was used? Neither of the MS datasets find β -catenin nor TCF7. Why is this?

Response: To enhance the LC-MS/MS sensitivity, we increased the number of Bcl11b-overexpressed organoids from one 24-well plate to two entire 24-well plates per one sample; the immunoprecipitants-beads conjugations were washed twice with 50mM NH₄HCO₃ instead of five times with H₂O; peptide supernatants were analyzed using a Vanquish Neo UHPLC system (Thermo Fisher Scientific), directly interfaced with an Orbitrap Astral mass spectrometer; all proteomic data were searched against the mouse proteome (uniprot reviewed sequences downloaded 14 April 2023) using directDIA approach with Spectronaut software (version 18) following manufacture instructions.

Using this modified method, we detected β -catenin in the LC-MS/MS data (New Table S4, row 1196). Several previous protein candidates, including the NuRD components, were also present in the new data. To make it clear, we picked up and displayed these proteins from Table S4 using a heatmap (Response Figure 6). We have replaced the LC-MS/MS data with the new results in the revised manuscript (new figure 5A and S5A). Furthermore, GO analysis of enriched Bcl11b-interacting proteins showed that the NuRD complex, which mediates epigenetic modification and chromatin remodeling, was enriched (new Figure 5A). These results are consistent with our previous data.

Figure for reviewers removed

Sorry for missing search parameters and analysis information. We have added a detailed description of LC-MS/MS protein identification in pages 23-26, lines 499-549. During the pulsar search setting, the search parameters were as follows: (i) full tryptic specificity was required; (ii) PSM FDR, peptide FDR and protein group FDR were all 0.01; (iii) Minimum number of amino acids allowed for a peptide is 7; (iv) up to two missed tryptic cleavage sites were allowed; (v) carbamidomethylation (C) were set as the fixed modifications; (vi) the correction factor of precursor ion and fragment ion mass tolerances were set at 1; and (vii) variable protein modifications were allowed for methionine oxidation and protein N-terminal acetylation. Detected peptides and proteins were controlled in 1% FDR by q-value. Peak ms1 area was used to quantify protein group. Precursor and protein posterior Error Probability (PEP) cutoff were 0.2 and 0.75, respectively.

The different abundance analysis was conducted using R (version 4.3.1). All proteins with missing values were filtered and the remaining data were log2 transformed for heteroscedasticity correction and normalization of data distribution. Following data preprocessing, the limma package (version 3.56.2) was used to identify differentially abundant proteins between the IgG and IP contrast (van Ooijen *et al.*, 2018). The design formula was set using *model.matrix* function. *lmFit* function fits the linear model and calculates the t-statistic for each protein. *eBayes* function conducts the empirical Bayes test for t-statistic adjustment and returns

log₂FoldChange, P value, and adjusted P value with Benjamini–Hochberg correction. Proteins that are significantly enriched in the IP group (log₂FoldChange >1.95 and P.adjusted <0.00025) were subjected to Gene Ontology enrichment analysis.

Besides the LC-MS/MS data, the interaction between ectopic Bcl11b and endogenous β -catenin was detected by immunoblotting, and the Bcl11b- β -catenin interaction was confirmed by SPR assay using purified proteins (Fig. 4F-G). In addition, the endogenous β -catenin-Tcf4 interaction was decreased in Bcl11b KO crypts, also supporting the promoting role of Bcl11b in Wnt signaling (figure 4H).

Referee #2:

General summary and opinion: This study by Li et al is the description of a comprehensive data set that shows that *Bcl11b*, a gene that is downregulated upon BMP pathway activation (for example upon treatment with endogenous BMP ligands), act as a transcriptional regulator of WNT target genes. It is nicely shown that *Bcl11b*-deficient cells have lower expression of WNT target genes, whereas *Bcl11b*-overexpressed cells have higher Wnt target gene expression. Mechanistically, the authors on the one hand propose a direct interaction of *Bcl11b* with WNT components B-Cat/TCF4 for Wnt pathway specific aspects, on the other hand suggest a more general transcriptional regulation of *Bcl11b*, through modulating of the NuRD complex. Furthermore, the mechanistic work is supported by sturdy in vivo data showing that *Bcl11b* is important for reparative mechanisms in the intestine (radiation of small intestine, and using DSS-colitis in the colon), as well as potentially relevant for tumorigenesis. Indeed showing an important physiological role for *Bcl11b* in the intestinal stem cell system. Overall, the study is comprehensive and technically sound, and sufficient replicates etc. was used. The data is also presented clearly.

Response: Thank you for your positive comments.

Specific major concerns:

- BMP regulating *Bcl11b* expression: The strongest evidence that this is a direct regulatory mechanism actually only comes from Fig. 1H (Smad occupancy at gene). All the data before just shows that *Bcl11b* behaves similar to 'stem cell' genes, and the changes (increase/decrease) can be a consequence of a change in # of stem cells (like in the *Alk3* KO (similar to their previous study), that will get an expansion of the stem cell zone). The authors could do a short term (1/4/8h or so) BMP4 treatment to show rapid repression (preferably prior to changes in Wnt genes, as this would be a consequence of the changed *Bcl11b* expression such as the authors propose), which would support direct target gene hypothesis. As is, there's not sufficient data to support that *Bcl11b* is a bona fide BMP-repressed gene.

Response: Thank you for your great suggestion. We have performed the time course of BMP4 treatment as you suggested and found that *Bcl11b* expression decreased very quickly. The response time of *Bcl11b* is similar to the BMP4 target gene *Id1* (Response Figure 7, new Figure S1B). We have added these results in the revised manuscript.

Figure for reviewers removed

Fig. 1H is further a bit confusing as this is ChIP data from sorted Lgr5 cells (which should not have active BMP signaling, nor is this a cell where Bcl11b is repressed (see rest of the figure!)). It would be helpful to have Smad1/4 negative genes as a control qPCR to show that this is specific for the site.

Response: Thanks for your insightful suggestion. The ChIP data of Smad1 and Smad4 were from Lgr5⁺ (both Lgr5 high and low) cells, not just Lgr5 high cells. As Bcl11b is absent in Lgr5 negative cells, they were not used for this assay. The detailed method was added in the text (page 29, lines 625-634). In brief, the organoids derived from Lgr5-EGFP-IRES-creERT2 mice were cultured with ER medium for 4 hrs. Then, 1,000,000 Lgr5⁺ cells were sorted from the organoids for each sample. We agree that BMP signaling is low in Lgr5⁺ cells, but it is still active (Qi et al, 2017). To confirm the specific binding, we did the negative control qPCR as you suggested. We have included the Smad1/4-unbound region as a negative control (Response Figure 8, new Fig. 1H), as you suggested. All primer sequence was list in the new Table S1.

Figure for reviewers removed

- Fig 2 and throughout (also the methods). The Lgr5 mouse line is a mosaic line where only ~30-40% expresses GFP. In the methods it is described that random parts were assessed for GFP levels during microscopy, however, this could lead to randomly determining positive vs negative GFP sites. Fig. 2A shows complete absence, but this is perhaps more likely due to negative patch (as the qPCR shows just a 50% reduction in 2B).

Response: As the reviewer pointed out, indeed only partial ISCs can be labeled with

GFP in this line. Therefore, in addition to the image data in Figure 2A, we also used FACS to quantify the Lgr5⁺ cells (New figure S1E). The data in new Figure S1E showed that Lgr5⁺ cells decreased from 7% to 2%. As the intensity of GFP was also decreased, this decrease may result in the phenomenon that GFP⁺ cells seemed to disappear in the images, but cells with weak GFP near the scale bar were still quantified. Also, as qPCR is very sensitive, a low level of Lgr5 mRNA in cells was detected in Figure 2B. Together, we measured the Lgr5 expression from various angles, and all the results indicated that *Bcl11b* depletion resulted in ISC decrease.

Related to this figure, the authors suggest no changes were found in Paneth cells. However, changes in PCs take longer (4-6 weeks after deletion) to be observed. Original Sato Paneth cell 'niche' paper (Sato et al 2011, Nature) shows deletion of certain genes takes 6 weeks to observe reduction of Paneth cells. I recommend deleting *Bcl11b* and look 4-6 weeks later.

Response: Thank you for pointing this out, and we measured Paneth cell number in a long-term time point as you suggested, including 4 weeks and 8 weeks. There are no changes in Paneth cells upon *Bcl11b* deletion (Response Figure 9). We have included the result of 4 weeks in the new Figure S1D.

Figure for reviewers removed

- Specificity of *Bcl11b* in regulating Wnt: ChIP-seq identified ~7000 genes in ISCs to be under the control of *Bcl11b*. To me, this is basically half if not more of all genes expressed in ISCs (genome = 20k genes, ~12-15k expressed in specific cell types). As

ISCs are high Wnt condition, it's likely to then find strong overlap. To make this specific, the authors should pick something that's also high in ISCs (such as Notch-Hes1 and its target genes), and show that Bcl11b does not regulate those genes (or similar line of reasoning).

Response: Thank you for the insightful comments. We have re-analyzed our ChIP-seq data with more restricted parameters and identified 1863 Bcl11b binding sites in 1729 genes, a number similar to the Bcl11b targets reported in breast cancer cells by Cai's group (Bai *et al.*, 2022). Following your suggestion, we checked the Notch-Hes1 target *Hey1* and *Heyl*, and found no binding of Bcl11b (Response Figure 10, new Figure S4F).

Figure for reviewers removed

To me, a more plausible conclusion is that Bcl11b is just important for gene expression, facilitating whatever signaling takes place (high Wnt in ISCs in this case). Don't get me wrong, I really like the paper, I just find that this specificity is in contrast to a gene that seems to control gene expression so broadly. Furthermore, such 'aspecific' hypothesis also fits better with the proteomics data (that didn't identify Bcat/TCF4, but found more broad partners in the nucleus), rather than the OE interaction data with B-cat (again a negative other relevant TF would be good as a negative control).

Response: Thank you for thoughtful comments. As reviewer #1 also raised concerns about the absence of β -catenin in our previous LC-MS/MS data, we tried to solve this problem by increasing the Bcl11b-overexpressed organoids number from one 24-well plate to two entire 24-well plates per one sample; the immunoprecipitants-beads conjugations were washed twice with 50mM NH_4HCO_3 instead of five times with H_2O ; peptide supernatants sample were analyzed using a Vanquish Neo UHPLC system (Thermo Fisher Scientific), directly interfaced with an Orbitrap Astral mass spectrometer; all proteomic data were searched against the mouse proteome (uniprot reviewed sequences downloaded 14 April 2023) by directDIA approach using Spectronaut software (version 18) following manufacture instructions.

Using the improved method, we detected β -catenin in the LC-MS/MS data (New Table S4, row 1196), as well as several previously identified proteins, including the NuRD components. To make it clear, we picked up and displayed these proteins from

the table S4 using heatmap (Response Fig 6), which shows that β -catenin and NuRD components were enriched in the IP group. We have replaced the LC-MS/MS data with new results in the revised manuscript (new Figure 5A and S5A).

GO cellular component analysis of Bcl11b-interacting proteins showed that the NuRD complex was enriched (new Fig. 5A). These results support our conclusion.

The interaction between ectopic Bcl11b and endogenous β -catenin was detected by immunoblotting, and the Bcl11b- β -catenin interaction was confirmed by SPR assay using purified proteins (Figure 4F-G). In addition, the endogenous β -catenin-Tcf4 interaction was decreased in Bcl11b KO crypts, also supporting the promoting role of Bcl11b in Wnt signaling (figure 4H). Functionally, Bcl11b KO decreased crypt length induced by adenovirus-mediated expression of Rspo-1 (Fig 4D). As per your suggestion, we added other relevant TFs, YAP and HES1, as negative controls (Response Figure 11, new Figure S4A). Response Figure 11 showed that there is no interaction between Bcl11b and YAP or HES1. Consistently, we did not find interaction between Bcl11b and YAP or HES1 in our LC-MS/MS analysis (Table S4). These data together support the important and specific role of Bcl11b in Wnt signaling.

Figure for reviewers removed

Of note, my first 2 points are important to address experimentally, or verify method-technically. My 3rd point is definitely open to interpretation, but feel should be addressed as the authors currently claim a Wnt-specific role and this is not fully supported by the data with a lack of negative controls. The lack of BMP/TGF β /TNF reporter line activity is not sufficient, rather a Yap or Hes1 reporter would indeed be more relevant to ISC signaling state.

Response: Thank for your suggestion. We have examined Bcl11b's effect on the Notch reporter (Rbpj-luciferase) and Hippo reporter (Cyr61-luciferase). As shown in revised Figure S3D-E, Response Figure 12, Bcl11b did not affect the Notch and Hippo signaling.

Figure for reviewers removed

Combining the results from reporter data (new Figure S3A-E), the absence of Bcl11b binding on Notch target genes (new Figure S4F), the lack of detection of HES1 and YAP in LC-MS/MS result (new Table S4), and the absence of interaction between Bcl11b and HES1 or YAP (Response Figure 11, new Figure S4A), we conclude that Bcl11b specifically regulates Wnt pathway.

Minor concerns:

- Please provide high magnification of Bcl11b staining (Fig. 1G) to see stem-cell / Paneth cell zone.

Response: Per your suggestion, we have included the high magnification of Bcl11b staining in new Figure 1G (Response Figure 13).

Figure for reviewers removed

- Throughout: please when using statistical analysis of low n, one cannot assume normal distribution (so use non-parametric tests)

Response: Thanks for your comments. We have carefully re-analyzed our data. Parametric tests were chosen when data was normally distributed based on Shapiro-Wilk test. Normally distributed data were analyzed using parametric t-test with Welch's correction, one-way ANOVA with Tukey's multiple comparisons test, two-way ANOVA with Tukey's multiple comparisons test. Specific tests and significance levels are indicated in corresponding revised figure legends.

- Please use a mouse = 1n, example: Fig. 2A has many data points, but I don't think these are different mice (just individual crypts). Same for Fig. 2C/2G/2I, n= a well not 1 organoid, and preferably from different mice.

Response: Thanks for your suggestions. We have included how many fields/crypts/organoids were analyzed from three biological repeats in corresponding revised figure legends.

- Ensure X-ray was used (normally it's gamma-radiation machines that is used in patients for therapy, not the X-ray to image). That said, x-ray is gamma-radiation type so if tuneable this could have been used.

Response: We used a Rs2000 irradiator (Radsource) to do the X-ray irradiation in Laboratory Animal Resources Center of Tsinghua University. The X-ray from Rs2000 irradiator is equal to the gamma-Ray from Cs137. We have added the information in the revised Methods section (page 17, lines 359-360).

- Wording: please make sure to state when using ectopic overexpression of a plasmid to determine its role, or when looking at interactions (IP). Compared to looking at endogenous levels. Similarly; OE and tagged IP experiments certainly are proof that proteins can interact, but also not proof that they are in a complex under normal conditions or in ISCs.

Response: We have checked and changed these descriptions in the revised manuscript (page 9, line 193-194; page 10, line 198; page 11, line 226).

Referee #3:

The manuscript by Li and team explores the role of Bcl11b in intestinal stem cells. In figure 1, the authors use multiple lines of evidence to demonstrate that Bcl11b is repressed by BMP signaling and enriched in stem and progenitor cells in the crypt. In figure 2, Bcl11b is shown to be required for a full number of Lgr5+ ISCs both in vivo and in organoid models. Gain and loss of function systems show that Bcl11b expression correlates with ISC marker expression. In figure 3, irradiation and DSS models are used to show that Bcl11b is important for the intestine to resist damage from these treatments. In figure 4, cell culture assays and co-IP experiments point to interactions between Bcl11b and TCF and Bcatenin. CHIPseq analysis shows overlap between Bcl11b and TCF4 and Bcatenin binding sites, further supporting a role in regulating the WNT signaling pathway. In figure 5, the authors dig deeper into how Bcl11b may be regulating genes and find that BCL11B interacts with components of the NURD complex, and that loss of BCL11B leads to changes in ATAC and H3K27ac and H3K27me3 that correspond to reduced WNT target gene expression in the BCL11B mutant. Finally, figure 6 provides evidence that BCL11B supports cancer growth, as might be expected based upon its support of the WNT pathway.

This work is generally well-executed and provides novel insights into how the WNT pathway is regulated in normal intestinal homeostasis and in colon cancer. The authors frequently use both gain and loss of function genetic approaches and state-of-the-art techniques to make their discoveries. While there are some important concerns to be addressed to improve the rigor and readability of the work, the work will certainly make exciting contributions to the field.

Response: Thank you for your positive comments.

Concerns:

The data haven't been made available to the reviewers and the number of replicates for many experiments is unclear. (for example, CHIPseq)

Response: Sorry for missing the information. We have added number of replicates in each figure legend and included links in the Data Availability section (page 33, line 708-712) to allow reviewers' examination:

To review GSE255138: go to <https://www.ncbi.nlm.nih.gov/geo/query/acc.cgi?acc=GSE255138>. Enter token wpyhgayulvitfcp into the box.

To review GSE247044: go to <https://www.ncbi.nlm.nih.gov/geo/query/acc.cgi?acc=GSE247044>. Enter token idstiqwwnheblan into the box.

Please indicate where the datasets arise from in the figures and figure legends. For example, figure 1C was from a previous publication, but it's not so easy to find the data source. It would help readers to have a more direct connection to the data.

Response: Sorry about it. We indicated the data source in the data availability

statement. To make it clearer, we have indicated the source in the revised figure legend now. Fig 1C is from GSE186917, Fig 1B from GSE247044, Fig 6A and S7A-B from GSE178341.

Please address this paper that finds opposite results in a mouse model of colon cancer: <https://pubmed.ncbi.nlm.nih.gov/25827435/>

Response: In the mentioned paper, Sakamaki et al. reported that Bcl11b represses intestinal epithelium regeneration and adenoma formation through inhibiting the Wnt/ β -catenin pathway. The discrepancy may be owing to different mice used in the studies: conventional *Bcl11b*^{+/-} mice were mainly utilized in their study, while we employed inducible intestinal epithelium-specific knockout mice. They observed conventional *Bcl11b*^{KO/+}; *Apc*^{min/+} mice developed more adenoma than the *Apc*^{min/+} mice. Given that Bcl11b is critical for T cell development, the observed effect of Bcl11b heterozygosity on tumor formation may possibly attribute to T cell deficiency. These authors reported that some of the Wnt target genes, including *Myc*, *Ccnd1* and *Jun*, were elevated in Lgr5⁺ cells from *Lgr5-CreERT2*; *Bcl11b*^{fl/fl} mice two months after 4-OHT induced *Bcl11b* knockout. However, more Wnt target genes including *Axin2*, *Sp5*, *Lef1*, *Mmp7*, *Tnfrsf19* decreased after *Bcl11b* knockout, which is consistent with our data. We have added this in the Discussion (pages 14, lines 299-311).

Cre-only controls should be used in the regeneration assays to test for off-target effects. It is not clear if these controls are in place.

Response: Thank you for pointing it out. Indeed, all control mice we used were referred to *Vil-CreERT2*; *Lgr5-GFP-IRES-creERT2* mice that were injected with the same amount of tamoxifen to avoid off-target and possible toxicity-induced phenotypes. We have clarified the genotype of control mice in the revised manuscript (page 6, lines 117-119) and related figure legends.

It does not appear that there is a description of the overexpression construct. This is one example of several in the methods section that appears to lack key experimental details and should be improved so that others can try and reproduce the work.

Response: Sorry for the missing information. We have now added the experimental details (pages 18-19, 21,23-26, 28-30). In particular, the Flag-Bcl11b-expressing lentivirus was described in pages 18-19.

Reference:

Baghdadi MB, Ayyaz A, Coquenlorge S, Chu B, Kumar S, Streutker C, Wrana JL, Kim TH (2022) Enteric glial cell heterogeneity regulates intestinal stem cell niches. *Cell Stem Cell* 29: 86-100 e106

Bai H, Lin M, Meng Y, Bai H, Cai S (2022) An improved CUT&RUN method for regulation network reconstruction of low abundance transcription factor. *Cell Signal* 96: 110361

Koppens MAJ, Davis H, Valbuena GN, Mulholland EJ, Nasreddin N, Colombe M, Antanaviciute A, Biswas S, Friedrich M, Lee L *et al* (2021) Bone Morphogenetic Protein Pathway Antagonism by Grem1 Regulates Epithelial Cell Fate in Intestinal Regeneration. *Gastroenterology* 161: 239-254 e239

Li Q, Sun Y, Jarugumilli GK, Liu S, Dang K, Cotton JL, Xiol J, Chan PY, DeRan M, Ma L *et al* (2020) Lats1/2 Sustain Intestinal Stem Cells and Wnt Activation through TEAD-Dependent and Independent Transcription. *Cell Stem Cell* 26: 675-692 e678

Li Y, Liu Y, Chiang YJ, Huang F, Li Y, Li X, Ning Y, Zhang W, Deng H, Chen YG (2019) DNA Damage Activates TGF-beta Signaling via ATM-c-Cbl-Mediated Stabilization of the Type II Receptor TbetaRII. *Cell Rep* 28: 735-745 e734

Li Y, Liu Y, Liu B, Wang J, Wei S, Qi Z, Wang S, Fu W, Chen YG (2018) A growth factor-free culture system underscores the coordination between Wnt and BMP signaling in Lgr5(+) intestinal stem cell maintenance. *Cell Discov* 4: 49

Liu L, Wang Y, Yu S, Liu H, Li Y, Hua S, Chen YG (2023a) Transforming Growth Factor Beta Promotes Inflammation and Tumorigenesis in Smad4-Deficient Intestinal Epithelium in a YAP-Dependent Manner. *Adv Sci (Weinh)* 10: e2300708

Liu Y, Chen YG (2020) Intestinal epithelial plasticity and regeneration via cell dedifferentiation.

Cell Regen 9: 14

Liu Y, Huang M, Wang X, Liu Z, Li S, Chen YG (2023b) Segregation of the stemness program from the proliferation program in intestinal stem cells. *Stem Cell Reports* 18: 1196-1210

Qi Z, Chen YG (2015) Regulation of intestinal stem cell fate specification. *Sci China Life Sci* 58: 570-578

Qi Z, Li Y, Zhao B, Xu C, Liu Y, Li H, Zhang B, Wang X, Yang X, Xie W *et al* (2017) BMP restricts stemness of intestinal Lgr5⁺ stem cells by directly suppressing their signature genes.

Nat Commun 8: 13824

Sato T, Ishikawa S, Asano J, Yamamoto H, Fujii M, Sato T, Yamamoto K, Kitagaki K, Akashi T, Okamoto R *et al* (2020) Regulated IFN signalling preserves the stemness of intestinal stem cells by restricting differentiation into secretory-cell lineages. *Nat Cell Biol* 22: 919-926

van Ooijen MP, Jong VL, Eijkemans MJC, Heck AJR, Andeweg AC, Binai NA, van den Ham HJ (2018) Identification of differentially expressed peptides in high-throughput proteomics data.

Brief Bioinform 19: 971-981

Wang S, Chen YG (2018) BMP signaling in homeostasis, transformation and inflammatory response of intestinal epithelium. *Sci China Life Sci* 61: 800-807

Wang Y, Huang M, Mu X, Song W, Guo Q, Zhang M, Liu Y, Chen YG, Ge L (2024) TMED10-mediated unconventional secretion of IL-33 regulates intestinal epithelium differentiation and homeostasis. *Cell Res* 34: 258-261

Zhao B, Qi Z, Li Y, Wang C, Fu W, Chen YG (2015) The non-muscle-myosin-II heavy chain Myh9 mediates colitis-induced epithelium injury by restricting Lgr5⁺ stem cells. *Nat Commun* 6:

7166

Dear Ye-Guang,

Thank you for submitting a revised version of your manuscript. I sincerely apologise for the protracted assessment process due to delays in referee report submission. We have now received input from two of the original reviewers, who now find that their previous concerns have been addressed satisfactorily and broadly recommend acceptance of the manuscript. Therefore, there now remain only a few mainly editorial points that need addressing before I can extend official acceptance of the manuscript:

1. Please submit a complete author checklist, which you can download from our author guidelines (<https://www.embopress.org/pb-assets/embo-site/EMBO%20Press%20Author%20Checklist-1642513524327.xlsx>). Please insert information in the checklist that is also reflected in the manuscript. The completed author checklist will also be part of the Review Process File.
2. Please upload the main and EV figures as individual production quality figure files in the .eps, .tif, or .jpg format (one file per figure).
3. Please reduce the number of keywords to maximum five.
4. Please check that the funding information is correct and identical both in the manuscript and our online system. Currently, grant 023YFA1800603 is missing in our online system.
5. CRedit has replaced the traditional author contributions section because it offers a systematic, machine-readable author contributions format that allows for more effective research assessment. Please remove the Authors Contributions from the manuscript and use the free text boxes beneath each contributing author's name in our online submission system to add specific details on the author's contribution. More information is available in our guide to authors.
6. Please rename "Conflict of interests" section into "Disclosure and competing interests statement" (further info: <https://www.embopress.org/page/journal/14602075/authorguide#conflictsofinterest>).
7. Please rename Table S1 into Table EV1; Tables S2, S3 and S4 should be renamed Dataset EV1, 2 and 3. All files need legends added to the excel files. For the datasets, the legends should be added in a separate tab.
8. Please note that the tables should be called out in the sequential order in the manuscript text.
9. The file with the supplementary figures and tables should be renamed Appendix and uploaded as a PDF. Page numbers should be added to the table of contents. The nomenclature should be corrected to "Appendix Figure S1" etc. The legends for the supplementary tables should be removed from the appendix and added directly to the excel files of the corresponding tables/datasets.
10. Please remove the list of abbreviations from the manuscript.
11. In our standard source data check, we have noted unexplained numerical duplications in the source data. I have attached the corresponding files with the detected duplications labelled in colour. Please take a look and correct as needed. A brief explanation would be very helpful.
12. Our data editors have flagged the following issues in figure legends that need correcting:
 - Please provide the exact p values in the legends of figures 1a, f-h; 2a-j; 3a-i; 4a-e, k; 5f; 6c-d, f-g.
 - Please indicate the statistical test used for data analysis in the legends of figures 4j; 5a, d.
 - Please note that in figures 1a, f-h; 4a-e, k; 5f; there is a mismatch between the annotated p values in the figure legend and the annotated p values in the figure file that should be corrected.
 - Please define the box plots in terms of minima, maxima, centre, bounds of box and whiskers, and percentile in the legends of figures 1g; 3d-e, g; 4d; 5f; 6b, d-e, g.
 - Please add information on the nature and number of replicates in the legends of figures 1e; 6e.
 - Please define the error bars in the legends of figures 3a-c, f, h-i; 6c, f.
 - Please define the scale bar for figure 1d.
13. Papers published in The EMBO Journal are accompanied online by a 'Synopsis' to enhance discoverability of the manuscript. It consists of A) a short (1-2 sentences) summary of the findings and their significance, B) 3-4 bullet points highlighting key results and C) a synopsis image that is 550x300-600 pixels large (width x height, jpeg or png format). You can either show a model or key data in the synopsis image. Please note that the image size is rather small and that text needs to be readable at the final size. Please send us this information together with the revised manuscript.

With best wishes,

Ieva

Ieva Gailite, PhD
Senior Scientific Editor

The EMBO Journal
Meyerhofstrasse 1
D-69117 Heidelberg
Tel: +4962218891309
i.gailite@embojournal.org

We realize that it is difficult to revise to a specific deadline. In the interest of protecting the conceptual advance provided by the work, we recommend a revision within 3 months (16th Dec 2024). Please discuss the revision progress ahead of this time with the editor if you require more time to complete the revisions.

Referee #1:

The authors have added new data and clarifications that address my major concerns. I agree with Reviewer 2 that even with the newly filtered ChIPseq data, the effect suggests Bcl11b is also broadly involved in regulation of gene expression.

Referee #2:

The authors have fully answered my questions, usually with additionally performed experiments or optimized analysis with higher stringency. I don't have any further concerns.

The authors addressed the minor editorial issues.

Dear Ye-Guang,

Thank you for addressing the final editorial points. I sincerely apologise for the delay in communicating the decision due to the high number of submissions we receive at the moment. I am now pleased to inform you that your manuscript has been accepted for publication.

Before we forward your manuscript to our publishers, I would like to propose some edits in the manuscript title, abstract and synopsis (please see below and the attached manuscript text file). I have also written a short blurb that will accompany the title of your manuscript in our online system. Please let me know if any corrections or adjustments are needed.

Title:

BMP suppresses Wnt signaling activation via the Bcl11b-regulated NuRD complex to maintain intestinal stem cells

Blurb:

The chromatin regulator Bcl11b promotes intestinal epithelium regeneration and the onset of colon cancer by activating Wnt signaling and inhibiting NuRD complex activity.

Synopsis:

Both Wnt and BMP signaling regulate maintenance of Lgr5+ intestinal stem cells, but their interplay remains incompletely understood. This study shows that the BMP-suppressed chromatin regulator Bcl11b promotes Lgr5+ intestinal stem cell maintenance and epithelial regeneration by enhancing Wnt signaling.

- BMP signaling suppresses Bcl11b expression.
- Bcl11b ensures Lgr5+ intestinal stem cell maintenance and epithelial regeneration upon injury.
- Bcl11b activates Wnt signaling by facilitating the β -catenin-TCF4 interaction and inhibiting the chromatin remodelling via the NuRD complex.
- Bcl11b promotes colon cancer formation.

Finally, we would like to promote your manuscript among the Chinese readership. Therefore, we would like to invite you to prepare a short summary of the manuscript in Chinese (1500-2000 Chinese characters), which we will promote on the WeChat platform 'BioArt' with more than 610,000 followers.

If you are interested in this opportunity, we recommend covering the article very close to its online publication date. Thus, ideally we would very much appreciate if you could send us a draft within the next 7 working days. Please let us know whether or not you would be interested in contributing such a short summary in Chinese.

I have included below some general guidelines on how to prepare a summary and a link to recent examples for your reference. Please let me know if you have any questions about this.

If you have any questions, please do not hesitate to contact the Editorial Office. Thank you for this contribution to The EMBO Journal and congratulations on a nice study!

With best wishes,

Ieva

Ieva Gailite, PhD
Senior Scientific Editor
The EMBO Journal

Meyerhofstrasse 1
D-69117 Heidelberg
Tel: +4962218891309
i.gailite@embojournal.org

General WeChat Summary Guidelines

1. These summary articles are meant to be targeting general audience so please limit the use of specialized technical terms, acronyms and jargon.
2. A summary usually starts with brief background information of the reported work, which is followed by explaining the findings in some detail, and ends with a short review of the conclusions as well as the implications of the work and future directions for the research.
3. The summary should at least contain one graphical item, such as a scheme or a figure from the paper.
4. Please provide ONE SINGLE document containing all text and graphical materials, ideally as a Word.docx or .doc file. Please DO NOT provide the document as a .pdf file.
5. Please DO NOT publicly release the document before the paper is officially published online.

Summary Examples

EMBO J | 罗招庆/欧阳松应揭示谷酰胺脱氨酶MvcA的去泛素化功能

EMBO J | 王松灵院士团队揭示组织内应力调控大型哺乳动物乳恒牙替换的新机制